# Whole genome study and construction of SHERLOCK detection method for endemic strains of *Burkholderia pseudomallei* in Hainan based on third-generation sequencing

Junjie Hu,[1,2] Shanshan Xu,[1] Zeng Zeng,[3] Wei Gong,[1] Weihua Xu,[1] Zhichao Ma,[1] Shengmiao Fu,[4] Linhai Li,[5] Bin Xiao,[5] Xinping Chen[1,2]

**ABSTRACT**   *Burkholderia pseudomallei (Bp)* is a gram-negative bacterium found in soil and surface water. It is also the pathogen that causes melioidosis disease in humans and animals. This study aimed to obtain the whole genome sequence of the endemic strain of *Bp* in Hainan, using third-generation sequencing (TGS) technology, and elucidate the genome structure, function, and genetic evolution. Additionally, the study aimed to achieve rapid and specific identification of these endemic strains using specific high-sensitivity enzymatic reporter unlocking (SHERLOCK) detection technology, providing a new strategy for the early diagnosis of melioidosis. Utilizing the PacBio platform for TGS technology, we completed whole genome sequencing of 16 *Bp* strains from Hainan. High-precision and complete genome sequences were obtained through quality control and genome assembly of the sequencing data. Additionally, we established a nucleic acid detection technology platform based on SHERLOCK, which could be completed from nucleic acid extraction to result reading within 1–2 hours, demonstrating good sensitivity and specificity (both are 100%). The lateral chromatography strip method does not require special equipment and holds promise as an immediate screening method for the early diagnosis of melioidosis.

**IMPORTANCE**   Melioidosis is a highly pathogenic infectious disease caused by a gram-negative bacterium of *Burkholderia pseudomallei (Bp)*. The traditional gold standard for diagnosing melioidosis is still isolation and culture from clinical samples. Although this method has high specificity, it has low sensitivity and is time-consuming, which often leads to misdiagnosis or missed diagnosis of melioidosis, affecting subsequent treatment. In this study, recombinase polymerase amplification technology and clustered regularly interspaced short palindromic repeats/Cas13a technology were combined to establish the Specific High-sensitivity Enzymatic Reporter Unlocking detection technology, which can achieve rapid and accurate identification of *Bp*, providing a new method for the early diagnosis of melioidosis.

**KEYWORDS**   *Burkholderia pseudomallei*, whole genome sequence, third generation sequencing, SHERLOCK, early diagnosis

*B*urkholderia pseudomallei *(Bp)* is a gram-negative bacterium commonly found in soil and water in tropical and subtropical regions, especially in northern Australia and Southeast Asia. In a genomic sequencing study of *Bp* from multiple countries, including 30 countries in Oceania, Asia, Africa, and Central and South America, the results showed that there were significant genetic differences between strains from Australia and Asia, and that strains from Australia had longer phylogenetic branches. Combined with pan-genome analysis data, the final results show that Australia is an early reservoir for the current global population of *Bp* and is spreading to other regions (1). It actively

**Peer Reviewer** Narisara Chantratita, Mahidol University, Bangkok, Thailand

Address correspondence to Xinping Chen, chenxinping52@126.com, or Bin Xiao, xiaobin2518@163.com.

The authors declare no conflict of interest.

invades cells through endocytosis, grows and multiplies within endocytic vesicles, and can infect nearly any organ in the body (2, 3). *Bp* demonstrates microecological stability, but environmental disturbances such as rainstorms, typhoons, or tsunamis can disrupt this balance, leading to disease spread and potential outbreaks. Annually, approximately 165,000 people globally are infected with *Bp*, resulting in about 89,000 deaths (4, 5). Hainan Island, China, situated in a tropical region, has a closed geographical location that fosters the growth and reproduction of *Bp*. The island's unique climate conditions also create favorable circumstances for *Bp* outbreaks. Reports of melioidosis in Hainan have been increasing in recent years (6). The local population is highly susceptible to *Bp*, with most infections occurring through contaminated skin, and some through inhalation or ingestion of contaminated water. Notably, farmers show a higher incidence of *Bp* infection, likely due to their greater exposure to contaminated soil and water (7).

The genome of *Bp* is notably diverse, with high guanine-cytosine (GC) content and numerous repetitive sequences, making it one of the largest and most complex bacterial genomes sequenced to date (8). The world's first strain of *Bp* (no. K96243) to undergo full genome sequencing was a patient from Thailand. In 2004, researchers at Sanger Laboratory in the UK completed its sequencing, assembly, and functional annotation. K96243 was used as an international standard strain, and its genomic information was uploaded to the National Center for Biotechnology Information (NCBI) in the USA to provide a reference for subsequent strain genomic research (9). Although many *Bp* genome sequences are stored in the NCBI, research on the whole genome of *Bp* primarily relies on second-generation sequencing technologies (10–12). Short-read sequencing often fails to provide complete genome and gene location information, and it struggles to differentiate between the nuclear genome and plasmids. In contrast, third-generation sequencing (TGS) technologies offer more accurate genome sequencing of *Bp*. For example, Teng compared whole genome sequencing (WGS) results of *Bp,* using the Illumina platform, with those obtained from the PacBio single-molecule real-time sequencing platform. The study demonstrated that PacBio sequencing, followed by *de novo* assembly, yielded a complete genome sequence for *Bp* without gaps or mismatches (13). Additionally, Ghazali used the PacBio RS II sequencing platform to sequence *Bp* strains from Malaysia, achieving a genome assembly completeness of 99.1% to 99.7%, as assessed by the Burkholderiales_odb5 lineage database (14). Utilizing the PacBio platform, we performed structural, functional, and comparative genomics analyses on endemic *Bp* strains in Hainan to further investigate the relationships among genome structure, function, and genetic evolution.

Currently, the isolation, culture, and identification of *Bp* strains remain the diagnostic gold standard. However, this method, while highly specific, is limited by low sensitivity and long processing times, hindering rapid early diagnosis. In 2017, Feng Zhang et al. demonstrated that clustered regularly interspaced short palindromic repeats (CRISPR)-associated protein 13a (Cas13a) can target RNA in mammalian cells and nonspecifically degrade other RNAs, including reporter RNA, through "collateral cleavage" (15). Building on this, Specific High-Sensitivity Enzymatic Reporter Unlocking (SHERLOCK) detection technology was developed, integrating recombinase polymerase amplification (RPA), T7 transcription, Cas13a detection, and fluorescence signal collection. SHERLOCK has shown great potential in diagnosing various pathogens (16), such as detecting Zika virus in blood or urine within hours, distinguishing genetic sequences of Zika strains from the USA and Africa, identifying specific bacterial types, and detecting resistance genes (17–19). Recently, research teams from China and Thailand have also reported using CRISPR-Cas12a technology to achieve rapid detection of *Bp*. However, the CRISPR RNA (crRNA) sequences used in the research are different, resulting in certain differences in the range of *Bp* populations detected (20, 21). Therefore, based on genomic sequence analysis of endemic *Bp* strains in Hainan, this study was aimed to design crRNA targeting specific and conserved gene sequences in the *Bp* strains genome, construct a SHERLOCK reaction system, and establish two technical platforms—fluorescence method and lateral

chromatography test strip—to verify the method's accuracy and reliability for the rapid diagnosis of endemic *Bp* strains in Hainan.

## MATERIALS AND METHODS

### Sample collection

A total of 53 clinical strains were used in this study, comprising 42 of *Bp*, 2 of *Burkholderia cepacia*, 2 of *Klebsiella pneumoniae*, 2 of *Pseudomonas aeruginosa*, 1 of *Escherichia coli*, 2 of *Staphylococcus aureus*, and 2 of *Streptococcus pneumoniae*. And all strains were stored at -80℃.

### Antibiotic susceptibility testing

Antibiotic susceptibility testing was performed according to the broth microdilution method recommended by CLSI (refer to CLSI M45-2016 for testing and interpretation). This testing covered three antibiotics: trimethoprim/sulfamethoxazole (SXT), ceftazidime (CAZ), and imipenem (IPM).

### DNA extraction and concentration determination

The Qiagen Genomic Tip 20/G kit (Qiagen, Beijing, China) was used for nucleic acid extraction from the strain, following the manufacturer's instructions precisely. DNA concentration and purity were assessed using a Qubit fluorometer (Thermo Fisher Scientific, USA) and a Nanodrop spectrophotometer (Thermo Fisher Scientific, USA). The extracted DNA was then stored at −80℃ for future use.

### Library construction and WGS

#### *Library construction and quality inspection*

First, the Covaris G-tube was used to randomly fragment the DNA sample into 15 kilobase pair (kb) fragments. Subsequently, the PacBio library preparation kit was used to remove single-stranded overhangs and repair DNA damage in the fragmented DNA. Dumbbell adapters were ligated to both the 3′ and 5′ ends of the DNA fragments, and excess fragments were degraded using exonuclease. Nucleic acid fragments were then purified with AMpure PB magnetic beads (PacBio, USA). The target fragments were recovered using the BluePippin instrument (Sage Science, USA) to create the sequencing libraries. The concentration of the libraries was quantitatively assessed using a Qubit fluorometer, while the size and suitability of the libraries were evaluated with an Agilent 2100 instrument (Santa Clara, USA).

#### *WGS*

*De novo* gene sequencing was conducted using the PacBio RS II System from Beijing Biomec Biotechnology Company. After the target library was bound to the primer and polymerase, sequencing was carried out on the Sequel II sequencer (PacBio, USA). In this study, we randomly selected 16 of 42 *Bp* strains for whole genome sequencing.

### Bioinformatics analysis

#### *Pre-processing of original sequencing data and genome assembly*

In WGS, the sequencing data initially consists of small fragments significantly shorter than the actual sequencing length. To obtain circular consensus sequencing (CCS) sequences, it is necessary to perform cross-correction of subreads within the zero-mode waveguides. Segments with fewer than four passes are removed, as a CCS accuracy greater than 99% requires at least four passes. Using SMRT Link v.8.0 software (https://pacbio.cn/support/software-downloads/), the raw data were processed with settings of

minPredictedAccuracy 0.9 and minPasses ≥5 to generate CCS sequences. Clean CCS reads were obtained by filtering out connectors and short CCS reads (length <2 kb). These clean CCS reads were then assembled into contiguous sequences (contigs) or scaffolds using Hifiasm software (https://hifiasm.readthedocs.io/en/latest/index.html). The starting site was adjusted with Circlator v.1.5.5 to complete the circularization, and Pilon v.1.22 software (https://github.com/broadinstitute/pilon/releases) was used to correct errors based on second-generation data. This process ultimately produced a high-accuracy whole genome sequence.

### Genome-wide component analysis

1. Encoding gene prediction: The Prodigal v.2.6.3 software (https://github.com/hyattpd/Prodigal) was used to predict coding genes.
2. Prediction of genomic repeats: The RepeatMasker v.4.0.5 software (https://www.repeatmasker.org/) was used to identify repetitive sequences in the genome.
3. Prediction of non-coding RNA: Non-coding RNAs were predicted using Infernal v.1.1.3 (http://eddylab.org/infernal/) for ribosomal RNA (rRNA) and tRNAscan-SE v.2.0 (http://lowelab.ucsc.edu/tRNAscan-SE/) for transfer RNA (tRNA).
4. Prediction of CRISPR: CRISPR sequences in the genome were predicted using CRISPRFinder (http://crispr.i2bc.paris-saclay.fr/Server/).
5. Prediction of prophage: Prophage regions were predicted using PhiSpy v.2.3 software (https://pypi.org/project/PhiSpy/).

### Whole genome gene function annotation

1. EggNOG database comments: The RPS-Blast tool was utilized to compare gene protein sequences with the EggNOG database (http://eggnog5.embl.de/#/app/home) to identify the most similar sequences. The annotation and classification information retrieved from the EggNOG database was then used to annotate and classify the corresponding genes in the sequenced genome.
2. Virulence factor database (VFDB) annotations: The protein sequences of the predicted genes were compared with the core data of the VFDB (http://www.mgc.ac.cn/VFs/), including experimentally confirmed virulence factors. The most similar sequences identified in the VFDB provided the annotation information for the corresponding genes in the sequenced strain's genome.
3. Comprehensive antibiotic resistance database (CARD) annotations: Protein sequences of the sequenced genes were compared with the CARD (https://card.mcmaster.ca/) to identify antibiotic resistance genes. The Resistance Gene Identifier (RGI) tool within CARD was used to find the most similar sequences, thus obtaining information on corresponding antibiotic resistance genes.

### Mapping of the genome circle

Using the genomic information obtained from assembly and prediction, the genome was visualized as a circular map using Circos v.0.66 software (http://circos.ca/software/download/circos/).

### Comparative genomics research

1. Gene family clustering and differential gene analysis: Gene family clustering is a crucial step in comparative genomics and taxonomic analysis for identifying subspecies genomes. Using the default parameters of OrthoMCL software (http://orthomcl.org/orthomcl/), homologous protein-coding sequences from 16 *Bp* strains and the reference strain K92643 were analyzed. OrthoMCL grouped these sequences into homologous clusters based on amino acid sequence similarity. The

software produced a genetic homology matrix illustrating the genetic relationships among the strains, which was visualized in a petal map. This analysis identified core gene families, consisting of homologous genes present in all strains, as well as specific gene families, which lack direct homologous genes. Additionally, differential gene analysis was performed based on the clustering results to count and compare core and specific genes among the 16 *Bp* strains and the reference strain K92643.

2. Phylogenetic analysis: The single-copy homologous gene of the sequenced strain was compared with those of the reference strain K96243. Phylogenetic relationships were studied by constructing a maximum-likelihood (ML) tree using PhyML software (http://www.atgc-montpellier.fr/phyml/). This analysis aimed to elucidate the evolutionary relationships between the species.

3. Gene collinearity analysis: Homologous gene locations on the genome were identified through protein sequence BLAST comparisons, and the collinearity relationships among genes were analyzed. Using MCScanX software (https://sourceforge.net/projects/mcscanx/), collinearity plots were generated to compare the reference strains with strain K96243.

## Establishment of RPA system

Primer design was performed using Primer Premier 6.0 (https://www.premierbiosoft.com/primerdesign/). In the CRISPR/Cas13a system, the crRNA targets specific RNA sequences. Therefore, a T7 promoter sequence (TAATACGACTCACTATAGGG) was included at the 5′ end of the forward primer. It facilitated the transcription of the RPA amplification product from DNA to RNA. From Table 1, a total of three forward primers (F1–F3) and three reverse primers (R1–R3) were designed to form nine pairs of different primers: F1+R1, F1+R2, F1+R3, F2+R1, F2+R2, F2+R3, F3+R1, F3+R2, F3+R3.

Under the same conditions of DNA concentration, amplification temperature, and amplification time, nine pairs of different primers were tested for amplification. Gel electrophoresis was used to compare the amplification efficiency, and primers with superior efficiency were selected. Adjustments were made as necessary according to the instructions for the TwistAmp Basic RPA Kit (TwistDx, London, UK, catalog number: TABASRT01KIT).

RPA amplification system (50 µL) included Primer Free Rehydration buffer (29.5 µL), DNase-Free water (11.2 µL), forward primer (2.4 µL, 10 µM), reverse primer (2.4 µL, 10 µM), template DNA (*Bp* DNA, 2 µL), and magnesium acetate (MgOAC, 2 µL, 280 mM). After adding MgOAC, the components were thoroughly mixed and subjected to brief centrifugation. The reaction mixture was promptly incubated at 39°C for 20 minutes. (Note: MgOAC should be added as the final step, and the reaction will start immediately upon addition.) The amplified RPA products were purified using the DNA Recovery Kit (TIANGEN, Beijing, China, catalog number: DP214).

For gel electrophoresis, a total of 2 µL of 6 × DNA loading buffer was mixed with 10 µL of the RPA amplification product and loaded into the sample wells. A 12 µL DNA marker was used as a reference. The electrophoresis was conducted at 125 V for 30 minutes. The progress of the bands was monitored, and electrophoresis was stopped when the bands

TABLE 1  Sequence of RPA primers[a]

| Name | Sequence (5′–3′) |
| --- | --- |
| Forward primer | F1: TAATACGACTCACTATAGGGGGCACGGCGGAGATTCTCGAATTGTC |
| | F2: TAATACGACTCACTATAGGGCACGCACGGCGGAGATTCTCGAATT |
| | F3: TAATACGACTCACTATAGGGGGCCACGCACGGCGGAGATTCTCGAATTGTC |
| Reverse primer | R1: GCAACCACAGCAACGGAAAGAGCAGA |
| | R2: CCACAGCAACGGAAAGAGCAGATTG |
| | R3: GCAACCACAGCAACGGAAAGAGCAGATTGAAG |

[a]Add T7 promoter sequence (TAATACGACTCACTATAGGG) to the 5′ end of the forward primer.

had migrated to the middle of the gel. The gel was then removed and analyzed using a gel imaging system to observe the results.

## Establishment of SHERLOCK system

### crRNA design

The crRNA for the CRISPR/Cas13a system consisted of two components: a conserved repeat region and a spacer region. For this study, the *Bp* TTS1-ORF2 sequence was used, with the detection target located between the amplified RPA fragments and avoiding overlap with the RPA primer region. The target sequence was 28 nucleotides long and was reverse complementary to the spacer region of the crRNA. The sequence of the crRNA used was as follows: GAUUUAGACUACCCCAAAAACGAAGGGGACUAAAA-CATCTCGGCCGCAAGCAACCGGTGTGGGA.

### Configuration of SHERLOCK reaction system

The SHERLOCK reaction system should be maintained at a low temperature throughout the preparation process and operated on ice. The SHERLOCK reaction system (20 µL) included the following components: 10.6 µL diethylpyrocarbonate (DEPC)-treated water, 2 µL 10× buffer, 0.8 µL ribonucleotide triphosphater, 0.6 µL T7 RNA polymerase, 1 µL ribonuclease inhibitor, 1 µL LwaCas13a (1 µM), 1 µL crRNA (1 µM), 1 µL Flu-RNA reporter/ lateral flow strip RNA reporter, and 2 µL RPA product.

### Establishment of the SHERLOCK fluorescence detection system

The reaction mixture was transferred to a quantitative polymerase chain reaction (qPCR) tube and placed in the qPCR instrument. The reaction temperature was set to 37℃, and fluorescence intensity was measured every minute to monitor changes in the fluorescence signal. Results were interpreted as follows: (i) a positive result was indicated by a significant change in fluorescence signal, with the curve displaying substantial warping or reaching a plateau; (ii) a negative result was indicated by the absence of a valid signal, characterized by a consistently flat curve.

### Establishment of SHERLOCK lateral immunochromatography strip system

The reaction mixture was transferred to a 0.2 mL PCR tube and placed in a PCR instrument. The reaction was conducted at 37℃ for 30 minutes. Following the reaction, 30 µL of DEPC-treated water was added to the centrifuge tube to bring the total volume to 50 µL. The mixture was then mixed and tested. The binding pad end of the test strip was inserted into the PCR tube, ensuring that the liquid level did not exceed the top of the binding pad. The test was allowed to proceed for 1 to 2 minutes to ensure full infiltration of the interpretation area. Once the quality control line (C line) developed color, the test strip was removed and read. Results should be interpreted within 10 minutes of the appearance of the C line, as results would be invalid after this time. Result interpretations were as follows: (i) a positive result was indicated if both the test line (T line) and C line showed red bands; (ii) a negative result was indicated if only the C line showed a red band and no red band was visible on the T line; (iii) an invalid result was indicated if no red bands were visible on both the T line and the C line, or if only the T line showed a red band and the C line was colorless.

### Analysis of the sensitivity, specificity, and limit of detection (LOD)

To evaluate the performance of the SHERLOCK detection system, we used the culture method as the gold standard. A total of 42 *Bp* strains were tested, along with 11 control strains representing common clinical bacteria, including *Burkholderia cepacia*, *Klebsiella pneumoniae*, *Pseudomonas aeruginosa*, *Escherichia coli*, *Staphylococcus aureus*, and *Streptococcus pneumoniae*. Additionally, DEPC-treated water was used as a negative control. DNA was extracted from these strains, and subsequent detection was performed

using both SHERLOCK fluorescence detection and lateral flow immunochromatographic strip detection after RPA. The results from these methods were compared to those obtained using traditional culture techniques. The sensitivity was calculated using the formula Sensitivity = True Positives / (True Positives + False Negatives) × 100%. The specificity was calculated using the formula Specificity = True Negatives / (True Negatives + False Positives) × 100%.

*Bp* DNA concentration was serially diluted from 1 ng/μL to $10^{-1}$, $10^{-2}$, $10^{-3}$, $10^{-4}$, $10^{-5}$, $10^{-6}$, and $10^{-7}$ ng/μL using equally proportional dilution method, and RPA was performed for each dilution. The negative control template was substituted with DEPC-treated water. The diluted DNA templates were then used for both SHERLOCK fluorescence detection and lateral flow immunochromatographic strip detection. Each concentration was tested in triplicate to observe and record the fluorescence curve changes and detection lines.

## RESULTS

### Structural genomics analysis

We randomly selected 16 strains from a total of 42 *Bp* strains for WGS. The results revealed that the genomes of these 16 strains each consisted of two chromosomes (*BP*-11 contains additional plasmids), with a total genome size ranging from 7.1 Mb to 7.3 Mb. Chromosome 1 was larger, while chromosome 2 was smaller. Gene prediction using Prodigal v.2.6.3 software estimated that each genome contains between 5,759 and 6,011 protein-coding genes. Additionally, all strains had 12 rRNA genes, including 4 each of 5S rRNA, 16S rRNA, and 23S rRNA. The number of tRNA genes ranged from a minimum of 62 to a maximum of 66. The length of genomic repeats in the 16 strains varied from 275,435 base pairs (bp) to 285,614 bp. All strains contained four CRISPR sequences on chromosome 1. The number of prophages varied between chromosomes 1 and 2, with a higher number of prophages found on chromosome 2 compared to chromosome 1.

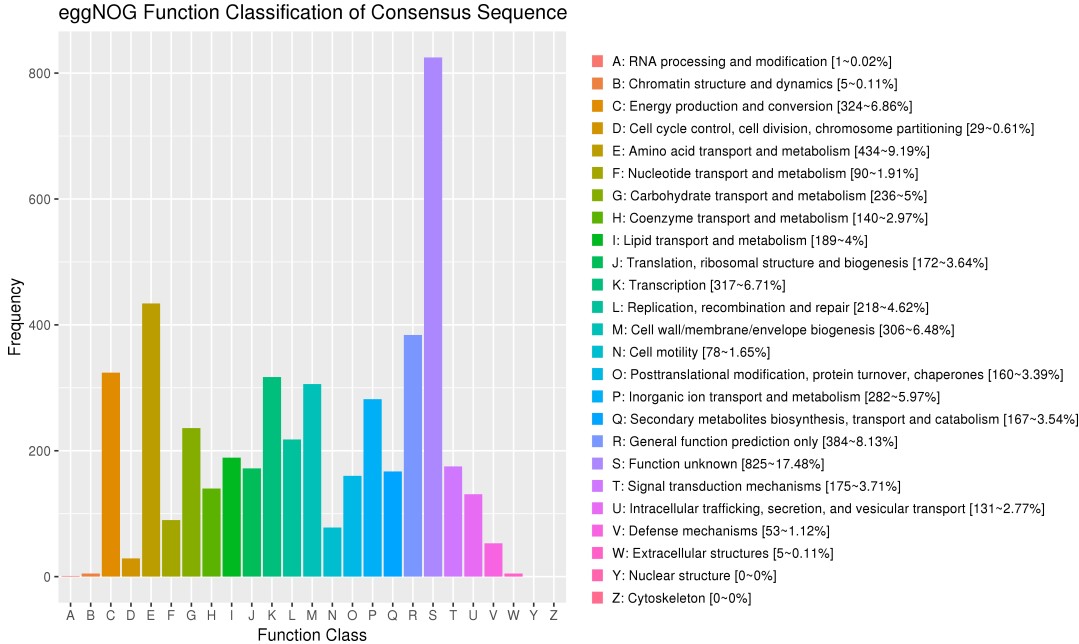

**FIG 1** Functional classification statistics of EggNOG functional genes on two chromosomes of *Bp*-11 genome. The EggNOG function annotation information of protein-coding genes (CDSs) was divided into 23 groups and distinguished by different colors. The horizontal coordinate represents each EggNOG classification category, and the vertical coordinate represents the number of corresponding functional genes. Different colors represent different EggNOG annotation functions, including the number and proportion of genes.

Notably, the genome of *Bp*-11 included three prophages: two located on its plasmid and one on chromosome 2.

## Functional genomics analysis

### EggNOG functional annotation

In the whole genome of strain *Bp*-11, besides the two chromosomes, a plasmid was also identified. The EggNOG functional annotation results for *Bp*-11 are shown in Fig. 1. The functional annotations of coding DNA sequences (CDSs) were categorized into 23 groups, each represented by a different color. Group S, whose functions were unknown, did not have any annotated representatives. The remaining 22 groups represented various functional annotations. Of the 4,614 annotated genes on the two chromosomes of *Bp*-11, 825 genes were unannotated and their functions were unknown, accounting for 17.48% of all CDSs. Figure 1 illustrates that the top 3 functional categories were groups E (amino acid transport and metabolism), R (general function prediction), and C (energy generation and conversion). Notably, no genes in the genome were annotated for group Y (nuclear structure) or group Z (cytoskeleton).

### Functional classification and genomic localization

The genome circle for *Bp*-11 is depicted as follows: Fig. 2I shows chromosome 1, which is 3,929,742 bp in size; Fig. 2II shows chromosome 2, which was 3,105,737 bp; and Fig. 2III shows the plasmid, a closed circular double-stranded DNA structure with a size of 217,237 bp. Compared to the chromosomes, the plasmid had fewer repeat sequences and lower GC content and lacked tRNA and rRNA genes.

### VFDB annotation

The VFDB was used for identifying virulence factors. To analyze the correlation between virulence genes and clinical outcomes, the 16 *Bp* strains were categorized into two groups: the death group (five strains) and the improved group (11 strains) based on patients' clinical outcomes (Table 2). A total of 17 virulence factors were identified using VFDB annotations. Virulence genes such as *katA*, *algU*, *clbD*, *clbF*, *Cj1435c*, *ybtS*, *recN*, *fbpC iron(III)*, *cdpA*, *cheA*, and *rcsB* were present in both the dead group and improved group. The *mrkD* gene, which encodes a type 3 pilin protein, was only found in the death group. The genes *porB*, *allB*, allC, *pvdQ*, and *bprA* were unique to strains *Bp*-7 and *Bp*-16 in the improved group. Detailed information is provided in Tables 3 and 4.

### CARD database annotation

The CARD was used for annotating antibiotic resistance genes. Prior to the CARD database annotation, antimicrobial susceptibility testing (AST) showed that all 16 *Bp* strains were sensitive to SXT, CAZ, and IPM (Table 5).

CARD database annotation identified a total of seven antibiotic resistance genes: *adeF*, *amrA*, *amrB*, *OXA-57*, *OXA-59*, *R39*, and *Omp38*. The annotation results for these genes are detailed in Table 6. The resistance mechanisms associated with these genes include *adeF*, *amrA*, and *amrB*, which are involved in antibiotic efflux pumps. *OXA-57*, *OXA-59*, and *R39* contribute to antibiotic inactivation, and *Omp38* reduces antibiotic permeability.

## Comparative genomic analysis

### Gene family clustering and differential gene analysis

Gene family clustering was performed for 16 *Bp* strains and the reference strain K96243, and the results were visualized using a petal diagram (Fig. 3). The analysis revealed that the 16 strains shared 5,192 gene families. Specific gene families were identified as

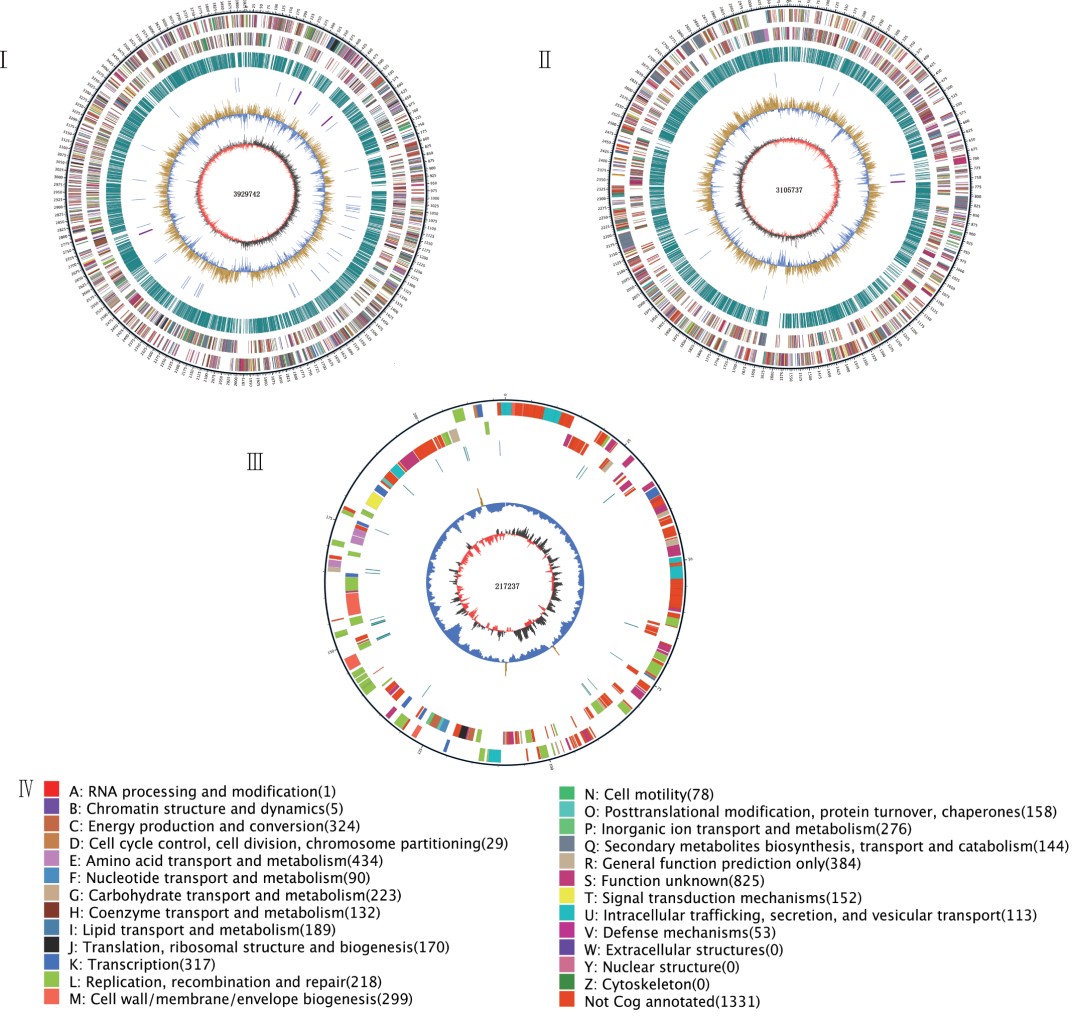

FIG 2 Functional annotation results of *Bp*-11 genome. (I) Chromosome 1 genosphere map of *Bp*-11. (II) Chromosome 2 genosphere map of *Bp*-11. (III) Plasmid genosphere map of *Bp*-11. (IV) Different colors correspond to I–III, representing different annotation functions and the number of genes on the annotation for that function. In I–III, from the outside to the inside: the outermost black circle, marked with a scale indicating genome size (1 scale represents 5 kb); the second circle shows the genes on the positive chain of the genome; the third circle shows the genes on the negative chain of the genome, with the second and third circles colored differently to distinguish EggNOG functions. The blue-green section of the fourth circle represents repeating sequences; in the fifth circle, blue indicates tRNA and purple indicates rRNA. The sixth circle shows GC content, with light yellow regions indicating higher GC content than the average genome GC content, and the higher the peak, the greater the difference. Blue regions indicate GC content lower than the average genome GC content. The seventh circle represents GC-skew, where the dark gray area indicates regions where the content of G is greater than C, and the red area indicates regions where C is greater than G. The innermost circle identifies the size (bp) of the genosphere map.

follows: *Bp*-8 had one specific gene family, *Bp*-11 had two, and K96243 had eight. No specific gene families were found in the remaining strains.

## Phylogenetic analysis

Gene clustering and phylogenetic analysis were conducted for the 16 *Bp* strains and the reference strain K96243, resulting in the construction of a phylogenetic tree (Fig. 4). Branches with bootstrap values greater than 75 were considered reliable. In this phylogenetic tree, the branches for strains *Bp*-2, *Bp*-7, *Bp*-13, and *Bp*-16 had a support value of 57, indicating that these branches were unreliable. In contrast, other branches had support values ranging from 89 to 100, demonstrating their reliability. Although the 16 *Bp* strains from Hainan were distributed across various branches of the phylogenetic tree, they ultimately converged at a single root node, indicating that they belong to the

**TABLE 2** Background data of 16 *Bp* strains[a,b]

| Sample number | Gender | Age | Risk factor | Sampling time | Sample source | Sample type | Patients' clinical outcomes |
|---|---|---|---|---|---|---|---|
| *Bp*-1 | F | 64 | Diabetes | 2021.12 | Wanning | Blood | Death |
| *Bp*-2 | M | 53 | Diabetes | 2022.02 | Chengmai | Blood | Improve |
| *Bp*-3 | F | 49 | – | 2020.10 | Chengmai | Blood | Death |
| *Bp*-4 | M | 30 | Systemic lupus erythematosus | 2021.08 | Changjiang | Blood | Death |
| *Bp*-5 | M | 65 | Diabetes | 2022.03 | Ledong | Pus | Improve |
| *Bp*-6 | M | 63 | Diabetes | 2020.11 | Ledong | Blood | Improve |
| *Bp*-7 | M | 54 | Diabetes | 2021.11 | Danzhou | Blood | Improve |
| *Bp*-8 | F | 24 | – | 2021.05 | Haikou | Pus | Improve |
| *Bp*-9 | M | 49 | – | 2021.06 | Haikou | Blood | Improve |
| *Bp*-10 | M | 30 | Diabetes | 2021.06 | Haikou | Pus | Improve |
| *Bp*-11 | M | 54 | – | 2021.06 | Dongfang | Liquor pericardii | Improve |
| *Bp*-12 | M | 42 | – | 2021.09 | Haikou | Drainage fluid | Improve |
| *Bp*-13 | M | 39 | Diabetes | 2021.10 | Haikou | Blood | Death |
| *Bp*-14 | M | 52 | Tuberculosis and diabetes | 2021.10 | Chengmai | Blood | Improve |
| *Bp*-15 | M | 59 | Diabetes | 2022.01 | Wanning | Blood | Death |
| *Bp*-16 | M | 36 | Diabetes | 2022.03 | Haikou | Blood | Improve |

[a]F, female; M, male.
[b]–, indicates absence of risk factor.

same group. Strains *Bp*-12 and *Bp*-14 were found to be the most closely related, followed by *Bp*-10 and *Bp*-9, with *Bp*-11 being most closely related to K96243.

### Genome collinearity analysis

Genome collinearity was analyzed by connecting homologous genes between the genomes of the 16 sequenced strains and the reference strain K96243. Different colors represented different homologous gene pairs in the collinear map (Fig. 5). The analysis revealed multiple homologous genes shared between the 16 *Bp* strains and the reference strain K96243.

## Establishment and optimization of specific SHERLOCK detection system

### Determine the target gene of Bp

Based on sequencing results and literature reports, *TTS1*, a component of the *Bp* secretion system critical for its pathogenicity, was present in all *Bp* strains and was highly conserved. In addition, *ORF2*, located within the coding region of the TTS1 gene cluster,

**TABLE 3** Functional classification of virulence factors in death group[a]

| Annotated gene | Carrier strain | Virulence factor | Function of genes | Strain number |
|---|---|---|---|---|
| *katA* | *Bp*-4, *Bp*-13 | KatA | Catalase | 2 |
| *algU* | *Bp*-4, *Bp*-13 | Alginate | Alginate | 2 |
| *clbD* | *Bp*-4, *Bp*-13 | Colibactin | Colicin | 2 |
| *clbF* | *Bp*-4, *Bp*-13 | Colibactin | Colicin | 2 |
| *Cj1435c* | *Bp*-1, *Bp*-3 | Capsule | Capsular phosphatase | 2 |
| *ybtS* | *Bp*-1, *Bp*-3 | Ybt | Yersinomycin | 2 |
| *recN* | *Bp*-1, *Bp*-3 | RecN | DNA repair protein | 2 |
| *fbpCiron (III)* | *Bp*-1, *Bp*-3 | FbpABC | Ferroportin | 2 |
| *cdpA* | *Bp*-15 | CdpA | Phosphodiesterase | 1 |
| *cheA* | *Bp*-15 | Pse5Ac7Ac | Chemotactic histidine kinase | 1 |
| *rcsB* | *Bp*-15 | RcsAB | Transcriptional regulatory factor | 1 |
| *mrkD* | *Bp*-15 | Type 3 fimbriae | Fimbriin type 3 | 1 |

[a]The death group contained five strains, which were strains *Bp*-1, *Bp*-3, *Bp*-4, *Bp*-13, and *Bp*-15.

**TABLE 4** Functional classification of toxicity factors in the improved group[a]

| Annotated gene | Carrier strain | Virulence factor | Function of genes | Strain number |
|---|---|---|---|---|
| katA | Bp-5, Bp-6, Bp-8, Bp-9, Bp-11, Bp-12 | KatA | Catalase | 6 |
| algU | Bp-5, Bp-6, Bp-8, Bp-9, Bp-11, Bp-12 | Alginate | Alginates | 6 |
| clbD | Bp-5, Bp-6, Bp-8, Bp-9, Bp-11, Bp-12 | Colibactin | Colicin | 6 |
| clbF | Bp-5, Bp-6, Bp-8, Bp-9, Bp-11, Bp-12 | Colibactin | Colicin | 6 |
| Cj1435c | Bp-2, Bp-10, Bp-14 | Capsule | Capsular phosphatase | 3 |
| ybtS | Bp-2, Bp-10, Bp-14 | Ybt | Yersinomycin | 3 |
| recN | Bp-2, Bp-10, Bp-14 | RecN | DNA repair protein | 3 |
| fbpCiron(III) | Bp-2, Bp-10, Bp-14 | FbpABC | Ferroportin | 3 |
| cdpA | Bp-7 | CdpA | Phosphodiesterase | 1 |
| cheA | Bp-7 | Pse5Ac7Ac | Chemotactic histidine kinase | 1 |
| rcsB | B-7 | RcsAB | Transcriptional regulatory factor | 1 |
| porB | Bp-16 | Porin | Momp | 1 |
| allB | Bp-16 | Allantoin utilization | Allantoinase | 1 |
| allC | Bp-16 | Allantoin utilization | Allantoic acid amide hydrolase | 1 |
| pvdQ | Bp-16 | Pyoverdine | Serine lactone acylase | 1 |
| bprA | Bp-7 | Bsa T3SS[b] | Hns-like regulatory protein | 1 |

[a] There were 11 strains in the improved group, including strains Bp-2, Bp-5, Bp-6, Bp-7, Bp-8, Bp-9, Bp-10, Bp-11, Bp-12, Bp-14, and Bp-16.
[b]T3SS, type 3 secretion system.

was specific to *Bp* and absent in closely related *Burkholderia* species such as *Burkholderia thailandensis* and *Burkholderia cepacia*. Consequently, the TTS1-ORF2 genome sequence (GenBank: AF074878) was chosen as the target fragment for RPA. After synthesizing crRNA complementary to the target sequence, the SHERLOCK detection system was established.

### RPA primer screening

Based on the structural characteristics of the *TTS1-ORF2* genome sequence, nine pairs of primers were designed and used for RPA amplification. Electrophoretic analysis of the amplification products revealed a target band of approximately 200 bp, with bands at 400 bp–500 bp attributed to non-specific amplification due to the T7 promoter (Fig. 6A). Among the primers, those pairs, F3+R1, F3+R2, and F3+R3, produced clear bands and demonstrated good amplification efficiency. ImageJ software was employed to assess the brightness of the bands, revealing that the F3+R1, F3+R2, and F3+R3 primer pairs exhibited the strongest brightness (Fig. 6B). Combining these observations, it was determined that the F3+R1, F3+R2, and F3+R3 primer pairs had the highest amplification efficiency for the TTS1-ORF2 gene in RPA. To evaluate their performance in the SHERLOCK fluorescence detection system, the F3+R1, F3+R2, and F3+R3 primer pairs

**TABLE 5** Antimicrobial susceptibility testing results of 16 *Bp* strains[a,b]

| Antibacterial agents | Interpretive standard MIC/ (µg·mL-1) | | | The data of this study | | |
|---|---|---|---|---|---|---|
| | S | I | R | Number | S% | R% |
| SXT | ≤2/38 | – | ≥4/72 | 16 | 100 | 0 |
| IPM | ≤4 | 8 | ≥16 | 16 | 100 | 0 |
| CAZ | ≤8 | 16 | ≥32 | 16 | 100 | 0 |

[a]MIC, minimum inhibitory concentration; S, susceptible; I, intermediate; R, resistance.
[b]"–" indicates no data.

**TABLE 6**  Results of drug resistance gene annotation

| Drug resistance gene | Corresponding antibiotic | Quantity (%) | Resistance mechanism |
| --- | --- | --- | --- |
| *adeF* | Tetracycline, fluoroquinolones | 14 (87.5) | Antibiotic efflux |
| *amrA* | Aminoglycosides | 10 (62.5) | Antibiotic efflux |
| *amrB* | Aminoglycosides | 9 (56.3) | Antibiotic efflux |
| *OXA-57* | Penicillin, cephalosporins | 2 (12.5) | Antibiotic inactivation |
| *OXA-59* | Penicillin, cephalosporins | 4 (25.0) | Antibiotic inactivation |
| *R39* | Penicillins | 9 (56.3) | Antibiotic inactivation |
| *Omp38* | Penicillins, cephalosporins, carbapenems | 9 (56.3) | Reduce antibiotic permeability |

were tested. Results showed that under identical amplification conditions, the F3+R3 primer pair produced the highest fluorescence signal increase (Fig. 6C), indicating the best amplification efficiency. Therefore, the F3+R3 primer pair was selected for subsequent studies.

## Sensitivity and specificity analysis of the SHERLOCK detection system

For the SHERLOCK fluorescence method, all 42 *Bp*-positive strains showed a clear upward trend in fluorescence curves compared to the control group. The minimum fluorescence value reached approximately 10,000, and most reactions can form obvious increasing signals within 20 minutes, indicating positive detection results (Fig. 7A and B). In contrast, no significant upward trend was observed in the curves for the 11 control strains and DEPC-treated water; these curves were similar to the negative control, resulting in negative test outcomes (Fig. 7C).

For the SHERLOCK strip test method, all 42 *Bp*-positive strains exhibited clear detection lines, confirming positive results (Fig. 7D). In the test strips corresponding to the 11 control strains, only the quality control bands were visible, and no T lines appeared, indicating negative results (Fig. 7E).

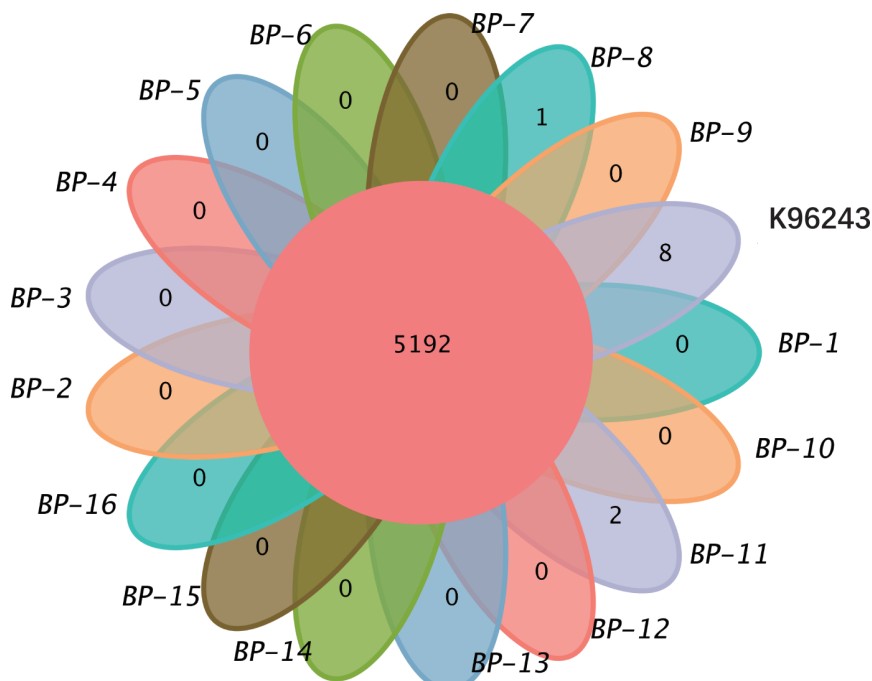

**FIG 3**  Gene family clustering of 16 *Bp* strains and K96243. Overlapping regions represent shared gene families between species, while non-overlapping regions indicate gene families specific to each species. Specific gene families (functions) were identified as follows: *Bp*-8 had one specific gene family (enoyl-acyl carrier protein reductase), *Bp*-11 had two (integrase core domain; transposase), and K96243 had eight (immunoglobulin I-set domain; resolvase, N terminal domain; H-NS histone family; helix-turn-helix domain; phage P2 GpE; LysR substrate binding domain; phage integrase family; adenylylsulfate kinase).

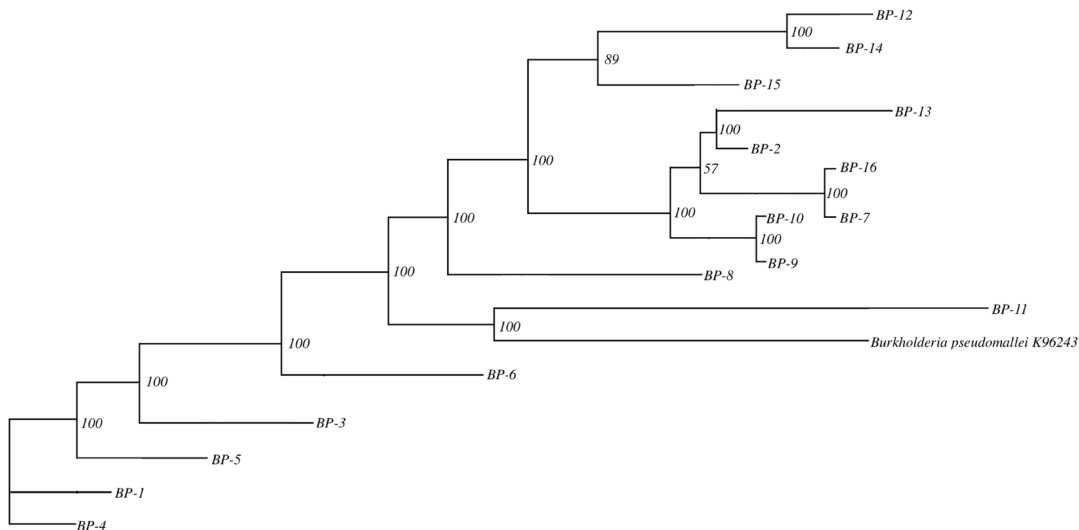

**FIG 4** Phylogenetic tree. The phylogenetic tree includes root, nodes, branches, and bootstrap values, which reflect the evolutionary relationships and support levels for the branches.

In summary, both the SHERLOCK fluorescence method and strip test method demonstrated consistent results, with a sensitivity of 100%. The detection of the 11 control strains was consistently negative, confirming that the SHERLOCK system has a high specificity (100%) and does not cross-react with other bacterial strains. In addition, the time for nucleic acid extraction (according to the manufacturer's instructions, it takes about 20 minutes) and RPA amplification (it takes about 20 minutes) is about 40 minutes. The final SHERLOCK fluorescence test (fluorescence values were read after 20 minutes) took a total of 60 minutes. The SHERLOCK test strip method requires incubation at 37°C for 30 minutes before testing; it takes about 70 minutes in total. Therefore, both methods can achieve rapid detection of *Bp* within 1–2 hours.

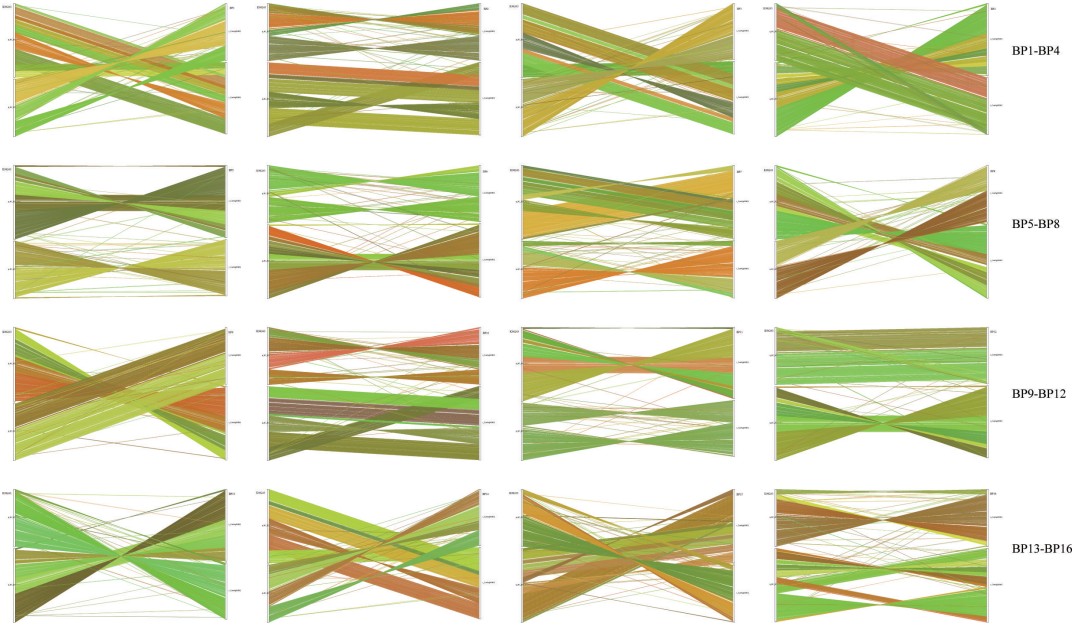

**FIG 5** Genome collinearity between 16 sequenced *Bp* strains and K96243. Different colors represent different homologous genes. The strains are linked linearly to the homologous genes of K96243, showing the collinearity between the genomes.

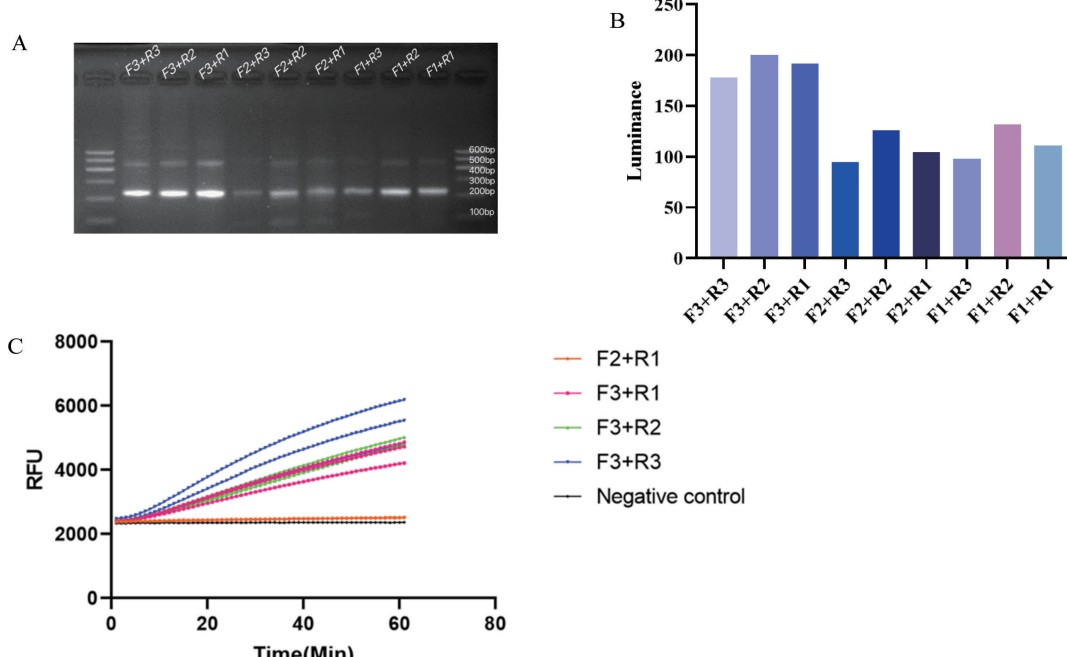

**FIG 6** Screening of RPA primers and construction of SHERLOCK fluorescence detection system. (A) Electrophoretic analysis of RPA products using different primer pairs. (B) Relative luminance of the RPA amplified target strips shown in A. (C) Comparison of fluorescence detection efficiency among three pairs of RPA primers using the SHERLOCK fluorescence detection method.

## Analysis of the LOD of the SHERLOCK fluorescence method

To determine the LOD of SHERLOCK fluorescence method and lateral flow immunochromatographic strip method for *Bp* nucleic acids, *Bp* strains were serially diluted using a 10-fold gradient method, ranging from $10^{-1}$ ng/μL to $10^{-7}$ ng/μL. The LOD was assessed using these diluted samples, with DEPC-treated water serving as a negative control. Each concentration was tested in triplicate to observe and record the fluorescence curve changes and detection lines (requiring a detection rate of 100%).

The results indicated that as the concentration of *Bp* nucleic acids was diluted, the real-time fluorescence values progressively decreased, correlating with a reduction in detection sensitivity. The LOD was determined to be $10^{-4}$ ng/μL (100 fg), as depicted in Fig. 8A. Additionally, fluorescence values were detectable within 20 minutes of the reaction's initiation, demonstrating that the SHERLOCK fluorescence method enables rapid detection of target nucleic acids within a short timeframe (Fig. 8B). In addition, the SHERLOCK test strip method was used for testing, and the final limit of detection was $10^{-3}$ ng/μL (Table 7).

## DISCUSSION

*Bp* is the causative agent of melioidosis, a zoonotic infectious disease with a mortality rate ranging from 20% to 50% in acute cases. This rate is even higher in resource-limited regions with inadequate diagnostic facilities and equipment, particularly among patients with severe comorbidities (22–24). Hainan Island's relatively isolated geographical location has contributed to a stable *Bp* strains genetic profile over the past 15 years, with only a few new sequence types emerging (25). Most of the genetic information within the cloned populations has remained consistent, ensuring the reliability of genomic data obtained from metagenomic analysis of strains from the island. There is no study on structural genomics, functional genomics, and comparative genomics analysis of Hainan *BP* strain using third-generation sequencing technology. In this study, we utilized the PacBio sequencing platform to obtain high-integrity and accurate

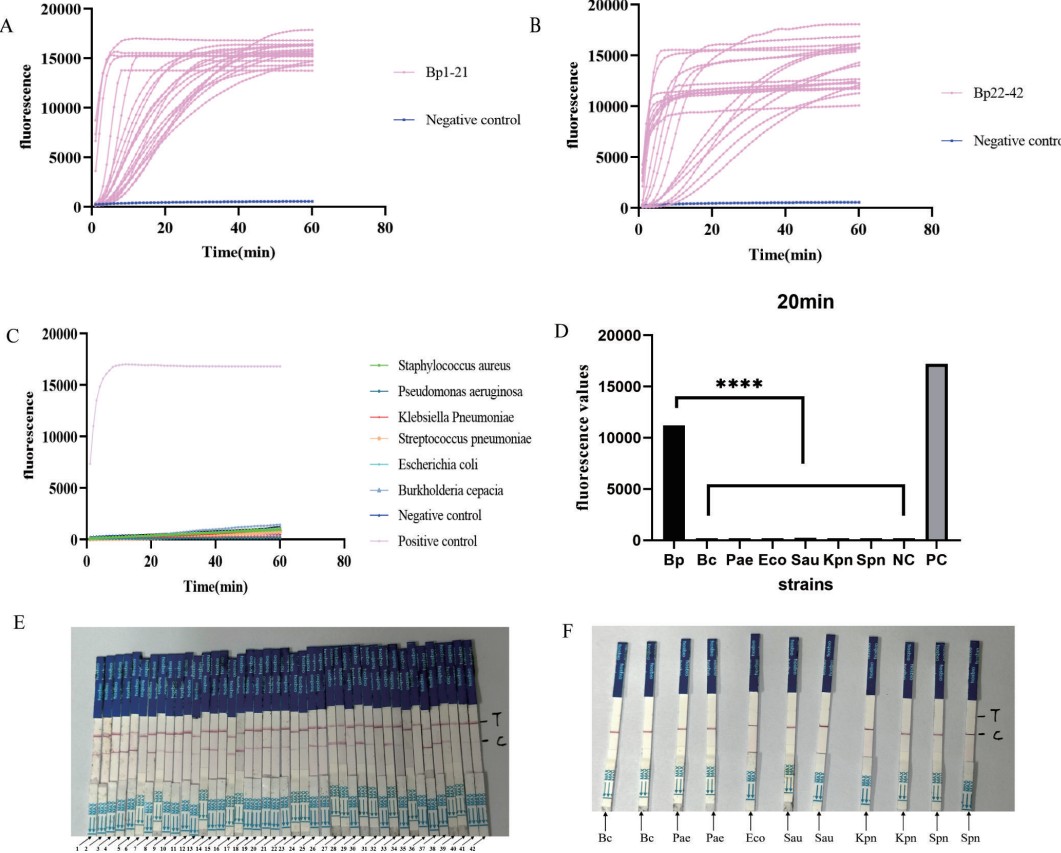

**FIG 7** Sensitivity and specificity analysis of SHERLOCK detection system. (A) SHERLOCK fluorescence detection results for strains *Bp*-1–21. (B) SHERLOCK fluorescence detection results for strains *Bp*-22–42. (C) SHERLOCK fluorescence detection results for the control group. (D) Fluorescence values detected by different strains after 20 minutes of reaction (unpaired *t*-test; ****, $P < 0.0001$). (E) SHERLOCK strip test results for strains *Bp*-1–42. (F) SHERLOCK strip test results for the control group.

whole genome sequences of endemic *Bp* strains from Hainan. In this study, we utilized the PacBio sequencing platform to obtain high-integrity and accurate whole genome sequences of endemic Bp strains from Hainan, completed research in structural genomics, functional genomics and comparative genomics. The genome sizes ranged from 7.1 Mb to 7.3 Mb, with GC content between 68.06% and 68.29%. The total number of CDSs in the assembled genomes ranged from 5,759 to 6,011. Each genome included

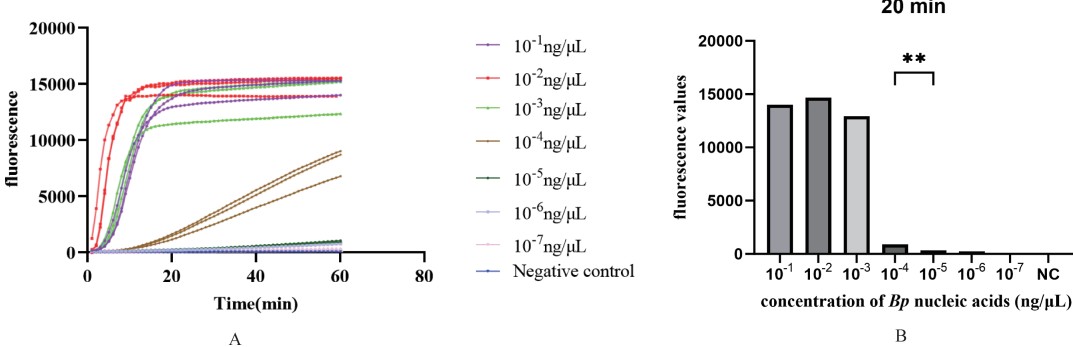

**FIG 8** Limit of detection of the SHERLOCK fluorescence detection system. (A) With the dilution of *Bp* nucleic acid concentration, the real-time fluorescence value gradually decreased, which was related to the decrease of detection sensitivity. (B) The lowest detection limit is $10^{-4}$ ng/µL (paired *t*-test; **, $P = 0.0041$); at this detection limit, a rapid increase in fluorescence values can be detected within 20 minutes of the start of the reaction.

**TABLE 7** Limit of detection of the SHERLOCK strip test method

| Concentration (ng/uL) | $10^{-1}$ | $10^{-2}$ | $10^{-3}$ | $10^{-4}$ | $10^{-5}$ | $10^{-6}$ | $10^{-7}$ | Negative control |
|---|---|---|---|---|---|---|---|---|
| Detection times | 3 | 3 | 3 | 3 | 3 | 3 | 3 | 3 |
| Number of detected | 3 | 3 | 3 | 0 | 0 | 0 | 0 | 0 |
| Detection rate (%) | 100 | 100 | 100 | 0 | 0 | 0 | 0 | 0 |

four 5S rRNAs, four 16S rRNAs, and four 23S rRNAs. The number of tRNAs ranged from 62 to 66, and the length of genomic repeat sequences ranged from 275,435 bp to 285,614 bp. Furthermore, all 16 strains contained four CRISPR sequences associated with bacterial immunity on chromosome 1. We also identified multiple prophages in the *Bp* strains' genomes. Previous research suggests that prophage structures may correlate with virulence and antibiotic resistance characteristics. Variations in the location and number of prophages among different strains suggest potential differences in virulence and antibiotic resistance profiles.

Compared with other databases, the VFDB is more focused on collecting and updating virulence factor information of bacterial pathogens and is widely used in bioinformatics research, especially in the prediction and analysis of virulence factors. This study identified a total of 17 virulence factors in the analyzed strains by VFDB. Among these, the genes *katA*, *algU*, *clbD*, *clbF*, *Cj1435c*, *ybtS*, *recN*, *fbpCiron(III)*, *cdpA*, *cheA*, and *rcsB* were present in both the dead group and improved group. The *mrkD* gene, which encodes type 3 fimbrial protein, was exclusively found in the dead group. This suggests that the presence of the *mrkD* gene might contribute to the higher virulence observed in the dead group, considering it is associated with the production of type 3 fimbrial protein, which is critical for bacterial adherence and pathogenicity. Previous studies have identified other virulence factors for *Bp*, including the type 3 secretion system gene cluster, the type 6 secretion system gene cluster, the *bimA* gene of the autotransporter complex, the *fliC* gene encoding flagellin, and the *wcb* gene cluster associated with capsular polysaccharides (26). The virulence factors identified in this study differ from those reported in previous research, indicating that the 16 *Bp* strains from Hainan may have unique virulence profiles. This suggests regional variations in the virulence characteristics of *Bp* strains, with potentially different distributions and impacts compared to strains from other regions. The virulence genes identified in this study not only provide insights into the pathogenic mechanisms of *Bp* but also offer potential targets for molecular diagnostics and the development of melioidosis vaccines.

*Bp* exhibits natural resistance to many commonly used antibiotics, limiting the treatment options available for clinical management. The mechanisms of resistance include enzymatic inactivation, altered target sites, and efflux from the bacterial cell. In addition, changes in certain specific genes are also responsible for natural resistance, such as mutations or overexpression of the *PenA* gene that can lead to CAZ resistance. The specific mutations within the *amrR* gene contribute to the upregulation of amrAB-oprA efflux pump transcriptional levels, which are linked to meropenem resistance (27). The CARD database can not only search for identified resistance genes in target strains, but also predict potential resistance genes with its RGI tool. Although it requires users to have certain professional knowledge and operational skills, it is still one of the most popular tools for drug resistance gene research. In this study, 16 *Bp* strains from Hainan were analyzed by the CARD database, and seven antibiotic resistance genes were identified: *adeF*, *amrA*, *amrB*, *OXA-57*, *OXA-59*, *R39*, and *Omp38*. Among the identified resistance genes, *adeF*, *amrA*, and *amrB* were associated with the production of antibiotic efflux pumps. Specifically, *AdeF* is involved in mediating resistance to tetracyclines and fluoroquinolones by actively pumping these antibiotics out of the bacterial cell. Specifically, *adeF* is involved in mediating resistance to tetracyclines and fluoroquinolones by actively pumping these antibiotics out of the bacterial cell. *AmrA* and *amrB* are two important components of the multiple antibiotic efflux pump AmrAB-OprA of Bp, which are related to resistance to aminoglycoside antibiotics (28). In contrast, *OXA-57* and *OXA-59* are common genes in the genome of *Bp*, mainly expressing class D β-lactamases.

However, most studies have shown that the resistance of *Bp* to cephalosporins is mainly related to class A β-lactamases, which is mediated by the resistance gene *PenA* (29). The *omp38* is a unique gene in the genome of *Bp*, mainly encoding a pore protein located on the cell membrane of *Bp*, but studies have shown that it has nothing to do with antibiotic resistance (30). In addition, there is no relevant research report on the drug resistance gene *R39* of *Bp*. According to the functional annotation results, it is speculated that it may produce β-lactamases, which are related to penicillin antibiotic resistance, and further analysis is needed. The AST of the 16 *Bp* strains demonstrated sensitivity to SXT, CAZ, and IPM. The absence of the BpeEF-OprC efflux pump structure in the genome of these strains aligns with their 100% sensitivity to SXT.

Gene family analysis of the 16 *Bp* strains and the reference strain *Bp* K96243 revealed a core genome consisting of 5,192 genes. Phylogenetic analysis indicated that all 16 strains are closely related, sharing a common ancestor and belonging to the same group. Interestingly, *Bp*-1, -3, -4, and -5 were closely related, but *Bp*-1, -3, and -4 appeared in the death group, while *Bp*-5 only appeared in the improvement group. This may indicate that although the genetic relationship is close, the level of virulence may be different, which enhances the complexity of clinical manifestations after *Bp* infection. Genome collinearity analysis demonstrated a substantial number of homologous genes between the 16 sequenced *Bp* strains and the reference strain K96243. The collinearity maps showed variability in gene location and color, reflecting the genetic diversity among the strains. These findings suggest that the *Bp* strains from Hainan exhibit notable genetic diversity. This diversity may be attributed to selection pressures that have driven adaptive evolution in some strains, leading to the observed variations in their genomes.

The SHERLOCK technique has been used to detect many pathogens since it was invented. In recent years, there have been related reports on the diagnosis of pseudomallei. However, the specific target gene sequences of *Bp* detected based on the CRISPR/Cas system are different. For example, Zhang et al. identified 44 specific sequence tags from the core genome sequence of chromosome 1 and chromosome 2 of *Bp* by bioinformatics, two of which were selected for the development of dual-target RPA-CRISPR/Cas12a detection method, which finally realized the detection of *Bp* and showed high specificity (20). In another study in Thailand, researchers searched for a conserved CRISPR-Cas12a target through computer simulation and named the obtained highly specific target sequence of *Bp* crBP34. Based on this sequence, a CRISPR-Cas12a diagnostic method was developed that was able to detect all clinical strains of endemic *B. pseudomallei* while distinguishing humans from other pathogens, including its closely related species *B. thailandensis* (21). Different from the above studies, in this study, we developed a SHERLOCK system for the rapid detection of *Bp* strains by targeting TTS1-ORF2, a specific and conserved sequence in *Bp*. This system integrates the CRISPR/Cas13a technology with RPA, offering an effective method for the swift detection of melioidosis in this region. Methodological evaluation demonstrated that the SHERLOCK system can detect Hainan endemic strains of melioidosis with a minimum detection limit of 100 fg. Importantly, the system showed no cross-reactivity with common pathogens that might be confused with *Bp* strains in clinical settings. Results can be interpreted through fluorescence signal changes or lateral flow immunochromatographic strips. Compared to traditional isolation and culture methods, this system reduced the time required from nucleic acid extraction to result reading to just 1–2 hours, making it suitable for early and rapid diagnosis of melioidosis. Unlike serological tests, this method was not dependent on antibody titers and provides stable sensitivity and specificity (27). Additionally, the SHERLOCK test strip method did not require specialized equipment or complex procedures, making it easy to use and transport. The test strips were user-friendly and cost-effective, ideal for deployment in primary medical institutions, especially in melioidosis-endemic areas. Furthermore, the CRISPR/Cas13a system's ability to detect single-base mismatches could help minimize non-specific reactions and reduce the incidence of false-positive results.

## CONCLUSION

In summary, this study adopted TGS technology to acquire accurate whole genome sequences of *Bp*, coupled with bioinformatics analysis to enhance our understanding of the molecular structure, function, and pathogenic mechanisms of *Bp*. It provided valuable insights into the drug resistance characteristics and genetic evolution of the pathogen. Additionally, we pioneered the use of the third generation of sequencing results combined with the CRISPR/Cas system for detecting *Bp*, establishing a novel detection method specifically tailored for melioidosis in Hainan. This new approach represents a significant advancement in rapid diagnostic techniques. Future studies will focus on further optimizing the design of crRNAs and RPA primers to enhance the sensitivity and efficiency of the detection method.

## ACKNOWLEDGMENTS

We would like to thank our laboratory colleagues for their assistance in the data and sample collection, and laboratory analysis. We would also like to thank the research teams of the Second Affiliated Hospital of Hainan Medical University and Sanya Central Hospital for their support for this study, because all samples were obtained from these two hospitals.

Key Research and Development Project of Hainan Province, ZDYF2022SHFZ023; Key Research and Development Project of Hainan Province, ZDYF2021SHFZ085; Health Science and Technology Innovation Joint Project of Hainan Province, WSJK2024QN103.

J.H.: Conceptualization, Formal analysis, Methodology, Visualization, Writing – original draft. S.X.: Writing – review & editing. Z.Z.: Writing – review & editing. W.G.: Writing – review & editing. W.X.: Writing – review & editing. Z.C.: Writing – review & editing. S.F.: Resources, Writing – review & editing. L.L.: Resources, Writing – review & editing. B.X.: Supervision, Validation, Writing – review & editing. X.C.: Supervision, Validation, Writing – review & editing.

The authors declare that the research was conducted in the absence of any commercial or financial relationships that could be construed as a potential conflict of interest.

## AUTHOR AFFILIATIONS

[1]Department of clinical Laboratory, Affiliated Cancer Hospital of Hainan Medical University, Hainan Cancer Hospital, Haikou, Hainan, People's Republic of China
[2]The First Clinical School of Hainan Medical University, Haihou, Hainan, People's Republic of China
[3]Sanya Hospital of Traditional Chinese Medicine, Sanya, Hainan, People's Republic of China
[4]Hainan Lvtou Medical Laboratory Center, Haikou, Hainan, People's Republic of China
[5]Department of Laboratory Medicine, The Affiliated Qingyuan Hospital (Qingyuan People's Hospital), Guangzhou Medical University, Qingyuan, Guangdong, China

## AUTHOR ORCIDs

Junjie Hu  http://orcid.org/0000-0002-3341-7588
Bin Xiao  http://orcid.org/0000-0002-8822-2542
Xinping Chen  http://orcid.org/0009-0003-1193-2890

## AUTHOR CONTRIBUTIONS

Junjie Hu, Conceptualization, Formal analysis, Methodology, Visualization, Writing – original draft | Shanshan Xu, Writing – review and editing | Zeng Zeng, Writing – review and editing | Wei Gong, Writing – review and editing | Weihua Xu, Writing – review and editing | Zhichao Ma, Writing – review and editing | Shengmiao Fu, Resources, Writing – review and editing | Linhai Li, Resources, Writing – review and editing | Bin

Xiao, Supervision, Validation, Writing – review and editing | Xinping Chen, Supervision, Validation, Writing – review and editing

## DATA AVAILABILITY

The datasets analyzed during the current study are available in the NCBI Sequence Read Archive (SRA) database with accession number: PRJNA1280284.

## ETHICS APPROVAL

The studies were conducted in accordance with the local legislation and institutional requirements. This study uses strains obtained from the remaining samples after clinical testing. Ethics Committee of Hainan Cancer Hospital did not require the study to be reviewed or approved by an ethics committee because it used post-test samples.

## ADDITIONAL FILES

The following material is available online.

### Open Peer Review

**PEER REVIEW HISTORY (review-history.pdf).** An accounting of the reviewer comments and feedback.

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
