## [Reviewer comments · Microbiology Spectrum]

Microbiology Spectrum

Whole Genome Study and Construction of SHERLOCK Detection Method for Endemic Strains of *Burkholderia pseudomallei* in Hainan based on Third-generation Sequencing

Junjie Hu, Shanshan Xu, Zeng Zeng, Wei Gong, Weihua Xu, Zhichao Ma, Shengmiao Fu, Lin-hai Li, Bin Xiao, and Xinping Chen

Corresponding Author(s): Xinping Chen, Hainan Medical University

Review Timeline:

Submission Date:	February 28, 2025
Editorial Decision:	May 11, 2025
Revision Received:	June 24, 2025
Editorial Decision:	July 8, 2025
Revision Received:	July 16, 2025
Accepted:	July 31, 2025

Editor: Vittal Ponraj

Reviewer(s): Disclosure of reviewer identity is with reference to reviewer comments included in decision letter(s). The following individuals involved in review of your submission have agreed to reveal their identity: Narisara Chantratita (Reviewer #2)

Transaction Report:

DOI: <https://doi.org/10.1128/spectrum.00592-25>

Re: Spectrum00592-25 (**Whole Genome Study and Construction of SHERLOCK Detection Method for Endemic Strains of Melioidosis in Hainan based on Third-generation Sequencing**)

Dear Dr. Xinping Chen:

Thank you for the privilege of reviewing your work. Below you will find my comments, instructions from the Spectrum editorial office, and the reviewer comments. Please consider addressing all the reviewer's comments and queries in order to be considered for further review, and or publication.

Revision Guidelines

Sincerely,

Vittal Prakash Ponraj Ph.D.,
Editor
Microbiology Spectrum

Reviewer #1 (Comments for the Author):

The authors did a great job and massive analysis were performed in this study. The authors performed third-genome sequencing to acquire full genomes of local Hainan Burkholderia pseudomallei isolates. The authors also proposed and evaluated a rapid test to identify B.pseudomallei isolates by nucleic acid detection technology on a fluorescence-based and colorimetric-based

platform. Overall, the manuscript needs to be restructured and rearranged in a smooth flow for easier comprehension especially on the methods and results section. Line numbering is required.

These are my suggestions:

Abstract:

1. Results: Please indicate the value of "good sensitivity and specificity".

Methods:

1. There are different number of isolates used in various applications. Please state the numbers clearly in the methods section.
2. Please rephrase the sentence "All samples were obtained from the remaining samples after clinical testing".
3. The authors should include the manufacturer of the equipment/reagents used in this study.
4. Is it broth microdilution method used to detect MIC? Which CLSI version is used to interpret antibiotic susceptibility for BP? Please include citation.
5. What does "The single-copy gene sequences" referring to? Are the authors comparing whole genome/core/accessory genes?
6. Please give full name of "RPA system" on first use.
7. The authors need to specify the conditions of the amplification to ensure the results are reproducible.
8. The authors may consider to create a table for easier identification of all the oligos used for each application. Please cite in the text for the table/figure.
9. Please clarify the interpretation of positive result. Is the result valid" if only the T line showed red bands and the C line was colorless"?
10. What does the term "clinically positive" referring to?
11. Please define the cases. For examples, true positive, false negative, etc.
12. Is the term "double dilution" correct?

Results:

1. In section 3.2, The authors analyse only BP-11, how about the other 15 isolates? Please explain. These large data may be included as supplementary material.
2. Please clearly label in Figure 2 the location of I, II and III.
3. Did the authors check on the plasmid sequence? It will be interesting to know if there's similarity with other BP or organism, possible acquired?
4. It will be easier to compare if the information in Table 3 and 4 to be incorporated in Table 2.
5. The authors should include the footnote and define the abbreviations used in the tables/figures.
6. The authors may remove the word "clinical" and change to antibiotic susceptibility testing. AST is an in vitro method therefore it will be confusing to use "clinical" term.
7. The authors are suggested to check the presence of these genes from the whole genome analysis. Majority of the genes listed are chromosomally encoded in BP genomes. It will be good to discuss on the absence of the genes in typical isolates.
8. In Table 6, the term "carrier rate" is confusing. The authors may use "number of isolates" instead.
9. What are the specific gene families shown from the analysis?
10. From the phylogenetic tree, it looks like the BP1,3 and 5 (death outcome) are diverged from the others. It will be good to include this in discussion.
11. Please recheck on the sentence "Conversely, ORF2, located ...
12. Did the authors perform the LOD limit on strip tests?

Discussion:

The authors may improve on the discussion.

Reviewer #2 (Comments for the Author):

Reviewer Comments:

This manuscript presents a whole genome analysis of 16 *Burkholderia pseudomallei* isolates from Hainan, China, coupled with the development of a CRISPR-based SHERLOCK diagnostic assay. The study addresses an important topic with potential public health implications. The manuscript generally provides valuable genomic insights and a new diagnostic tool. However, there are several points that require clarification or revision before publication:

1. Comparative Genomics and Epidemiology

While the study asserts that *B. pseudomallei* strains in Hainan have unique epidemiological characteristics, it would be more compelling to provide concrete evidence through comparative genome structural analysis. I recommend including a comparative analysis of genome structures between Hainan isolates and publicly available global isolates to substantiate the claim of distinct epidemiological traits.

2. Petal Diagram Interpretation

In Figure 3, the petal diagram depicts core and unique gene families among the 16 isolates and the reference strain K96243. Please provide detailed information about the eight unique gene families of K96243 and the two unique gene families of BP-11. This information would enhance the understanding of gene content diversity and specificity.

3. Diagnostic Specificity and Related Species

The SHERLOCK detection system targeting the TTS1-ORF2 gene is promising. However, assay specificity could be further validated by including closely related Burkholderia species such as *B. thailandensis*, *B. mallei*, and others. This would strengthen confidence in its clinical applicability and help rule out cross-reactivity.

4. Calculation of Sensitivity in Methods (Section 2.7.1)

The sensitivity formula used in Section 2.7.1 is incorrect. The correct formula should be:

$$\text{Sensitivity} = \frac{\text{True Positives}}{\text{True Positives} + \text{False Negatives}} \times 100\%$$

Please revise this accordingly.

5. Clarity of Terms in Discussion

In the Discussion section (third paragraph), the phrases "dead and improved strains" and "dead group" are confusing and potentially misleading. Please clarify that these and similar words throughout the manuscript refer to patients' clinical outcomes, not the bacterial strains themselves.

6. Antibiotic Resistance Gene Analysis

The analysis of resistance genes is appreciated; however, it overlooks several key resistance determinants previously reported in *B. pseudomallei*. For example:

The PenA gene, known to confer resistance to ceftazidime and amoxicillin-clavulanate, should be discussed. See Hii SYF et al., *Antimicrob Agents Chemother*, 2021.

Recent evidence shows that *amrR* deletions contribute to carbapenem resistance. Refer to Nimnuan-ngam et al., 2025. Please investigate and discuss the presence or absence of these genes in your isolates.

7. Unreferenced Claims in Introduction and Discussion

Several claims throughout the Introduction and Discussion lack appropriate citations. For instance, the claim that the R39 gene "might produce beta-lactamases" should be supported by references or clarified as speculative. Please ensure that all assertions are backed by appropriate literature.

8. Table 2 contains patients' information. These require ethical approval.

Minor Suggestions:

- Please check for consistent gene naming (e.g., use italics for gene names such as *amrR*, *penA*, *katA* etc.).
- A visual summary or schematic of the SHERLOCK workflow could enhance readability for a broader audience.

Reviewer #3 (Comments for the Author):

(see attached file)

Reviewer Comments:

This manuscript presents a whole genome analysis of 16 *Burkholderia pseudomallei* isolates from Hainan, China, coupled with the development of a CRISPR-based SHERLOCK diagnostic assay. The study addresses an important topic with potential public health implications. The manuscript generally provides valuable genomic insights and a new diagnostic tool. However, there are several points that require clarification or revision before publication:

1. Comparative Genomics and Epidemiology

While the study asserts that *B. pseudomallei* strains in Hainan have unique epidemiological characteristics, it would be more compelling to provide concrete evidence through comparative genome structural analysis. I recommend including a comparative analysis of genome structures between Hainan isolates and publicly available global isolates to substantiate the claim of distinct epidemiological traits.

2. Petal Diagram Interpretation

In Figure 3, the petal diagram depicts core and unique gene families among the 16 isolates and the reference strain K96243. Please provide detailed information about the eight unique gene families of K96243 and the two unique gene families of BP-11. This information would enhance the understanding of gene content diversity and specificity.

3. Diagnostic Specificity and Related Species

The SHERLOCK detection system targeting the TTS1-ORF2 gene is promising. However, assay specificity could be further validated by including closely related *Burkholderia* species such as *B. thailandensis*, *B. mallei*, and others. This would strengthen confidence in its clinical applicability and help rule out cross-reactivity.

4. Calculation of Sensitivity in Methods (Section 2.7.1)

The sensitivity formula used in Section 2.7.1 is incorrect. The correct formula should be: $\text{Sensitivity} = \frac{\text{True Positives}}{\text{True Positives} + \text{False Negatives}} \times 100\%$
Please revise this accordingly.

5. Clarity of Terms in Discussion

In the Discussion section (third paragraph), the phrases “dead and improved strains” and “dead group” are confusing and potentially misleading. Please clarify that these and similar words throughout the manuscript refer to *patients’ clinical outcomes*, not the bacterial strains themselves.

6. Antibiotic Resistance Gene Analysis

The analysis of resistance genes is appreciated; however, it overlooks several key resistance determinants previously reported in *B. pseudomallei*. For example:

The *PenA* gene, known to confer resistance to ceftazidime and amoxicillin-clavulanate, should be discussed. See Hii SYF et al., *Antimicrob Agents Chemother*, 2021.

Recent evidence shows that *amrR* deletions contribute to carbapenem resistance. Refer to Nimnuan-ngam et al., 2025. Please investigate and discuss the presence or absence of these genes in your isolates.

7. Unreferenced Claims in Introduction and Discussion

Several claims throughout the Introduction and Discussion lack appropriate citations. For instance, the claim that the *R39* gene “might produce beta-lactamases” should be supported by references or clarified as speculative. Please ensure that all assertions are backed by appropriate literature.

8. Table 2 contains patients’ information. These require ethical approval.

Minor Suggestions:

- Please check for consistent gene naming (e.g., use italics for gene names such as *amrR*, *penA*, *katA* etc.).
- A visual summary or schematic of the SHERLOCK workflow could enhance readability for a broader audience.

Review of the manuscript: “Whole genome study and construction of SHERLOCK detection method for endemic strains of melioidosis in Hainan based on third-generation sequencing” by Junjie Hu et al.

This manuscript presents a genomic characterization of 16 *Burkholderia pseudomallei* (Bp) isolates from Hainan, China, using third-generation sequencing technology (PacBio), and describes the development of a CRISPR-based SHERLOCK diagnostic assay. The study identifies potential genomic signatures specific to local isolates and demonstrates the utility of CRISPR diagnostics for rapid and sensitive detection of Bp.

While the study addresses an important topic and employs modern technologies, the manuscript requires significant revisions to enhance clarity, rigor, and presentation.

General Comments

1. The title may be misleading, as it implies the SHERLOCK assay is specific to Hainan isolates. Consider clarifying whether the assay targets region-specific signatures or general Bp markers.
2. The manuscript should include line numbers to facilitate reviewer feedback.
3. There are numerous typographical and formatting inconsistencies (e.g., “Eggnog,” “EggNOG,” and “eggnog” appear interchangeably). Standardize terminology throughout.
4. Figure legends lack sufficient detail. Each should be self-contained—describing the figure's content, purpose, and analytical methods.
5. Several figures contain small or non-English labels and low-resolution images. High-quality, English-labeled vector graphics should be submitted.
6. Many factual statements are unsupported by citations. All claims, especially comparative or quantitative ones, must be appropriately referenced.

Abstract

- The statement that detection is achieved in “1–2 hours” should be substantiated in the Results section, with clear experimental data or protocol timing.

Introduction

1. The review of genomic studies is incomplete and omits important contributions from others such as Australian and Thai research groups, for example.
2. Similarly, the CRISPR diagnostics overview does not cite prior Bp-specific CRISPR assays developed by other groups, including those from China and Thailand. Acknowledging existing work will help situate the novelty of this study.

Methods

1. The Methods section lacks sufficient detail for replication. Information such as reagent sources, catalog numbers, and software versions should be included. If space is a concern, detailed protocols can be placed in Supplementary Materials.
2. Culture conditions for *B. pseudomallei* and other bacteria should be clearly described, including media and incubation parameters.
3. It is unclear whether clinical isolates or direct clinical specimens were used in the SHERLOCK assay. This distinction must be clarified in both the Methods and Results.
4. The limit of detection (LOD) experiment should include the number of replicates performed. A minimum of three independent replicates per data point is generally expected.

Results

1. Several analyses (e.g., CRISPR loci, functional annotations) are presented abruptly. Reorganize related results under thematic subheadings and provide better contextual transitions.
2. Justify the choice of isolate BP-11 for detailed analysis.
3. Figures labeled “Figure 2-I,” “-II,” and “-III” are not clearly marked. Label all panels explicitly.
4. Figure 3 appears simplistic and lacks scientific value. Reconsider whether it adds to the narrative or replace it with more informative content.
5. A whole-genome phylogenetic analysis including all available Bp strains would strengthen the manuscript by placing Hainan isolates in global context.
6. Genome collinearity should be quantitatively described and discussed.
7. The transition from genome analysis to SHERLOCK diagnostics lacks coherence. Currently, the CRISPR assay section seems disconnected. Consider integrating the genomic data to support assay design.
8. The panel of non-*Bp* organisms (n=6) used to assess assay specificity is limited. Consider expanding the panel to include more closely related species or increase numbers of isolates
9. Statistical analysis is generally lacking across experiments. Include appropriate statistical tests to support claims.
10. While VFDB and CARD databases are useful, discuss potential limitations and whether cross-validation with other tools was considered.
11. All sequencing data must be deposited in a public database prior to publication.

Discussion

1. The Discussion is relatively stronger than other sections, providing interpretation and context. However, some of this information (e.g., rationale and implications) should be incorporated earlier into the Results section to help guide readers.
2. A comparative discussion of the current SHERLOCK assay versus existing CRISPR diagnostics for Bp would be valuable for highlighting novelty and advantages.

Conclusion

- The phrase “we pioneered the use of the CRISPR/Cas” is not accurate given the existence of prior studies using CRISPR diagnostics for Bp. Consider rephrasing to reflect the study's specific contributions (e.g., adaptation to local strains or integration with third-generation sequencing data).

Dear Editor and Reviewers,

Thank you for offering us an opportunity to improve the quality of our submitted manuscript "Whole Genome Study and Construction of SHERLOCK Detection Method for Endemic Strains of Melioidosis in Hainan based on Third-generation Sequencing"(Spectrum00592-25). We appreciated very much the reviewers' constructive and insightful comments. In this revision, we have addressed all of these comments/suggestions. We hope the revised manuscript has now met the publication standard of your journal.

We have highlighted all the revisions in red

On the next pages, point-to-point responses to the queries raised by the reviewers are listed.

Reviewer #1

Abstract

Comment 1: Results: Please indicate the value of "good sensitivity and specificity".

Response: Thanks for your advice. Sensitivity and specificity values have been added at the end of the sentence. "both are 100%".

Methods

Comment 1: There are different number of isolates used in various applications. Please state the numbers clearly in the methods section.

Response: We explained the number of isolates used in different subsequent experiments and marked them red in the method section. First, 16 strains of *Burkholderia pseudomariae* were randomly selected from 42 strains of *Burkholderia pseudoeriae* for full genome sequencing. In the later experiment, 42 strains of *Burkholderia pseudomariae* and 11 strains of the control group were used for sensitivity and specificity analysis of the SHERLOCK detection system.

Comment 2: Please rephrase the sentence "All samples were obtained from the remaining samples after clinical testing".

Response: Rephrase the sentence as "A total of 53 clinical isolates were collected from human disease".

Comment 3: The authors should include the manufacturer of the equipment/reagents used in this study.

Response: Thanks for your advice. We supplemented the manufacturer of the equipment/reagent used with the full text.

Comment 4: Is it broth microdilution method used to detect MIC? Which CLSI version is used to interpret antibiotic susceptibility for BP? Please include citation.

Response: Susceptibility tests were performed according to the broth microdilution method recommended by CLSI (CLSI M100-Ed33 Performance Standards for Antimicrobial Susceptibility Testing). We also supplemented this in the method section.

Comment 5: What does "The single-copy gene sequences" referring to? Are the authors comparing whole genome/core/accessory genes?

Response: It's referring to "single-copy homologous gene", the single-copy homologous gene sequence of the sequenced strain was compared with the single-copy homologous gene sequence of the model strain K96243, and the evolutionary tree was constructed by the maximum likelihood method using PhyML software, and then the evolutionary relationship between the strains was analyzed. Thanks to Reviewer for his reminder, we have improved the expression.

Comment 6: Please give full name of "RPA system" on first use.

Response: Thanks for your advice. The full name of "RPA" is "Recombinase Polymerase Amplification", it has already been introduced in the "Introduction" section.

Comment 7: The authors need to specify the conditions of the amplification to ensure the results are reproducible.

Response: The amplification conditions are as follows: "RPA amplification system (50 μ L) included Primer Free Rehydration buffer (29.5 μ L), DNase-Free water (11.2 μ L), Forward primer (2.4 μ L, 10 μ M), Reverse primer (2.4 μ L, 10 μ M), template DNA (*Bp* DNA, 2 μ L), Magnesium Acetate (MgOAC, 2 μ L, 280 mM). After adding MgOAC, the components were thoroughly mixed and subjected to brief centrifugation. The reaction mixture was promptly incubated at 39°C for 20 minutes."

Comment 8: The authors may consider to create a table for easier identification of all

the oligos used for each application. Please cite in the text for the table/figure.

Response: Thanks for your advice. We have summarized all oligos in Table 1 and cited in the text.

Comment 9: Please clarify the interpretation of positive result. Is the result valid" if only the T line showed red bands and the Cline was colorless"?

Response: Result interpretations were as follows: (1) A positive result was indicated if both the test line (T line) and C line showed red bands, or if only the T line showed red bands and the C line was colorless; (2) A negative result was indicated if only the C line showed a red band and no red band was visible on the T line.

Comment 10: What does the term "clinically positive" referring to?

Response: "clinically positive" refers to a previously collected clinical isolate of *Bp*, which may not be clear enough and has been deleted.

Comment 11: Please define the cases. For examples, true positive, false negative, etc.

Response: Thanks for your advice. The previous sensitivity and specificity formulas are not suitable and have been changed to the following formulas:

Sensitivity = True Positives / (True Positives + False Negatives) × 100%

Specificity = True Negatives / (True Negatives + False Positives) × 100%

Comment 12: Is the term "double dilution" correct?

Response: Thanks for your advice. It has been modified to "equal proportion dilution".

Results:

Comment 1: In section 3.2, The authors analyse only *BP*-11, how about the other 15 isolates? Please explain. These large data may be included as supplementary material.

Response: Sequencing results showed that the genomes of 16 strains were all

composed of 2 chromosomes, and there were similarities in structure, but only BP-11 was found to have genetic structure-plasmid except 2 chromosomes in the whole genome, so BP-11 was used as a representative to display the egg NOG function annotation results of *Bp*. We've added content and optimized image presentation.

Comment 2: Please clearly label in Figure 2 the location of I, II and III.

Response: Thanks for your advice. Figure 2 has been labeled.

Comment 3: Did the authors check on the plasmid sequence? It will be interesting to know if there's similarity with other BP or organism, possible acquired?

Response: Only BP-11 has plasmid sequence, so we carried out preliminary analysis of plasmid and found 1,331 genes without eggNOG annotation. We suspect that these genes may be mainly related to DNA replication, recombination and repair, but further research is needed to support this claim.

Comment 4: It will be easier to compare if the information in Table 3 and 4 to be incorporated in Table 2.

Response: Thanks for your advice. Table 2 presents background information of 16 isolates, and Tables 3 and 4 show information such as functional classification of virulence factors in groups. We tried to combine all information together, but compared with the previous data, it seems that the data are more chaotic, so we still hope to divide them into 3 tables for description of results. I hope to get the approval of the review experts, thank you.

Comment 5: The authors should include the footnote and define the abbreviations used in the tables/figures.

Response: Thanks for your advice. The relevant content has been supplemented.

Comment 6: The authors may remove the word “clinical” and change to antibiotic susceptibility testing. AST is an in vitro method therefore it will be confusing to use

"clinical" term.

Response: Thanks for your advice. We've removed the word "clinical" and change to "Antimicrobial Susceptibility Testing".

Comment 7: The authors are suggested to check the presence of these genes from the whole genome analysis. Majority of the genes listed are chromosomally encoded in BP genomes. It will be good to discuss on the absence of the genes in typical isolates.

Response: Thanks for your advice. This study summarizes the basic characteristics of the genome of the Hainan epidemic strain of melioidosis and annotates the function of the genome. However, our ultimate goal is to find conserved sequences in the genomic sequence. According to sequencing results and literature reports, TS1, as a part of the Bp secretion system that plays a key pathogenic role, exists in all Bp and is highly conserved, so it was selected for follow-up research.

Comment 8: In Table 6, the term "carrier rate" is confusing. The authors may use "number of isolates" instead.

Response: Thanks for your advice. We have modified the terms in the table.

Comment 9: What are the specific gene families shown from the analysis?

Response: Specific gene family is one that is unique to the species and does not exist in any of the other organisms analyzed. In Figure 3, overlapping regions represent shared gene families between species, while non-overlapping regions indicate gene families specific to each species.

Comment 10: From the phylogenetic tree, it looks like the BP1,3 and 5 (death outcome) are diverged from the others. It will be good to include this in discussion.

Response: Thanks for your advice. We tried to analyze in the discussion section. For example, *BP*1,3,4 and 5 were closely related, but *BP* 1, 3 and 4 appeared in the death group, while *BP* 5 only appeared in the improvement group. This may indicate that although the genetic relationship is close, the level of virulence may be different,

which enhances the complexity of clinical manifestations after BP infection.

Comment 11: Please recheck on the sentence “Conversely, ORF2, located ...”

Response: Thanks for your advice. We've removed the word “Conversely” and change to “In addition, ORF2, located within the coding region of the TTS1 gene cluster, was specific to *Bp* and absent in closely related *Burkholderia species* such as *Burkholderia thailandensis*, and *Burkholderia cepacia*. Consequently, the TTS1-ORF2 genome sequence was chosen as the target fragment for RPA (GenBank: AF074878)”.

Comment 12: Did the authors perform the LOD limit on strip tests?

Response: Thanks for your advice. Limit of detection of the SHERLOCK strip test method has been supplemented.

Discussion:

Comment 1: The authors may improve on the discussion.

Response: Thanks for your advice. The discussions have been revised and supplemented accordingly.

Reviewer #2

Comment 1: Comparative Genomics and Epidemiology

While the study asserts that *B. pseudomallei* strains in Hainan have unique epidemiological characteristics, it would be more compelling to provide concrete evidence through comparative genome structural analysis. I recommend including a comparative analysis of genome structures between Hainan isolates and publicly available global isolates to substantiate the claim of distinct epidemiological traits.

Response: According to the whole genome structure of 16 strains of *BP* isolated from this third-generation sequencing and the genome structure of international standard strain K96243, it can be seen that the binary genome structure is the structural characteristics of the genetic material of *BP*. It is worth noting that a new genome structure except two chromosomes was found in the whole genome of BP-11. Sequencing analysis showed that the genome structure was a closed circular double-stranded DNA-plasmid with a length of 217,237bp. Before this, no plasmid was found in the genome structure of *BP*. Because the genome sequence of a large number of *BP* samples was completed by second generation sequencing technology, one of the disadvantages of second-generation sequencing technology is that it is difficult to distinguish plasmid and nuclear genome. Therefore, it is necessary to obtain sufficient genome sequencing results by three generation sequencing technology to verify the credibility of plasmid structure obtained by *BP* sequencing.

Comment 2: Petal Diagram Interpretation

In Figure 3, the petal diagram depicts core and unique gene families among the 16 isolates and the reference strain K96243. Please provide detailed information about the eight unique gene families of K96243 and the two unique gene families of BP-11. This information would enhance the understanding of gene content diversity and specificity.

Response: Thanks for your advice. We will display specific information such as the

speculated functions as follows, but the results need further verification.

Strain	Number of specific gene families	Function
BP-8	1	Enoyl- (Acyl carrier protein) reductase
BP-11	2	Integrase core domain ; Transposase
K96243	8	Immunoglobulin I-set domain ; Resolvase, N terminal domain ; H-NS histone family ; Helix-turn-helix domain ; Phage P2 GpE ; LysR substrate binding domain ; Phage integrase family ; Adenylylsulphate kinase

Comment 3: Diagnostic Specificity and Related Species

The SHERLOCK detection system targeting the TTS1-ORF2 gene is promising. However, assay specificity could be further validated by including closely related *Burkholderia* species such as *B. thailandensis*, *B. mallei*, and others. This would strengthen confidence in its clinical applicability and help rule out cross-reactivity.

Response: According to sequencing results and literature reports, TS1, as a part of the *Bp* secretion system that plays a key pathogenic role, exists in all *Bp* and is highly conserved. The ORF2 in the coding region of the TS1 gene cluster is limited to *Bp* and does not exist in other closely related *Burkholderia* species such as *Burkholderia thailai*, *Burkholderia cepacia*, *Burkholderia mallei*, etc., thus ruling out the possibility of cross-reactions in subsequent experiments.

Comment 4: Calculation of Sensitivity in Methods (Section 2.7.1)

The sensitivity formula used in Section 2.7.1 is incorrect. The correct formula should be:

Sensitivity = True Positives / (True Positives + False Negatives) × 100% Please

revise this accordingly.

Response: Thanks for your advice. The previous sensitivity and specificity formulas are not suitable and have been changed to the following formulas:

Sensitivity = True Positives / (True Positives + False Negatives) × 100%

Specificity = True Negatives / (True Negatives + False Positives) × 100%

Comment 5: Clarity of Terms in Discussion

In the Discussion section (third paragraph), the phrases "dead and improved strains" and "dead group" are confusing and potentially misleading. Please clarify that these and similar words throughout the manuscript refer to patients' clinical outcomes, not the bacterial strains themselves.

Response: Thanks for your advice. This refers to grouping patients according to their clinical outcomes, divided into death group and improve group, rather than the strain itself, which we have revised in manuscript.

Comment 6: Antibiotic Resistance Gene Analysis

The analysis of resistance genes is appreciated; however, it overlooks several key resistance determinants previously reported in *B. pseudomallei*. For example:

The *PenA* gene, known to confer resistance to ceftazidime and amoxicillin-clavulanate, should be discussed. See Hii SYF et al., *Antimicrob Agents Chemother*, 2021.

Recent evidence shows that *amrR* deletions contribute to carbapenem resistance. Refer to Nimnuan-ngam et al., 2025. Please investigate and discuss the presence or absence of these genes in your isolates.

Response: Thanks for your advice. We read the literature mentioned by the reviewers and learned that mutations or over-expression of the *PenA* gene can lead to CAZ resistance. In addition, the RND operon, encoding an inner membrane protein (*amrA*) and RND transporter (*amrB*), works with an OMP (*oprA*) to form a tripartite complex

under the regulation of *amrR*. These proteins work together to actively pump a variety of antimicrobial compounds out of the bacterial cell into the external environment. The specific mutations within the *amrR* gene contribute to the upregulation of *amrAB-oprA* efflux pump transcriptional levels, which are linked to MEM resistance. However, our research did not find changes in these genes, such as mutations or deletions. This may be related to the small number of strains tested in our experiments. We very much agree with the reviewers' opinions and mentioned relevant content during the discussion.

Comment 7: Unreferenced Claims in Introduction and Discussion

Several claims throughout the Introduction and Discussion lack appropriate citations. For instance, the claim that the *R39* gene "might produce beta-lactamases" should be supported by references or clarified as speculative. Please ensure that all assertions are backed by appropriate literature.

Response: Thanks for your advice. We reviewed the introduction and discussion sections and added some citations as needed. In addition, there is currently a lack of relevant research reports on the *R39* gene. We speculate based on the functional annotation results that it may produce beta lactamases, which is related to resistance to penicillin antibiotics. Of course, this conclusion needs further research and verification.

Comment 8: Table 2 contains patients' information. These require ethical approval.

Minor Suggestions:

- Please check for consistent gene naming (e.g., use italics for gene names such as *amrR*, *penA*, *katA* etc.).
- A visual summary or schematic of the SHERLOCK workflow could enhance readability for a broader audience.

Response: For some sample information shown in Table 2, we have conducted ethical

approval, and do not include patient details, which does not have any impact on patient interests. In addition, we have checked the naming and format of genes, and we have also added to SHERLOCK workflow, hoping to be accepted by reviewers.

1. DNA Extraction from Strain Suspension

2. Establishment of RPA system

3. Establishment of SHERLOCK system

Reviewer #3

General Comments

Comment 1: The title may be misleading, as it implies the SHERLOCK assay is specific to Hainan isolates. Consider clarifying whether the assay targets region-specific signatures or general Bp markers.

Response: Thanks for your advice. Our study was originally aimed at epidemic strains in Hainan, and SHERLOCK method was designed to detect them according to genome sequencing results.

Comment 2: The manuscript should include line numbers to facilitate reviewer feedback.

Response: Thanks for your advice. We will ask the editorial board for advice on the basic requirements for manuscript format, whether to add line numbers to the manuscript.

Comment 3: There are numerous typographical and formatting inconsistencies (e.g., “Eggnog,” “EggNOG,” and “eggnog” appear interchangeably). Standardize terminology throughout.

Response: Thanks for your advice. We checked the manuscript and corrected these incorrect formatting.

Comment 4: Figure legends lack sufficient detail. Each should be self-contained—describing the figure's content, purpose, and analytical methods.

Response: Thanks for your advice. We re-edited the annotation of each figure and described it in detail according to the serial number.

Comment 5: Several figures contain small or non-English labels and low-resolution

images. Highquality, English-labeled vector graphics should be submitted.

Response: Thanks for your advice. We have submitted higher resolution images and edited them as requested by reviewers.

Comment 6: Many factual statements are unsupported by citations. All claims, especially comparative or quantitative ones, must be appropriately referenced.

Response: Following the reviewer's suggestion, we checked the manuscript and added citations have been highlighted in red.

Abstract

Comment 1: The statement that detection is achieved in “1–2 hours” should be substantiated in the Results section, with clear experimental data or protocol timing.

Response : Thanks for your advice. Regarding the testing time, we have added a description to the results, and the details are as follows: “In addition, the time for nucleic acid extraction (according to the manufacturer's instructions, it takes about 20 minutes) and RPA amplification (it takes about 20 minutes) is about 40 minutes. The final SHERLOCK fluorescence test (fluorescence values were read after 20 minutes) took a total of 60 minutes. The SHERLOCK test strip method requires incubation at 37 degrees Celsius for 30 minutes before testing, it takes about 70 minutes in total. Therefore, both methods can achieve rapid detection of BP within 1-2 hours.”

Introduction

Comment 1: The review of genomic studies is incomplete and omits important contributions from others such as Australian and Thai research groups, for example.

Response: Thanks for your advice. We have supplemented relevant content. “The world's first strain of *Bp* (No. K96243) to undergo full genome sequencing was a patient from Thailand. In 2004, researchers at Sanger Laboratory in the United Kingdom completed its sequencing, assembly and functional annotation. K96243 was

used as an international standard strain and its genomic information was uploaded to NCBI in the United States to provide a reference for subsequent strain genomic research. In addition, in a genomic sequencing study of *Bp* from multiple countries, including 30 countries in Oceania, Asia, Africa, Central and South America, the results showed that there were significant genetic differences between isolates from Australia and Asia, and that isolates from Australia had longer phylogenetic branches. Combined with pan-genome analysis data, the final results show that Australia is an early reservoir for the current global population of *Bp* and is spreading to other regions.”

Comment 2: Similarly, the CRISPR diagnostics overview does not cite prior *Bp*-specific CRISPR assays developed by other groups, including those from China and Thailand. Acknowledging existing work will help situate the novelty of this study.

Response: Thanks for your advice. We supplemented relevant research from China and Thailand and presented the content as follows: “Recently, research teams from China and Thailand have also reported using CRISPR-Cas12a technology to achieve rapid detection of *BP*. However, the CRISPR RNA (crRNA) sequences used in the research are different, resulting in certain differences in the range of BP populations detected. Therefore, based on genomic sequence analysis of endemic melioidosis strains in Hainan, this study was aimed to design crRNA targeting specific and conserved gene sequences in the melioidosis genome, construct a SHERLOCK reaction system, and establish two technical platforms—fluorescence method and lateral chromatography test strip—to verify the method's accuracy and reliability for the rapid diagnosis of endemic melioidosis strains in Hainan.”

Methods

Comment 1: The Methods section lacks sufficient detail for replication. Information such as reagent sources, catalog numbers, and software versions should be included.

If space is a concern, detailed protocols can be placed in Supplementary Materials.

Response: Thanks for your advice. Information such as reagent source has been supplemented.

Comment 2: Culture conditions for *B. pseudomallei* and other bacteria should be clearly described, including media and incubation parameters.

Response: Thanks for your advice. In the method section, it is introduced that the collected strain samples are directly stored at -80°C and then used for subsequent experiments, without involving the cultivation of strains. Therefore, the conditions for culture are not described.

Comment 3: It is unclear whether clinical isolates or direct clinical specimens were used in the SHERLOCK assay. This distinction must be clarified in both the Methods and Results.

Response: Clinical isolates were used, as described in the methods.

Comment 4: The limit of detection (LOD) experiment should include the number of replicates performed. A minimum of three independent replicates per data point is generally expected.

Response: As the reviewer said, we also wrote in the method, “Each concentration was tested in triplicate to observe and record the fluorescence curve changes and detection lines.”

Results

Comment 1: Several analyses (e.g., CRISPR loci, functional annotations) are presented abruptly. Reorganize related results under thematic subheadings and provide better contextual transitions.

Response: Thanks for your advice. The results have been reorganized and provided

for better contextual transitions, with modifications highlighted with red marks.

Comment 2: Justify the choice of isolate BP-11 for detailed analysis.

Response: Sequencing results showed that the genomes of 16 strains were all composed of 2 chromosomes, and there were similarities in structure, but only BP-11 was found to have genetic structure-plasmid except 2 chromosomes in the whole genome, so *BP-11* was used as a representative to display the egg NOG function annotation results of *Bp*. We've added content and optimized image presentation.

Comment 3: Figures labeled “Figure 2-I,” “-II,” and “-III” are not clearly marked. Label all panels explicitly.

Response: Thanks for your advice. Figure 2 has been labeled.

Comment 4: Figure 3 appears simplistic and lacks scientific value. Reconsider whether it adds to the narrative or replace it with more informative content.

Response: Thanks for your advice. The petal diagram is mainly used to show the results of gene family clustering analysis between 16 isolates and the reference strain K96243.

Comment 5: A whole-genome phylogenetic analysis including all available *Bp* strains would strengthen the manuscript by placing Hainan isolates in global context.

Response: We randomly selected 16 strains from a total of 42 *Bp* strains for WGS. The results revealed that the genomes of these 16 strains each consisted of two chromosomes, with a total genome size ranging from 7.1 Mb to 7.3 Mb. Chromosome 1 was larger, while Chromosome 2 was smaller. The sequencing results showed that the genomic structure of the epidemic melioidosis strains in Hainan was similar. In subsequent experiments, conserved sequences were obtained from them to prepare for the establishment of the SHERLOCK system. Therefore, no in-depth analysis was

conducted in aspects such as genome-wide phylogenetic analysis, because this may require more samples and more financial investment.

Comment 6: Genome collinearity should be quantitatively described and discussed.

Response: Thanks for your advice. Collinearity analysis was used to describe the position and orientation of homologous genes. The alignment sequence and position of homologous genes were similar, so collinearity regions could be formed. Due to the limitation of methods, quantitative analysis could not be carried out.

Comment 7: The transition from genome analysis to SHERLOCK diagnostics lacks coherence. Currently, the CRISPR assay section seems disconnected. Consider integrating the genomic data to support assay design.

Response: Thanks for your advice. We made adjustments to the transition from genomic analysis to SHERLOCK diagnosis, and provided for better contextual transitions, with modifications highlighted with red marks.

Comment 8: The panel of non-Bp organisms (n=6) used to assess assay specificity is limited. Consider expanding the panel to include more closely related species or increase numbers of isolates

Response: *Burkholderia cepacia*, *Klebsiella pneumoniae*, *Pseudomonas aeruginosa*, *Escherichia coli*, *Staphylococcus aureus*, *Streptococcus pneumoniae* and other clinically common bacteria were set as control strains. The main purpose is to detect the specific differences between *Burkholderia pseudomallei* and other common clinical strains.

Comment 9: Statistical analysis is generally lacking across experiments. Include appropriate statistical tests to support claims.

Response: Thanks for your advice. Appropriate statistical analysis will be supplemented where necessary.

Comment 10: While VFDB and CARD databases are useful, discuss potential limitations and whether cross-validation with other tools was considered.

Response: Thanks for your advice. We described the pros and cons of using VFDB and CARD databases.

Comment 11: All sequencing data must be deposited in a public database prior to publication.

Response: Thanks for your advice. Sequencing data have been stored in the NCBI SRA database under the following number: PRJNA1280284.

Discussion

Comment 1: The Discussion is relatively stronger than other sections, providing interpretation and context. However, some of this information (e.g., rationale and implications) should be incorporated earlier into the Results section to help guide readers.

Response: Thanks for your advice. The content has been re-adjusted to make it more convenient for readers to read, with modifications highlighted with red marks.

Comment 2: A comparative discussion of the current SHERLOCK assay versus existing CRISPR diagnostics for Bp would be valuable for highlighting novelty and advantages.

Response: Thanks for your advice. Based on the timing of our research results, we found previous studies on Thailand and other places in China and compared them. We included the relevant content in the discussion section.

Conclusion

Comment 1: The phrase “we pioneered the use of the CRISPR/Cas” is not accurate

given the existence of prior studies using CRISPR diagnostics for Bp. Consider rephrasing to reflect the study's specific contributions (e.g., adaptation to local strains or integration with third-generation sequencing data).

Response: Thanks for your advice. Rephrasing this sentence for: “Additionally, we pioneered the use of the third-generation of sequencing results combined with the CRISPR/Cas system for detecting Bp...”

Re: Spectrum00592-25R1 (**Whole Genome Study and Construction of SHERLOCK Detection Method for Endemic Strains of Melioidosis in Hainan based on Third-generation Sequencing**)

Dear Dr. Xinping Chen:

Thank you for the privilege of reviewing your work. Below you will find my comments, instructions from the Spectrum editorial office, and the reviewer comments.

Revision Guidelines

Sincerely,

Vittal Ponraj Ph.D
Editor
Microbiology Spectrum

Reviewer #1 (Public repository details (Required)):

The authors need to specify the accession numbers for all the isolates used in this study.

Reviewer #1 (Comments for the Author):

The authors conducted a massive work on 3rd generation sequencing, optimisation of CRISPR-Cas technology and rapid

detection of *Burkholderia pseudomallei*. It is good that the author has prepared a graphical abstract as summary. However, there are several enquiries that need to be answered.

Generally, it is confusing on what the authors are trying to present in regards to the aim of the study as mentioned in the introduction. The authors should focus more on the rapid detection itself compared to other available target/mechanism in the discussion section. A proper citation on earlier works in methods and discussion section is very much appreciated.

Introduction

1) The authors need to use the correct term. It is very confusing. Melioidosis refers to the disease, *B. pseudomallei* is the bacteria causing melioidosis. Please do the necessary changes.

Methods

The methodology part is lengthy. The authors may consider to include the specific details in supplementary file.

1) Please use just one term: isolate or strain. It is misleading.

2)Line 144 The use of "human disease" is not appropriate. Instead, the authors may include the isolation site/sample type (Table 2).

3)Line 218 Antibiotic susceptibility testing should be sectioned separately. Please refer to CLSI M45, 2016 for testing and interpretation.

4)Line 298: A qPCR machine is required for fluorescence detection. Is it only one cycle at 37°C used? The authors need to specify the time required for the incubation and is ct value a factor to consider for this test?

5) Line 315: Is it correct that the result is still valid when the C line does not show colour. What does the C line incorporated with? Please ensure correct interpretation criteria is met.

6) The tested DNA were extracted from bacterial isolates or clinical samples? The sentence Line 326-327 is misleading.

Results

1)It will be easier to understand if the authors stated in Line 342 that 1 strain contains additional plasmid.

2)Did the authors observe the missing of group Y/Z in other BP genomes. It looks like BP11 has slightly smaller full genome size even with an additional plasmid. Does the presence of plasmid results in genes deletion in BP11?

3)How many bootstrap replicates used? There is no distance value available in Figure 4. Is this phylogenetic tree compared shared 5192 genes clusters generated from 3.3.1?

4)The authors may need to elaborate on the phylogenetic analysis. It is better to view in a global perspective. Recommended to add more isolates including earlier Hainan's (as mentioned in introduction) and from other countries.

5)Line 469: How many replicates were used for serial dilution testing?

Discussion

The content is lengthy and there is little discussion related to the aim of the study.

Reviewer #2 (Comments for the Author):

The authors have addressed all of my concerns.

Whole Genome Study and Construction of SHERLOCK Detection Method for Endemic Strains of Melioidosis in Hainan based on Third-generation Sequencing

Junjie Hu^{1,2}, Shanshan Xu¹, Zeng Zeng⁵, Wei Gong¹, Weihua Xu¹, Zhichao Ma¹, Shengmiao Fu⁴, Linhai Li³, Bin Xiao^{3*}, Xinping Chen^{1,2*}

1.Department of clinical Laboratory, Affiliated Cancer Hospital of Hainan Medical University, Hainan Cancer Hospital, Hainan 570311, P.R.China.

2.The First Clinical College of Hainan Medical University, Haihou, Hainan 571199, P.R.China.

3.Department of Laboratory Medicine, The Affiliated Qingyuan Hospital (Qingyuan People's Hospital), Guangzhou Medical University, Qingyuan, Guangdong 511518, P.R.China.

4.Hainan Lvtou Medical Laboratory Center, Haikou, Hainan 570206, P.R.China.

5.Sanya Hospital of Traditional Chinese Medicine, Sanya, Hainan 572000, P.R.China.

*Corresponding author: Xinping Chen, Bin Xiao.

E-mail: chenxinping52@126.com(Xinping Chen), xiaobin2518@163.com(Bin Xiao).

1. DNA Extraction from Strain Suspension

2. Establishment of RPA system

3. Establishment of SHERLOCK system

SHERLOCK workflow

ABSTRACT *Burkholderia pseudomallei* (*Bp*) is a gram-negative bacterium found in soil and surface water. It is also the pathogen that causes melioidosis disease in humans and animals. This study was aimed to obtain the whole genome sequence of the endemic strain of *Bp* in Hainan using third-generation sequencing (TGS) technology, and elucidate the genome structure, function, and genetic evolution. Additionally, the study aimed to achieve rapid and specific identification of these endemic strains using Specific High-sensitivity Enzymatic Reporter Unlocking (SHERLOCK) detection technology, providing a new strategy for the early diagnosis of melioidosis. Utilizing the PacBio platform for TGS technology, we completed whole genome sequencing (WGS) of 16 isolates from Hainan. High-precision and complete genome sequences were obtained through quality control and genome assembly of the sequencing data. Additionally, we established a nucleic acid detection technology platform based on SHERLOCK, which could be completed from nucleic acid extraction to result reading within 1-2 hours, demonstrating good sensitivity and specificity (both are 100%). The lateral chromatography strip method does not require special equipment and holds promise as an immediate screening method for the early diagnosis of melioidosis.

KEYWORDS *Burkholderia pseudomallei*, Whole genome sequence, Third generation sequencing, SHERLOCK; Early diagnosis

1 Introduction

Burkholderia pseudomallei (*Bp*) is a Gram-negative bacterium commonly found in soil and water in tropical and subtropical regions, especially in northern Australia and Southeast Asia. It actively invades cells through endocytosis, grows and multiplies within endocytic vesicles, and can infect nearly any organ in the body (1, 2). *Bp* demonstrates microecological stability, but environmental disturbances such as rainstorms, typhoons, or tsunamis can disrupt this balance, leading to disease spread and potential outbreaks. Annually, approximately 165,000 people globally are infected with *Bp*, resulting in about 89,000 deaths (3, 4). The world's first strain of *Bp* (No. K96243) to undergo full genome sequencing was a patient from Thailand. In 2004, researchers at Sanger Laboratory in the United Kingdom completed its sequencing, assembly and functional annotation. K96243 was used as an international standard strain and its genomic information was uploaded to NCBI in the United States to provide a reference for subsequent strain genomic research. In addition, in a genomic sequencing study of *Bp* from multiple countries, including 30 countries in Oceania, Asia, Africa, Central and South America, the results showed that there were significant genetic differences between isolates from Australia and Asia, and that isolates from Australia had longer phylogenetic branches. Combined with pan-genome analysis data, the final results show that Australia is an early reservoir for the current global population of *Bp* and is spreading to other regions (5, 6).

Hainan Island, China, situated in a tropical region, has a closed geographical location that fosters the growth and reproduction of *Bp*. The island's unique climate conditions also create favorable circumstances for *Bp* outbreaks. Reports of melioidosis

in Hainan have been increasing in recent years (7). The local population is highly susceptible to *Bp*, with most infections occurring through contaminated skin, and some through inhalation or ingestion of contaminated water. Notably, farmers show a higher incidence of *Bp* infection, likely due to their greater exposure to contaminated soil and water (8). In a study on the molecular epidemiology and population structure of *Bp* in Hainan Province, Xiong Zhu et al. analyzed 166 *Bp* isolates from 2002 to 2014 using Multilocus Variable Number Tandem Repeat Analysis (MLVA) and Multilocus Sequence Typing (MLST) techniques (9). The results identified 99 MLVA_4 genotypes among the strains, with 101 isolates containing 34 common genotypes and a clustering rate of 60.8% (101/166), suggesting a common infection source. The remaining 65 isolates displayed different genotypes, indicating that 39.2% (65/166) of melioidosis cases in Hainan were epidemiologically unrelated or sporadic. Additionally, the 166 isolates were categorized into 48 sequence types (ST), with 21 STs unique to single strains, highlighting the unique clonal populations of *Bp* in Hainan. Therefore, melioidosis in Hainan exhibits distinct epidemiological characteristics and pathogenic traits. Developing new detection technologies based on these metagenomic characteristics could improve diagnostic efficiency and reduce the mortality rate of melioidosis patients, particularly for Hainan's tropical disease profile.

The genome of *Bp* is notably diverse, with high guanine-cytosine (GC) content and numerous repetitive sequences, making it one of the largest and most complex bacterial genomes sequenced to date (10). Although many *Bp* genome sequences are stored in the National Center for Biotechnology Information (NCBI), most were obtained using second-generation sequencing technologies, with few reports based on third-generation sequencing (TGS). The Teng team compared *Bp* sequencing results from Illumina and PacBio RS platforms, confirming that PacBio's TGS technology offers superior completeness and accuracy (11). Utilizing the PacBio platform, we performed structural, functional, and comparative genomics analyses on endemic melioidosis strains in Hainan to further investigate the relationships among genome structure, function, and genetic evolution.

Currently, the isolation, culture, and identification of melioidosis remain the diagnostic gold standard. However, this method, while highly specific, is limited by low sensitivity and long processing times, hindering rapid early diagnosis. In 2017, Feng Zhang et al. demonstrated that CRISPR-associated protein 13a (Cas13a) can target RNA in mammalian cells and nonspecifically degrade other RNAs, including reporter RNA, through "collateral cleavage" (12). Building on this, Specific High-Sensitivity Enzymatic Reporter Unlocking (SHERLOCK) detection technology was developed, integrating Recombinase Polymerase Amplification (RPA), T7 transcription, Cas13a detection, and fluorescence signal collection. SHERLOCK has shown great potential in diagnosing various pathogens (13), such as detecting Zika virus in blood or urine within hours, distinguishing genetic sequences of Zika strains from the USA and Africa, identifying specific bacterial types, and detecting resistance genes (14, 15, 16). Recently, research teams from China and Thailand have also reported using CRISPR-Cas12a technology to achieve rapid detection of *Bp*. However,

the CRISPR RNA (crRNA) sequences used in the research are different, resulting in certain differences in the range of *Bp* populations detected (17, 18). Therefore, based on genomic sequence analysis of endemic melioidosis strains in Hainan, this study was aimed to design crRNA targeting specific and conserved gene sequences in the melioidosis genome, construct a SHERLOCK reaction system, and establish two technical platforms—fluorescence method and lateral chromatography test strip—to verify the method's accuracy and reliability for the rapid diagnosis of endemic melioidosis strains in Hainan.

2 Methods

2.1 Sample Collection

A total of 53 clinical isolates were collected from human disease, comprising 42 of *Bp*, 2 of *Burkholderia cepacia*, 2 of *Klebsiella pneumoniae*, 2 of *Pseudomonas aeruginosa*, 1 of *Escherichia coli*, 2 of *Staphylococcus aureus*, and 2 of *Streptococcus pneumoniae*. And all strains were stored at -80°C.

2.2 DNA extraction and concentration determination

The QIAGEN Genomic Tip 20/G kit (QIAGEN, Beijing, China) was used for nucleic acid extraction from the strain, following the manufacturer's instructions precisely. DNA concentration and purity were assessed using a Qubit fluorometer (Thermo Fisher Scientific, USA) and a Nanodrop spectrophotometer (Thermo Fisher Scientific, USA). The extracted DNA was then stored at -80°C for future use.

2.3 Library construction and whole genome sequencing (WGS)

2.3.1 Library construction and quality inspection

First, the Covaris G-tube was used to randomly fragment the DNA sample into 15-kilobase pairs (kb) fragments. Subsequently, the PacBio library preparation kit was used to remove single-stranded overhangs and repair DNA damage in the fragmented DNA. Dumbbell adapters were ligated to both the 3' and 5' ends of the DNA fragments, and excess fragments were degraded using exonuclease. Nucleic acid fragments were then purified with AMPure PBmagnetic beads (PACBIO, USA). The target fragments were recovered using the BluePippin instrument (Sage Science, USA) to create the sequencing libraries. The concentration of the libraries was quantitatively assessed using a Qubit fluorometer, while the size and suitability of the libraries were evaluated with an Agilent 2100 instrument (Santa Clara, USA).

2.3.2 WGS

De novo gene sequencing was conducted using the PacBio RS II System from Beijing Biomec Biotechnology Company. After the target library was bound to the primer and polymerase, sequencing was carried out on the Sequel II sequencer (PACBIO, USA). In this study, we randomly selected 16 of 42 *Bp* strains for whole genome sequencing.

2.4 Bioinformatics analysis

2.4.1 Pre-processing of original sequencing data and genome assembly

In WGS, the sequencing data initially consists of small fragments significantly shorter than the actual sequencing length. To obtain Circular Consensus Sequencing (CCS) sequences, it is necessary to perform cross-correction of subreads within the

Zero-Mode Waveguides (ZMWs). Segments with fewer than four passes are removed, as a CCS accuracy greater than 99% requires at least four passes. Using SMRT Link v8.0 software (<https://pacbio.cn/support/software-downloads/>), the raw data were processed with settings of minPredictedAccuracy 0.9 and minPasses ≥ 5 to generate CCS sequences. Clean CCS reads were obtained by filtering out connectors and short CCS reads (length < 2 kb). These clean CCS reads were then assembled into contiguous sequences (Contigs) or scaffolds using Hifiasm software (<https://hifiasm.readthedocs.io/en/latest/index.html>). The starting site was adjusted with Circlator v1.5.5 to complete the circularization, and Pilon v1.22 software (<https://github.com/broadinstitute/pilon/releases>) was used to correct errors based on second-generation data. This process ultimately produced a high-accuracy whole genome sequence.

2.4.2 Genome-wide component analysis

- (1) Encoding gene prediction: The Prodigal v2.6.3 software (<https://github.com/hyattpd/Prodigal>) was used to predict coding genes.
- (2) Prediction of genomic repeats: The RepeatMasker v4.0.5 software (<https://www.repeatmasker.org/>) was used to identify repetitive sequences in the genome.
- (3) Prediction of non-coding RNA: Non-coding RNAs were predicted using Infernal v1.1.3 (<http://eddylab.org/infernal/>) for ribosomal RNA (rRNA) and tRNAscan-SE v2.0 (<http://lowelab.ucsc.edu/tRNAscan-SE/>) for transfer RNA (tRNA).
- (4) Prediction of clustered regularly interspaced short palindromic repeats (CRISPR): CRISPR sequences in the genome were predicted using CRISPRfinder (<http://crispr.i2bc.paris-saclay.fr/Server/>).
- (5) Prediction of prophage: Prophage regions were predicted using PhiSpy v2.3 software (<https://pypi.org/project/PhiSpy/>).

2.4.3 Whole genome gene function annotation

- (1) EggNOG database comments: The RPS-Blast tool was utilized to compare gene protein sequences with the EggNOG database (<http://eggnog5.embl.de/#/app/home>) to identify the most similar sequences. The annotation and classification information retrieved from the EggNOG database were then used to annotate and classify the corresponding genes in the sequenced genome.
- (2) Virulence factor database (VFDB) database annotations: The protein sequences of the predicted genes were compared with the core data of the VFDB (<http://www.mgc.ac.cn/VFs/>), including experimentally confirmed virulence factors. The most similar sequences identified in the VFDB provided the annotation information for the corresponding genes in the sequenced strain's genome.
- (3) Comprehensive antibiotic resistance database (CARD) database annotations: Protein sequences of the sequenced genes were compared with the CARD (<https://card.mcmaster.ca/>) to identify antibiotic resistance genes. The Resistance Gene Identifier (RGI) tool within CARD was used to find the most similar sequences, thus obtaining information on corresponding antibiotic resistance genes. Additionally, susceptibility tests were performed according to the broth microdilution method recommended by CLSI (CLSI M100-Ed33 Performance Standards for Antimicrobial

Susceptibility Testing). This testing covered three antibiotics: Trimethoprim/sulfamethoxazole (SXT), Ceftazidime (CAZ), and Imipenem (IPM).

2.4.4 Mapping of the genome circle

Using the genomic information obtained from assembly and prediction, the genome was visualized as a circular map using Circos v0.66 software (<http://circos.ca/software/download/circos/>).

2.4.5 Comparative genomics research

(1) Gene family clustering and differential gene analysis: Gene family clustering is a crucial step in comparative genomics and taxonomic analysis for identifying subspecies genomes. Using the default parameters of OrthoMCL software (<http://orthomcl.org/orthomcl/>), homologous protein-coding sequences from 16 isolated strains and the reference strain K92643 were analyzed. OrthoMCL grouped these sequences into homologous clusters based on amino acid sequence similarity. The software produced a genetic homology matrix illustrating the genetic relationships among the strains, which was visualized in a petal map. This analysis identified core gene families, consisting of homologous genes present in all strains, as well as specific gene families, which lack direct homologous genes. Additionally, differential gene analysis was performed based on the clustering results to count and compare core and specific genes among the 16 isolates and the reference strain K92643.

(2) Phylogenetic analysis: The single-copy homologous gene of the sequenced strain were compared with those of the reference strain K96243. Phylogenetic relationships were studied by constructing a maximum likelihood (ML) tree using PhyML software (<http://www.atgc-montpellier.fr/phyml/>). This analysis aimed to elucidate the evolutionary relationships between the species.

(3) Gene collinearity analysis: Homologous gene locations on the genome were identified through protein sequence BLAST comparisons, and the collinearity relationships among genes were analyzed. Using MCScanX software (<https://sourceforge.net/projects/mcscanx/>), collinearity plots were generated to compare the reference strains with strain K96243.

2.5 Establishment of RPA system

Primer design was performed using Primer Premier 6.0 (<https://www.premierbiosoft.com/primerdesign/>). In the CRISPR/Cas13a system, the crRNA targets specific RNA sequences. Therefore, a T7 promoter sequence (TAATACGACTCACTATAGGG) was included at the 5' end of the forward primer. It facilitated the transcription of the RPA amplification product from DNA to RNA. From Table1, a total of 3 forward primers (F1-F3) and 3 reverse primers (R1-R3) were designed to form nine pairs of different primers: F1+R1, F1+R2, F1+R3, F2+R1, F2+R2, F2+R3, F3+R1, F3+R2, F3+R3.

Under the same conditions of DNA concentration, amplification temperature, and amplification time, nine pairs of different primers were tested for amplification. Gel electrophoresis was used to compare the amplification efficiency, and primers with superior efficiency were selected. Adjustments were made as necessary according to the instructions for the TwistAmp® Basic RPA Kit (TwistDx, London, UK, Catalog Number: TABASRT01KIT).

RPA amplification system (50 μ L) included Primer Free Rehydration buffer (29.5 μ L), DNase-Free water (11.2 μ L), Forward primer (2.4 μ L, 10 μ M), Reverse primer (2.4 μ L, 10 μ M), template DNA (*Bp* DNA, 2 μ L), Magnesium Acetate (MgOAC, 2 μ L, 280 mM). After adding MgOAC, the components were thoroughly mixed and subjected to brief centrifugation. The reaction mixture was promptly incubated at 39°C for 20 minutes. (Note: MgOAC should be added as the final step, and the reaction will start immediately upon addition.) The amplified RPA products were purified using the DNA Recovery Kit (TIANGEN, Beijing, China, Catalog Number: DP214).

For gel electrophoresis, a total of 2 μ L of 6 \times DNA loading buffer was mixed with 10 μ L of the RPA amplification product and loaded into the sample wells. A 12 μ L DNA Marker was used as a reference. The electrophoresis was conducted at 125 V for 30 minutes. The progress of the bands was monitored, and electrophoresis was stopped when the bands had migrated to the middle of the gel. The gel was then removed and analyzed using a gel imaging system to observe the results.

2.6 Establishment of SHERLOCK system

2.6.1 CrRNA design

The crRNA for the CRISPR/Cas13a system consisted of two components: a conserved repeat region and a spacer region. For this study, the *Bp* TTS1-ORF2 sequence was used, with the detection target located between the amplified RPA fragments and avoiding overlap with the RPA primer region. The target sequence was 28 nucleotides long and was reverse complementary to the spacer region of the crRNA. The sequence of the crRNA used was as follows: GAUUUAGACUACCCCAAAAACGAAGGGGACUAAAACATCTCGGCCGCAA GCAACCGGTGTGGGA.

2.6.2 Configuration of SHERLOCK reaction system

The SHERLOCK reaction system should be maintained at a low temperature throughout the preparation process and operated on ice. The SHERLOCK reaction system (20 μ L) included the following components: 10.6 μ L diethylpyrocarbonate DEPC-treated water, 2 μ L 10 \times buffer, 0.8 μ L rNTP, 0.6 μ L T7 RNA Polymerase, 1 μ L Ribonuclease Inhibitor, 1 μ L LwaCas13a (1 μ M), 1 μ L crRNA (1 μ M), 1 μ L Flu-RNA reporter/LFS-RNA reporter, 2 μ L RPA product.

2.6.3 Establishment of SHERLOCK fluorescence detection system

The reaction mixture was transferred to a quantitative polymerase chain reaction (qPCR) tube and placed in the qPCR instrument. The reaction temperature was set to 37°C, and fluorescence intensity was measured every minute to monitor changes in the fluorescence signal. Results were interpreted as follows: (1) A positive result was indicated by a significant change in fluorescence signal, with the curve displaying substantial warping or reaching a plateau; (2) A negative result was indicated by the absence of a valid signal, characterized by a consistently flat curve.

2.6.4 Establishment of SHERLOCK lateral immunochromatography strip system

The reaction mixture was transferred to a 0.2 mL PCR tube and placed in a PCR instrument. The reaction was conducted at 37°C for 30 minutes. Following the reaction, 30 μ L of DEPC-treated water was added to the centrifuge tube to bring the total volume to 50 μ L. The mixture was then mixed and tested. The binding pad end of the test strip

was inserted into the PCR tube, ensuring that the liquid level did not exceed the top of the binding pad. The test was allowed to proceed for 1 to 2 minutes to ensure full infiltration of the interpretation area. Once the quality control line (C line) developed color, the test strip was removed and read. Results should be interpreted within 10 minutes of the appearance of C line, as results would be invalid after this time. Result interpretations were as follows: (1) A positive result was indicated if both the test line (T line) and C line showed red bands, or if only the T line showed red bands and the C line was colorless; (2) A negative result was indicated if only the C line showed a red band and no red band was visible on the T line.

2.7 Analysis of the sensitivity, specificity and limit of detection (LOD)

To evaluate the performance of the SHERLOCK detection system, we used the culture method as the gold standard. A total of 42 *Bp* isolates were tested, along with 11 control strains representing common clinical bacteria, including *Burkholderia cepacia*, *Klebsiella pneumoniae*, *Pseudomonas aeruginosa*, *Escherichia coli*, *Staphylococcus aureus*, and *Streptococcus pneumoniae*. Additionally, DEPC-treated water was used as a negative control. DNA was extracted from the sample, and subsequent detection was performed using both SHERLOCK fluorescence detection and lateral flow immunochromatographic strip detection after RPA. The results from these methods were compared to those obtained using traditional culture techniques. The sensitivity was calculated using the formula: Sensitivity = True Positives / (True Positives + False Negatives) × 100%. The specificity was calculated using the formula: Specificity = True Negatives / (True Negatives + False Positives) × 100%.

Bp DNA concentration was serially diluted from 1 ng/μL to 10⁻¹, 10⁻², 10⁻³, 10⁻⁴, 10⁻⁵, 10⁻⁶, and 10⁻⁷ ng/μL using equally proportional dilution method, and RPA was performed for each dilution. The negative control template was substituted with DEPC-treated water. The diluted DNA templates were then used for both SHERLOCK fluorescence detection and lateral flow immunochromatographic strip detection. Each concentration was tested in triplicate to observe and record the fluorescence curve changes and detection lines.

3 Results

3.1 Structural genomics analysis

We randomly selected 16 strains from a total of 42 *Bp* strains for WGS. The results revealed that the genomes of these 16 strains each consisted of two chromosomes, with a total genome size ranging from 7.1 Mb to 7.3 Mb. Chromosome 1 was larger, while Chromosome 2 was smaller. Gene prediction using Prodigal v2.6.3 software estimated that each genome contains between 5,759 and 6,011 protein-coding genes. Additionally, all strains had 12 rRNA genes, including 4 each of 5S rRNA, 16S rRNA, and 23S rRNA. The number of tRNA genes ranged from a minimum of 62 to a maximum of 66. The length of genomic repeats in the 16 isolates varied from 275,435 base pairs (bp) to 285,614 bp. All isolates contained four CRISPR sequences on chromosome 1. The number of prophages varied between chromosomes 1 and 2, with a higher number of prophages found on chromosome 2 compared to chromosome 1. Notably, the genome of strain *Bp*-11 included three prophages: two located on its plasmid and one on

chromosome 2.

3.2 Functional genomics analysis

3.2.1 EggNOG functional annotation

In the whole genome of strain *Bp*-11, besides the two chromosomes, a plasmid was also identified. The EggNOG functional annotation results for *Bp*-11 are shown in Figure 1. The functional annotations of coding DNA sequences (CDSs) were categorized into 23 groups, each represented by a different color. Group S, whose functions were unknown, did not have any annotated representatives. The remaining 22 groups represented various functional annotations. Of the 4,614 annotated genes on the two chromosomes of *Bp*-11, 825 genes were unannotated and their functions were unknown, accounting for 17.48% of all CDSs. Figure 1 illustrates that the top three functional categories were Groups E (amino acid transport and metabolism), R (general function prediction), and C (energy generation and conversion). Notably, no genes in the genome were annotated for Group Y (nuclear structure) or Group Z (cytoskeleton).

3.2.2 Functional classification and genomic localization

The genome circle for *Bp*-11 is depicted as follows: Figure 2I shows chromosome 1, which is 3,929,742 bp in size; Figure 2II shows chromosome 2, which was 3,105,737 bp; and Figure 2III shows the plasmid, a closed circular double-stranded DNA structure with a size of 217,237 bp. Compared to the chromosomes, the plasmid had fewer repeat sequences and lower GC content, and lacked tRNA and rRNA genes.

3.2.3 VFDB database annotation

The VFDB was used for identifying virulence factors. To analyze the correlation between virulence genes and clinical outcomes, the 16 isolates were categorized into two groups: the death group (5 isolates) and the improved group (11 isolates) based on patients' clinical outcomes (Table 2). A total of 17 virulence factors were identified using VFDB annotations. Virulence genes such as *katA*, *algU*, *clbD*, *clbF*, *Cj1435c*, *ybtS*, *recN*, *fbpC iron(III)*, *cdpA*, *cheA*, and *rcsB* were present in both the dead group and improved group. The *mrkD* gene, which encodes a type 3 pilin protein, was only found in the death group. The genes *porB*, *allB*, *allC*, *pvdQ*, and *bprA* were unique to strains *Bp*-7 and *Bp*-16 in the improved group. Detailed information is provided in Table 3 and 4.

3.2.4 CARD database annotation

The CARD was used for annotating antibiotic resistance genes. Prior to CARD database annotation, antimicrobial susceptibility testing (AST) showed that all 16 isolates were sensitive to SXT, CAZ, and IPM (Table 5).

CARD database annotation identified a total of 7 antibiotic resistance genes: *adeF*, *amrA*, *amrB*, *OXA-57*, *OXA-59*, *R39*, and *Omp38*. The annotation results for these genes are detailed in Table 6. The resistance mechanisms associated with these genes include: *adeF*, *amrA*, and *amrB* are involved in antibiotic efflux pumps; *OXA-57*, *OXA-59*, and *R39* contribute to antibiotic inactivation; and *Omp38* reduces antibiotic permeability.

3.3 Comparative genomic analysis

3.3.1 Gene family clustering and differential gene analysis

Gene family clustering was performed for 16 isolates and the reference strain

K96243, and the results were visualized using a petal diagram (Figure 3). The analysis revealed that the 16 strains shared 5,192 gene families. Specific gene families were identified as follows: *Bp*-8 had 1 specific gene family, *Bp*-11 had 2, and K96243 had 8. No specific gene families were found in the remaining strains.

3.3.2 Phylogenetic analysis

Gene clustering and phylogenetic analysis were conducted for the 16 isolates and the reference strain K96243, resulting in the construction of a phylogenetic tree (Figure 4). Branches with bootstrap values greater than 75 were considered reliable. In this phylogenetic tree, the branches for strains *Bp*-2, *Bp*-7, *Bp*-13, and *Bp*-16 had a support value of 57, indicating that these branches were unreliable. In contrast, other branches had support values ranging from 89 to 100, demonstrating their reliability. Although the 16 isolates of melioidosis from Hainan were distributed across various branches of the phylogenetic tree, they ultimately converged at a single root node, indicating that they belong to the same group. Strains *Bp*-12 and *Bp*-14 were found to be the most closely related, followed by *Bp*-10 and *Bp*-9, with *Bp*-11 being most closely related to K96243.

3.3.3 Genome collinearity analysis

Genome collinearity was analyzed by connecting homologous genes between the genomes of the 16 sequenced strains and the reference strain K96243. Different colors represented different homologous gene pairs in the collinear map (Figure 5). The analysis revealed multiple homologous genes shared between the 16 isolates and the reference strain K96243.

3.4 Establishment and optimization of specific SHERLOCK detection system

3.4.1 Determine the target gene of *Bp*

Based on sequencing results and literature reports, TTS1, a component of the *Bp* secretion system critical for its pathogenicity, was present in all *Bp* strains and was highly conserved. In addition, ORF2, located within the coding region of the TTS1 gene cluster, was specific to *Bp* and absent in closely related *Burkholderia* species such as *Burkholderia thailandensis*, and *Burkholderia cepacia*. Consequently, the TTS1-ORF2 genome sequence (GenBank: AF074878) was chosen as the target fragment for RPA, after synthesizing crRNA complementary to the target sequence, SHERLOCK detection system was established.

3.4.2 RPA Primer Screening

Based on the structural characteristics of the TTS1-ORF2 genome sequence, nine pairs of primers were designed and used for RPA amplification. Electrophoretic analysis of the amplification products revealed a target band of approximately 200 bp, with bands at 400–500 bp attributed to non-specific amplification due to the T7 promoter (Figure 6A). Among the primers, those pairs F3+R1, F3+R2, and F3+R3 produced clear bands and demonstrated good amplification efficiency. ImageJ software was employed to assess the brightness of the bands, revealing that the F3+R1, F3+R2, and F3+R3 primer pairs exhibited the strongest brightness (Figure 6B). Combining these observations, it was determined that the F3+R1, F3+R2, and F3+R3 primer pairs had the highest amplification efficiency for the TTS1-ORF2 gene in RPA. To evaluate their performance in the SHERLOCK fluorescence detection system, the F3+R1, F3+R2, and F3+R3 primer pairs were tested. Results showed that under identical

amplification conditions, the F3+R3 primer pair produced the highest fluorescence signal increase (Figure 6C), indicating the best amplification efficiency. Therefore, the F3+R3 primer pair was selected for subsequent studies.

3.5 Sensitivity and Specificity Analysis of SHERLOCK Detection System

For the SHERLOCK fluorescence method, all 42 *Bp*-positive isolates showed a clear upward trend in fluorescence curves compared to the control group. The minimum fluorescence value reached approximately 10,000, and most reactions can form obvious increasing signals within 20 minutes, indicating positive detection results (Figure 7A and Figure 7B). In contrast, no significant upward trend was observed in the curves for the 11 control strains and DEPC-treated water; these curves were similar to the negative control, resulting in negative test outcomes (Figure 7C).

For the SHERLOCK strip test method, all 42 *Bp*-positive isolates exhibited clear detection lines, confirming positive results (Figure 7D). In the test strips corresponding to the 11 control strains, only the quality control bands were visible, and no T lines appeared, indicating negative results (Figure 7E).

In summary, both the SHERLOCK fluorescence method and strip test method demonstrated consistent results, with a sensitivity of 100%. The detection of the 11 control strains was consistently negative, confirming that the SHERLOCK system has a high specificity (100%) and does not cross-react with other bacterial strains. In addition, the time for nucleic acid extraction (according to the manufacturer's instructions, it takes about 20 minutes) and RPA amplification (it takes about 20 minutes) is about 40 minutes. The final SHERLOCK fluorescence test (fluorescence values were read after 20 minutes) took a total of 60 minutes. The SHERLOCK test strip method requires incubation at 37°C degrees Celsius for 30 minutes before testing, it takes about 70 minutes in total. Therefore, both methods can achieve rapid detection of *Bp* within 1-2 hours.

3.6 Analysis of the LOD of the SHERLOCK fluorescence method

To determine the LOD of the SHERLOCK fluorescence method for *Bp* nucleic acids, *Bp* strains were serially diluted using a 10-fold gradient method, ranging from 10^{-1} ng/ μ L to 10^{-7} ng/ μ L. The LOD was assessed using these diluted samples, with DEPC-treated water serving as a negative control.

The results indicated that as the concentration of *Bp* nucleic acids was diluted, the real-time fluorescence values progressively decreased, correlating with a reduction in detection sensitivity. The LOD was determined to be 10^{-4} ng/ μ L (100 fg), as depicted in Figure 8A. Additionally, fluorescence values were detectable within 20 minutes of the reaction's initiation, demonstrating that the SHERLOCK fluorescence method enables rapid detection of target nucleic acids within a short timeframe (Figure 8B). In addition, the SHERLOCK test strip method was used for testing (repeated testing for each concentration 3 times, requiring a detection rate of 100%), and the final limit of detection was 10^{-3} ng/ μ L (Table 7).

4 Discussion

Bp is the causative agent of melioidosis, a zoonotic infectious disease with a mortality rate ranging from 20% to 50% in acute cases. This rate is even higher in resource-limited regions with inadequate diagnostic facilities and equipment,

particularly among patients with severe comorbidities (19, 20, 21). Currently, research on the whole genome of *Bp* primarily relies on second-generation sequencing technologies (22, 23, 24). Short-read sequencing often fails to provide complete genome and gene location information, and it struggles to differentiate between the nuclear genome and plasmids. In contrast, TGS technologies offer more accurate genome sequencing of *Bp*. For example, Teng compared WGS results of *Bp* using the Illumina platform with those obtained from the PacBio single-molecule real-time sequencing platform. The study demonstrated that PacBio sequencing, followed by de novo assembly, yielded a complete genome sequence for *Bp* without gaps or mismatches (11). Additionally, Ghazali used the PacBio RS II sequencing platform to sequence *Bp* strains from Malaysia, achieving a genome assembly completeness of 99.1% to 99.7%, as assessed by the Burkholderiales_odb5 lineage database (25).

Hainan Island's relatively isolated geographical location has contributed to a stable melioidosis genetic profile over the past 15 years, with only a few new STs emerging (26). Most of the genetic information within the cloned populations has remained consistent, ensuring the reliability of genomic data obtained from metagenomic analysis of strains isolated from the island. Kang et al. successfully isolated the HNBP001 strain from the blood of a patient with melioidosis and pneumonia in Hainan, sequencing it using the Illumina HiSeq4000 platform. The results were consistent with those reported by Johnson et al. (27, 28). In this study, we utilized the PacBio sequencing platform to obtain high-integrity and accurate whole genome sequences of Hainan endemic strains of melioidosis. The genome sizes ranged from 7.1 Mb to 7.3 Mb, with GC content between 68.06% and 68.29%. The total number of CDSs in the assembled genomes ranged from 5,759 to 6,011. Each genome included four 5S rRNAs, four 16S rRNAs, and four 23S rRNAs. The number of tRNAs ranged from 62 to 66, and the length of genomic repeat sequences ranged from 275,435 bp to 285,614 bp. Furthermore, all 16 strains contained four CRISPR sequences associated with bacterial immunity on chromosome 1. We also identified multiple prophages in the melioidosis genomes. Previous research suggests that prophage structures may correlate with virulence and antibiotic resistance characteristics. Variations in the location and number of prophages among different strains suggest potential differences in virulence and antibiotic resistance profiles.

Compared with other databases, VFDB database is more focused on collecting and updating virulence factor information of bacterial pathogens, and is widely used in bioinformatics research, especially in the prediction and analysis of virulence factors. This study identified a total of 17 virulence factors in the analyzed strains by VFDB database. Among these, the genes *katA*, *algU*, *clbD*, *clbF*, *Cj1435c*, *ybtS*, *recN*, *fbpCiron(III)*, *cdpA*, *cheA*, and *rscB* were present in both the dead group and improved group. The *mrkD* gene, which encodes type 3 fimbrial protein, was exclusively found in the dead group. This suggests that the presence of the *mrkD* gene might contribute to the higher virulence observed in the dead group, considering it is associated with the production of type 3 fimbrial protein, which is critical for bacterial adherence and pathogenicity. Previous studies have identified other virulence factors for *Bp*, including the type 3 secretion system (T3SS) gene cluster, the type 6 secretion system (T6SS)

gene cluster, the *bimA* gene of the autotransporter complex, the *fliC* gene encoding flagellin, and the *wcb* gene cluster associated with capsular polysaccharides (29). The virulence factors identified in this study differ from those reported in previous research, indicating that the 16 isolates from Hainan may have unique virulence profiles. This suggests regional variations in the virulence characteristics of *Bp* isolates, with potentially different distributions and impacts compared to isolates from other regions. The virulence genes identified in this study not only provide insights into the pathogenic mechanisms of *Bp* but also offer potential targets for molecular diagnostics and the development of melioidosis vaccines.

Bp exhibits natural resistance to many commonly used antibiotics, limiting the treatment options available for clinical management. The mechanisms of resistance include enzymatic inactivation, altered target sites, and efflux from the bacterial cell. In addition, changes in certain specific genes are also responsible for natural resistance, such as mutations or over-expression of the *PenA* gene can lead to CAZ resistance. The specific mutations within the *amrR* gene contribute to the upregulation of *amrAB-oprA* efflux pump transcriptional levels, which are linked to MEM resistance (30). CARD database can not only search for identified resistance genes in target strains, but also predict potential resistance genes with its RGI tool. Although it requires users to have certain professional knowledge and operational skills, it is still one of the most popular tools for drug resistance gene research. In this study, 16 *Bp* isolates from Hainan were analyzed by CARD database, and seven antibiotic resistance genes were identified: *adeF*, *amrA*, *amrB*, *OXA-57*, *OXA-59*, *R39*, and *Omp38*. Among the identified resistance genes, *adeF*, *amrA*, and *amrB* were associated with the production of antibiotic efflux pumps. Specifically, *AdeF* is involved in mediating resistance to tetracyclines and fluoroquinolones by actively pumping these antibiotics out of the bacterial cell. While *amrA* and *amrB* are two important components of the multiple antibiotic efflux pump *AmrAB-OprA* of *Bp*, which are related to resistance to aminoglycoside antibiotics (30). In contrast, *OXA-57* and *OXA-59* are common genes in the genome of *Bp*, mainly expressing class D β -lactamases. However, most studies have shown that the resistance of *Bp* to cephalosporins is mainly related to class A β -lactamases, which is mediated by the resistance gene *PenA* (31). The *omp38* is a unique gene in the genome of *Bp*, mainly encoding a pore protein located on the cell membrane of *Bp*, but studies have shown that it has nothing to do with antibiotic resistance (32). In addition, there is no relevant research report on the drug resistance gene *R39* of *Bp*. According to the functional annotation results, it is speculated that it may produce β -lactamases, which are related to penicillin antibiotic resistance, and further analysis is needed. The AST of the 16 isolates demonstrated sensitivity to SXT, CAZ, and IPM. The absence of the *BpeEF-OprC* efflux pump structure in the genome of these strains aligns with their 100% sensitivity to SXT.

Gene family analysis of the 16 isolates and the reference strain *Bp* K96243 revealed a core genome consisting of 5,192 genes. Phylogenetic analysis indicated that all 16 strains are closely related, sharing a common ancestor and belonging to the same group. Interestingly, *Bp*1,3,4 and 5 were closely related, but *Bp* 1, 3 and 4 appeared in the death group, while *Bp* 5 only appeared in the improvement group. This may

indicate that although the genetic relationship is close, the level of virulence may be different, which enhances the complexity of clinical manifestations after *Bp* infection. Genome collinearity analysis demonstrated a substantial number of homologous genes between the 16 sequenced strains and the reference strain K96243. The collinearity maps showed variability in gene location and color, reflecting the genetic diversity among the isolates. These findings suggest that the melioidosis isolates from Hainan exhibit notable genetic diversity. This diversity may be attributed to selection pressures that have driven adaptive evolution in some strains, leading to the observed variations in their genomes.

SHERLOCK technique has been used to detect many pathogens since it was invented. In recent years, there have been related reports on the diagnosis of pseudomallei. However, the specific target gene sequences of *Bp* detected based on CRISPR/Cas system are different. For example, Zhang et al. identified 44 specific sequence tags from the core genome sequence of chromosome 1 and chromosome 2 of *Bp* by bioinformatics, two of which were selected for the development of dual-target RPA-CRISPR/Cas 12a detection method, which finally realized the detection of *Bp* and showed high specificity (17). In another study in Thailand, researchers searched for a conserved CRISPR-Cas12a target through computer simulation, and named the obtained highly specific target sequence of *Bp* crBP34. Based on this sequence, a CRISPR-Cas12a diagnostic method was developed that was able to detect all clinical isolates of endemic *B. pseudomallei* while distinguishing humans from other pathogens, including its closely related species *B. thailandensis* (18). Different from the above studies, in this study, we developed a SHERLOCK system for the rapid detection of melioidosis by targeting TTS1-ORF2, a specific and conserved sequence in *Bp*. This system integrates the CRISPR/Cas13a technology with RPA, offering an effective method for the swift detection of melioidosis in this region. Methodological evaluation demonstrated that the SHERLOCK system can detect Hainan endemic strains of melioidosis with a minimum detection limit of 100 femtograms (fg). Importantly, the system showed no cross-reactivity with common pathogens that might be confused with melioidosis in clinical settings. Results can be interpreted through fluorescence signal changes or lateral flow immunochromatographic strips. Compared to traditional isolation and culture methods, this system reduced the time required from nucleic acid extraction to result reading to just 1-2 hours, making it suitable for early and rapid diagnosis of melioidosis. Unlike serological tests, this method was not dependent on antibody titers and provides stable sensitivity and specificity (33). Additionally, the SHERLOCK test strip method did not require specialized equipment or complex procedures, making it easy to use and transport. The test strips were user-friendly and cost-effective, ideal for deployment in primary medical institutions, especially in melioidosis-endemic areas. Furthermore, the CRISPR/Cas13a system's ability to detect single-base mismatches could help minimize non-specific reactions and reduce the incidence of false positive results.

5 Conclusion

In summary, this study adopted TGS technology to acquire accurate whole

genome sequences of *Bp*, coupled with bioinformatics analysis to enhance our understanding of the molecular structure, function, and pathogenic mechanisms of *Bp*. It provided valuable insights into the drug resistance characteristics and genetic evolution of the pathogen. Additionally, we pioneered the use of the third-generation of sequencing results combined with the CRISPR/Cas system for detecting *Bp*, establishing a novel detection method specifically tailored for melioidosis in Hainan. This new approach represents a significant advancement in rapid diagnostic techniques. Future studies will focus on further optimizing the design of crRNAs and RPA primers to enhance the sensitivity and efficiency of the detection method.

Data Availability Statement

The labeled dataset used to support the findings of this study are available from the corresponding author upon request.

Ethics statement

The studies were conducted in accordance with the local legislation and institutional requirements. This study uses strains obtained from the remaining samples after clinical testing. Ethics Committee of Hainan Cancer Hospital did not require the study to be reviewed or approved by an ethics committee because it used post-test samples.

Author contributions

JjH: Conceptualization, Formal analysis, Methodology, Visualization, Writing – original draft. SsX: Writing – review & editing. ZZ: Writing – review & editing. WG: Writing – review & editing. WhX: Writing – review & editing. ZzC: Writing – review & editing. SmF: Resources, Writing – review & editing. LhL: Resources, Writing – review & editing. BX: Supervision, Validation, Writing – review & editing. XpC: Supervision, Validation, Writing – review & editing.

Funding

Key Research and Development Project of Hainan Province, ZDYF2022SHFZ023
Key Research and Development Project of Hainan Province, ZDYF2021SHFZ085
Health Science and Technology Innovation Joint Project of Hainan Province, WSJK2024QN103

Acknowledgments

We would like to thank our laboratory colleagues for their assistance in the data and sample collection, and laboratory analysis. We would also like to thank the research teams of the Second Affiliated Hospital of Hainan Medical University and Sanya Central Hospital for their support for this study, because all samples were obtained from these two hospitals.

Conflict of interest

The authors declare that the research was conducted in the absence of any commercial or financial relationships that could be construed as a potential conflict of interest.

References

1. Katherine A Rhodes, Herbert P Schweizer. (2016). Antibiotic resistance in Burkholderia species. *Drug Resist Updat* 28, 82-90. doi: 10.1016/j.drup.2016.07.003
2. Dawson P, Duwell MM, Elrod MG, Thompson RJ, Crum DA, Jacobs RM, Gee JE, Kolton CB, Liu L, Blaney DD, Thomas LG, Sockwell D, Weiner Z, Bower WA, Hoffmaster AR, Salzer JS. (2021). Human Melioidosis Caused by Novel Transmission of Burkholderia pseudomallei from Freshwater Home Aquarium, United States. *Emerg Infect Dis* 27(12):3030-3035. doi: 10.3201/eid2712.211756
3. Direk Limmathurotsakul, Nick Golding, David A B Dance, Jane P Messina, David M Pigott, Catherine L Moyes, et al. (2016). Predicted global distribution of Burkholderia pseudomallei and burden of melioidosis. *Nature Microbiology* 1: 15008. doi: 10.1038/nmicrobiol.2015.8
4. Currie BJ. (2022). Melioidosis and Burkholderia pseudomallei: progress in epidemiology, diagnosis, treatment and vaccination. *Curr Opin Infect Dis* 35(6):517-523. doi: 10.1097/QCO.0000000000000869
5. Holden MT, Titball RW, Peacock SJ, Cerdeño-Tárraga AM, Atkins T, Crossman LC, Pitt T, Churcher C, Mungall K, Bentley SD, Sebahia M, Thomson NR, Bason N, Beacham IR, Brooks K, Brown KA, Brown NF, Challis GL, Cherevach I, Chillingworth T, Cronin A, Crossett B, Davis P, DeShazer D, Feltwell T, Fraser A, Hance Z, Hauser H, Holroyd S, Jagels K, Keith KE, Maddison M, Moule S, Price C, Quail MA, Rabinowitsch E, Rutherford K, Sanders M, Simmonds M, Songsivilai S, Stevens K, Tumapa S, Vesaratchavest M, Whitehead S, Yeats C, Barrell BG, Oyston PC, Parkhill J. (2004). *Proceedings of the National Academy of Sciences of the United States of America* 101(39):14240-14245. doi: 10.1073/pnas.0403302101
6. Chewapreecha C, Holden MT, Vehkala M, Välimäki N, Yang Z, Harris SR, Mather AE, Tuanyok A, De Smet B, Le Hello S, Bizet C, Mayo M, Wuthiekanun V, Limmathurotsakul D, Phetsouvanh R, Spratt BG, Corander J, Keim P, Dougan G, Dance DA, Currie BJ, Parkhill J, Peacock SJ. (2017). Global and regional dissemination and evolution of Burkholderia pseudomallei. *Nat Microbiol* 2:16263. doi: 10.1038/nmicrobiol.2016.263
7. Wu H, Huang D, Wu B, Pan M, Lu B. (2019). Fatal deep venous thrombosis and pulmonary embolism secondary to melioidosis in China: case report and literature review. *BMC Infect Dis* 19(1):984. doi: 10.1186/s12879-019-4627-6
8. Wang Y, Li X, Li A, Chen C, Fang J, Luo N, Tian S, Chen L, Wu X, Song X, Tan J, Zhang Y, Zhu Q, Li Y, Xiong Y, Pei H, Xia Q. (2024). The genetic diversity and evolution analysis of the Hainan melioidosis outbreak strains. *Infect Genet Evol* 123:105654. doi: 10.1016/j.meegid.2024.105654
9. Zhu X, Chen H, Li S, Wang LC, Wu DR, Wang XM, Chen RS, Li ZJ, Liu ZG. Molecular Characteristics of Burkholderia pseudomallei Collected From Humans in

Hainan, China. (2020). *Front Microbiol* 11:778. doi: 10.3389/fmicb.2020.00778

10. Taitt CR, Leski TA, Compton JR, Chen A, Berk KL, Dorsey RW, Sozhamannan S, Dutt DL, Vora GJ. (2024). Impact of template denaturation prior to whole genome amplification on gene detection in high GC-content species, *Burkholderia mallei* and *B. pseudomallei*. *BMC Res Notes* 17(1):70. doi: 10.1186/s13104-024-06717-8
11. Teng JLL, Yeung ML, Chan E, Jia L, Lin CH, Huang Y, Tse H, Wong SSY, Sham PC, Lau SKP, Woo PCY. (2017). PacBio But Not Illumina Technology Can Achieve Fast, Accurate and Complete Closure of the High GC, Complex *Burkholderia pseudomallei* Two-Chromosome Genome. *Front Microbiol* 8:1448. doi: 10.3389/fmicb.2017.01448
12. Cox DBT, Gootenberg JS, Abudayyeh OO, Franklin B, Kellner MJ, Joung J, Zhang F. (2017). RNA editing with CRISPR-Cas13. *Science* 358(6366):1019-1027. doi: 10.1126/science.aaq0180
13. Gootenberg JS, Abudayyeh OO, Lee JW, Essletzbichler P, Dy AJ, Joung J, Verdine V, Donghia N, Daringer NM, Freije CA, Myhrvold C, Bhattacharyya RP, Livny J, Regev A, Koonin EV, Hung DT, Sabeti PC, Collins JJ, Zhang F. (2017). Nucleic acid detection with CRISPR-Cas13a/C2c2. *Science* 356(6336):438-442. doi: 10.1126/science.aam9321
14. Myhrvold C, Freije CA, Gootenberg JS, Abudayyeh OO, Metsky HC, Durbin AF, Kellner MJ, Tan AL, Paul LM, Parham LA, Garcia KF, Barnes KG, Chak B, Mondini A, Nogueira ML, Isern S, Michael SF, Lorenzana I, Yozwiak NL, MacInnis BL, Bosch I, Gehrke L, Zhang F, Sabeti PC. (2018). Field-deployable viral diagnostics using CRISPR-Cas13. *Science* 360(6387):444-448. doi: 10.1126/science.aas8836
15. East-Seletsky A, O'Connell MR, Burstein D, Knott GJ, Doudna JA. (2017). RNA Targeting by Functionally Orthogonal Type VI-A CRISPR-Cas Enzymes. *Mol Cell* 66(3):373-383.e3. doi: 10.1016/j.molcel.2017.04.008
16. Knott GJ, East-Seletsky A, Cofsky JC, Holton JM, Charles E, O'Connell MR, Doudna JA. (2017). Guide-bound structures of an RNA-targeting A-cleaving CRISPR-Cas13a enzyme. *Nat Struct Mol Biol* 24(10):825-833. doi: 10.1038/nsmb.3466
17. Zhang JX, Xu JH, Yuan B, Wang XD, Mao XH, Wang JL, Zhang XL, Yuan Y. (2023). Detection of *Burkholderia pseudomallei* with CRISPR-Cas12a based on specific sequence tags. *Front Public Health* 11: 1153352. doi: 10.3389/fpubh.2023.1153352
18. Wongpalee SP, Thananchai H, Chewapreecha C, Roslund HB, Chomkatekaew C, Tananupak W, Boonklang P, Pakdeerat S, Seng R, Chantratita N, Takarn P, Khamnoi P. (2022). Highly specific and sensitive detection of *Burkholderia pseudomallei* genomic DNA by CRISPR-Cas12a. *PLoS Negl Trop Dis* 16(8): e0010659. doi: 10.1371/journal.pntd.0010659
19. Saravu K, Mukhopadhyay C, Vishwanath S, Valsalan R, Docherla M, Vandana KE, Shastry BA, Bairy I, Rao SP. (2010). Melioidosis in southern India: epidemiological and clinical profile. *Southeast Asian J Trop Med Public Health* 41(2):401-409. doi: 10.1097/OLQ.0b013e3181bf542c

20. K Howard and T J J Inglis. (2003). Novel selective medium for isolation of *Burkholderia pseudomallei*. *J Clin Microbiol* 41(7):3312-3316. doi: 10.1128/JCM.41.7.3312-3316.2003
21. Wiersinga WJ, Virk HS, Torres AG, Currie BJ, Peacock SJ, Dance DAB, Limmathurotsakul D. (2018). Melioidosis. *Nature Reviews Disease Primers* 4: 17107. doi:10.1038/nrdp.2017.107
22. Zheng H, Qin J, Chen H, Hu H, Zhang X, Yang C, Wu Y, Li Y, Li S, Kuang H, Zhou H, Shen D, Song K, Song Y, Zhao T, Yang R, Tan Y, Cui Y. (2021). Genetic diversity and transmission patterns of *Burkholderia pseudomallei* on Hainan island, China, revealed by a population genomics analysis. *Microb Genom* 7(11):000659. doi: 10.1099/mgen.0.000659
23. Gee JE, Gulvik CA, Castelo-Branco DSCM, Sidrim JJC, Rocha MFG, Cordeiro RA, Brilhante RSN, Bandeira TJPG, Patrício I, Alencar LP, da Costa Ribeiro AK, Sheth M, Deka MA, Hoffmaster AR, Rolim D. (2021). Genomic Diversity of *Burkholderia pseudomallei* in Ceara, Brazil. *mSphere* 6(1): e01259-20. doi: 10.1128/mSphere.01259-20
24. Zulkefli NJ, Teh CSJ, Mariappan V, Ngoi ST, Vadivelu J, Ponnampalavanar S, Chai LC, Chong CW, Yap IKS, Vellasamy KM. (2021). Genomic comparison and phenotypic profiling of small colony variants of *Burkholderia pseudomallei*. *PLoS One* 16(12): e0261382. doi: 10.1371/journal.pone.0261382
25. Ghazali AK, Eng SA, Khoo JS, Teoh S, Hoh CC, Nathan S. (2021). Whole-genome comparative analysis of Malaysian *Burkholderia pseudomallei* clinical isolates. *Microb Genom* 7(2):000527. doi: 10.1099/mgen.0.000527
26. Fang Y, Hu Z, Chen H, Gu J, Hu H, Qu L, Mao X. (2018). Multilocus sequencing-based evolutionary analysis of 52 strains of *Burkholderia pseudomallei* in Hainan, China. *Epidemiol Infect* 147: e22. doi: 10.1017/S0950268818002741
27. Kang X, Fu Z, Rajaofera MJN, Li C, Zhang N, Liu L, Sun Q, Chen C, Dong S, Xiu H, Ou X, Liu C, Pei H, He N, Xia Q. (2019). Whole-Genome Sequence of *Burkholderia pseudomallei* Strain HNBPO01, Isolated from a Melioidosis Patient in Hainan, China. *Microbiol Resour Announc* 8(36): e00471-19. doi: 10.1128/MRA.00471-19
28. Johnson SL, Bishop-Lilly KA, Ladner JT, Daligault HE, Davenport KW, Jaissle J, Frey KG, Koroleva GI, Bruce DC, Coyne SR, Broomall SM, Li PE, Teshima H, Gibbons HS, Palacios GF, Rosenzweig CN, Redden CL, Xu Y, Minogue TD, Chain PS. (2015). Complete genome sequences for 59 *Burkholderia* isolates, both pathogenic and near neighbor. *Genome Announc* 3(2): e00159-15. doi: 10.1128/genomeA.00159-15
29. Nicole M Bzdyl, Clare L Moran, Justine Bendo, Mitali Sarkar-Tyson. (2022). Pathogenicity and virulence of *Burkholderia pseudomallei*. *Virulence* 13, 1945-1965. doi: 10.1080/21505594.2022.2139063
30. Nimnuan-Ngam S, Hii SYF, Seng R, Saiprom N, Tandhavanant S, West TE, Chantratita N. (2025). Identification of novel *amrR* deletions as meropenem resistance mechanisms in clinical *Burkholderia pseudomallei* isolates. *Microbiol Spectr* 13(5): e0193624. doi: 10.1128/spectrum.01936-24.

31. Chirakul S, Norris MH, Pagdepanichkit S, Somprasong N, Randall LB, Shirley JF, Borlee BR, Lomovskaya O, Tuanyok A, Schweizer HP. (2018). Transcriptional and post-transcriptional regulation of PenA β -lactamase in acquired *Burkholderia pseudomallei* β -lactam resistance. *Sci Rep* 8(1):10652. doi: 10.1038/s41598-018-28843-7
32. Shafiq M, Ke B, Li X, Zeng M, Yuan Y, He D, Deng X, Jiao X. (2022). Genomic diversity of resistant and virulent factors of *Burkholderia pseudomallei* clinical strains recovered from Guangdong using whole genome sequencing. *Front Microbiol* 13:980525. doi: 10.3389/fmicb.2022.980525
33. Oslan SNH, Yusoff AH, Mazlan M, Lim SJ, Khoo JJ, Oslan SN, Ismail A. (2022). Comprehensive approaches for the detection of *Burkholderia pseudomallei* and diagnosis of melioidosis in human and environmental samples. *Microb Pathog* 169:105637. doi: 10.1016/j.micpath.2022.105637

Table 1 Sequence of RPA primers

Name	Sequence (5'-3')
Forward primer	F1: TAATACGACTCACTATAGGGGCACGGCGGAGATTCTCGAATTGTC
	F2: TAATACGACTCACTATAGGGGCACGCACGGCGGAGATTCTCGAATT
	F3: TAATACGACTCACTATAGGGGCCACGCACGGCGGAGATTCTCGAATTGTC
Reverse primer	R1: GCAACCACAGCAACGGAAAGAGCAGA
	R2: CCACAGCAACGGAAAGAGCAGATTG
	R3: GCAACCACAGCAACGGAAAGAGCAGATTGAAG

Note: Add T7 promoter sequence (TAATACGACTCACTATAGGG) to the 5 'end of the forward primer.

Table 2 Background data of 16 clinically isolated strains

Sample number	Gender	Age	Risk factor	Sampling time	Sample source	Sample type	Patients' clinical outcomes
Bp-1	F	64	Diabetes	2021.12	Wanning	blood	death
Bp-2	M	53	Diabetes	2022.02	Chengmai	blood	improve
Bp-3	F	49	-	2020.10	Chengmai	blood	death
Bp-4	M	30	Systemic Lupus Erythematosus	2021.08	Changjiang	blood	death
Bp-5	M	65	Diabetes	2022.03	Ledong	pus	improve
Bp-6	M	63	Diabetes	2020.11	Ledong	blood	improve
Bp-7	M	54	Diabetes	2021.11	Danzhou	blood	improve
Bp-8	F	24	-	2021.05	Haikou	pus	improve
Bp-9	M	49	-	2021.06	Haikou	blood	improve
Bp-10	M	30	Diabetes	2021.06	Haikou	pus	improve
Bp-11	M	54	-	2021.06	Dongfang	liquor pericardii	improve
Bp-12	M	42	-	2021.09	Haikou	drainage fluid	improve
Bp-13	M	39	Diabetes	2021.10	Haikou	blood	death
Bp-14	M	52	Tuberculosis and diabetes	2021.10	Chengmai	blood	improve
Bp-15	M	59	Diabetes	2022.01	Wanning	blood	death
Bp-16	M	36	Diabetes	2022.03	Haikou	blood	improve

Note: F means Female; M means Male.

Table 3 Functional classification of virulence factors in death group

Annotated gene	Carrier strain	Virulence factor	Function of genes	Strain number
katA	Bp-4 , Bp-13	KatA	Catalase	2
algU	Bp-4 , Bp-13	Alginate	Alginate	2
clbD	Bp-4 , Bp-13	Colibactin	Colicin	2
clbF	Bp-4 , Bp-13	Colibactin	Colicin	2
Cj1435c	Bp-1 , Bp-3	Capsule	Capsular phosphatase	2
ybtS	Bp-1 , Bp-3	Ybt	Yersinomycin	2
recN	Bp-1 , Bp-3	RecN	DNA repair protein	2
fbpCiron (III)	Bp-1 , Bp-3	FbpABC	Ferroportin	2
cdpA	Bp-15	CdpA	phosphodiesterase	1
cheA	Bp-15	Pse5Ac7Ac	Chemotactic histidine kinase	1
rscB	Bp-15	RcsAB	Transcriptional regulatory factor	1
mrkD	Bp-15	Type 3 fimbriae	Fimbriin type 3	1

Note: The death group contained 5 isolates, which were strains *Bp-1*, *Bp-3*, *Bp-4*, *Bp-13* and *Bp-15*.

Table 4 Functional classification of toxicity factors in the improve group

Annotated gene	Carrier strain	Virulence factor	Function of genes	Strain number
katA	Bp-5, Bp-6, Bp-8, Bp-9, Bp-11, Bp-12	KatA	Catalase	6
algU	Bp-5, Bp-6, Bp-8, Bp-9, Bp-11, Bp-12	Alginate	Alginates	6
clbD	Bp-5, Bp-6, Bp-8, Bp-9, Bp-11, Bp-12	Colibactin	Colicin	6
clbF	Bp-5, Bp-6, Bp-8, Bp-9, Bp-11, Bp-12	Colibactin	Colicin	6
Cj1435c	Bp-2, Bp-10, Bp-14	Capsule	Capsular phosphatase	3
ybtS	Bp-2, Bp-10, Bp-14	Ybt	Yersinomycin	3
recN	Bp-2, Bp-10, Bp-14	RecN	DNA repair protein	3
fbpCiron(III)	Bp-2, Bp-10, Bp-14	FbpABC	Ferroportin	3
cdpA	Bp-7	CdpA	Phosphodiesterase	1
cheA	Bp-7	Pse5Ac7Ac	Chemotactic histidine kinase	1
rcsB	Bp-7	RcsAB	Transcriptional regulatory factor	1
porB	Bp-16	Porin	Momp	1
allB	Bp-16	Allantion utilization	Allantoinase	1
allC	Bp-16	Allantion utilization	Allantoic acid amide hydrolase	1
pvdQ	Bp-16	pyoverdine	Serine lactone acylase	1
bprA	Bp-7	Bsa T3SS	Hns-like regulatory protein	1

Note: There were 11 isolates in the improve group, including strains *Bp-2, Bp-5, Bp-6, Bp-7, Bp-8, Bp-9, Bp-10, Bp-11, Bp-12, Bp-14* and *Bp-16*.

Table 5 Antimicrobial Susceptibility Testing results of 16 clinical isolates

Antibacterial agents	Interpretive standard MIC/($\mu\text{g}\cdot\text{mL}^{-1}$)			The data of this study		
	S	I	R	Number	S%	R%
SXT	$\leq 2/38$	-	$\geq 4/72$	16	100	0
IPM	≤ 4	8	≥ 16	16	100	0
CAZ	≤ 8	16	≥ 32	16	100	0

Note: SXT: Trimethoprim/sulfamethoxazole; IPM: Imipenem; CAZ: Ceftazidime.
MIC: Minimum Inhibitory Concentration; S: Susceptible; I: Intermediate; R: Resistance.

"-" indicates no data.

Table 6 Results of drug resistance gene annotation

Drug resistance gene	Corresponding antibiotic	Quantity (%)	Resistance mechanism
adeF	Tetracycline, fluoroquinolones	14 (87.5)	Antibiotic efflux
amrA	Aminoglycosides	10 (62.5)	Antibiotic efflux
amrB	Aminoglycosides	9 (56.3)	Antibiotic efflux
OXA-57	Penicillin, cephalosporins	2 (12.5)	Antibiotic inactivation
OXA-59	Penicillin, cephalosporins	4 (25.0)	Antibiotic inactivation
R39	Penicillins	9 (56.3)	Antibiotic inactivation
Omp38	Penicillins, cephalosporins, carbapenems	9 (56.3)	Reduce antibiotic permeability

FIG 1 Functional classification statistics of EggNOG functional genes on 2 chromosomes of *Bp-11* genome. The EggNOG function annotation information of protein coding genes CDSs was divided into 23 groups and distinguished by different colors. The horizontal coordinate represents each EggNOG classification category, and the vertical coordinate represents the number of corresponding functional genes. Different colors represent different EggNOG annotation functions, including the number and proportion of genes.

FIG 2 Functional annotation results of *Bp-11* genome. (I) Chromosome 1 gensphere map of *Bp-11*. (II) Chromosome 2 gensphere map of *Bp-11*. (III) Plasmid gensphere map of *Bp-11*. (IV) Different colors correspond to I-III, representing different annotation functions and the number of genes on the annotation for that function. In I-III, from the outside to the inside: the outermost black circle, marked with a scale indicating genome size (1 scale represents 5 kb); the second circle shows the genes on the positive chain of the genome; the third circle shows the genes on the negative chain of the genome, with the second and third circles colored differently to distinguish EggNOG functions. The blue-green section of the fourth circle represents repeating sequences; in the fifth circle, blue indicates tRNA and purple indicates rRNA. The sixth circle shows GC content, with light yellow regions indicating higher GC content than the average genome GC content, and the higher the peak, the greater the difference. Blue regions indicate GC content lower than the average genome GC content. The seventh circle represents GC-skew, where the dark gray area indicates regions where the content of G is greater than C, and the red area indicates regions where C is greater than G. The innermost circle identifies the size (bp) of the gensphere map.

FIG 3 Gene family clustering of 16 isolates and K96243. Overlapping regions represent shared gene families between species, while non-overlapping regions indicate gene families specific to each species. Specific gene families (functions) were identified as follows: Bp-8 had 1 specific gene family (Enoyl- Acyl carrier protein reductase), Bp-11 had 2 (Integrase core domain; Transposase), and K96243 had 8 (Immunoglobulin I-set domain; Resolvase, N terminal domain; H-NS histone family; Helix-turn-helix domain; Phage P2 GpE; LysR substrate binding domain; Phage integrase family; Adenylylsulphate kinase).

FIG 4 Phylogenetic tree. The phylogenetic tree includes root, nodes, branches, and bootstrap values, which reflect the evolutionary relationships and support levels for the branches.

FIG 5 Genome collinearity between 16 sequenced strains and K96243. Different colors represent different homologous genes. The isolates are linked linearly to the homologous genes of K96243, showing the collinearity between the genomes.

FIG 6 Screening of RPA primers and construction of SHERLOCK fluorescence detection system. (A) Electrophoretic analysis of RPA products using different primer pairs. (B) Relative luminance of the RPA amplified target strips shown in panel A. (C) Comparison of fluorescence detection efficiency among three pairs of RPA primers using the SHERLOCK fluorescence detection method.

FIG 7 Sensitivity and specificity analysis of SHERLOCK detection system. (A) SHERLOCK fluorescence detection results for strains *Bp* 1-21. (B) SHERLOCK fluorescence detection results for strains *Bp* 22-42. (C) SHERLOCK fluorescence detection results for control group. (D) Fluorescence values detected by different strains after 20 minutes of reaction (Unpaired t-test, **** ($p < 0.0001$)). (E) SHERLOCK strip test results for strains *Bp* 1-42. (F) SHERLOCK strip test results for the control group.

Note: Bc: *Burkholderia cepacia*; Pae: *Pseudomonas aeruginosa*; Eco: *Escherichia coli*; Sau: *Staphylococcus aureus*; Kpn: *Klebsiella pneumoniae*; Spn: *Streptococcus pneumoniae*. NC: Negative control; PC: Positive control.

FIG 8 Limit of Detection of the SHERLOCK Fluorescence Detection System. (A) With the dilution of *Bp* nucleic acid concentration, the real-time fluorescence value gradually decreased, which was related to the decrease of detection sensitivity. (B) The lowest detection limit is 10^{-4} ng/μL (Paired *t*-test, $** (p=0.0041)$), at this detection limit, a rapid increase in fluorescence values can be detected within 20 minutes of the start of the reaction.

Table7 Limit of Detection of the SHERLOCK Strip Test Method

Concentration(ng/uL)	10^{-1}	10^{-2}	10^{-3}	10^{-4}	10^{-5}	10^{-6}	10^{-7}	Negative control
Detection times	3	3	3	3	3	3	3	3
Number of detected	3	3	3	0	0	0	0	0
Detection rate (%)	100	100	100	0	0	0	0	0

Dear Editor and Reviewers,

Thank you for offering us an opportunity to improve the quality of our submitted manuscript "Whole Genome Study and Construction of SHERLOCK Detection Method for Endemic Strains of Melioidosis in Hainan based on Third-generation Sequencing"(Spectrum00592-25). We appreciated very much the reviewers' constructive and insightful comments. In this revision, we have addressed all of these comments/suggestions. We hope the revised manuscript has now met the publication standard of your journal.

We have highlighted all the revisions in red

On the next pages, point-to-point responses to the queries raised by the reviewers are listed.

Reviewer #1

Abstract

Comment 1: Results: Please indicate the value of "good sensitivity and specificity".

Response: Thanks for your advice. Sensitivity and specificity values have been added at the end of the sentence. "both are 100%".

Methods

Comment 1: There are different number of isolates used in various applications. Please state the numbers clearly in the methods section.

Response: We explained the number of isolates used in different subsequent experiments and marked them red in the method section. First, 16 strains of *Burkholderia pseudomariae* were randomly selected from 42 strains of *Burkholderia pseudoeriae* for full genome sequencing. In the later experiment, 42 strains of *Burkholderia pseudomariae* and 11 strains of the control group were used for sensitivity and specificity analysis of the SHERLOCK detection system.

Comment 2: Please rephrase the sentence "All samples were obtained from the remaining samples after clinical testing".

Response: Rephrase the sentence as "A total of 53 clinical isolates were collected from human disease".

Comment 3: The authors should include the manufacturer of the equipment/reagents used in this study.

Response: Thanks for your advice. We supplemented the manufacturer of the equipment/reagent used with the full text.

Comment 4: Is it broth microdilution method used to detect MIC? Which CLSI version is used to interpret antibiotic susceptibility for BP? Please include citation.

Response: Susceptibility tests were performed according to the broth microdilution method recommended by CLSI (CLSI M100-Ed33 Performance Standards for Antimicrobial Susceptibility Testing). We also supplemented this in the method section.

Comment 5: What does "The single-copy gene sequences" referring to? Are the authors comparing whole genome/core/accessory genes?

Response: It's referring to "single-copy homologous gene", the single-copy homologous gene sequence of the sequenced strain was compared with the single-copy homologous gene sequence of the model strain K96243, and the evolutionary tree was constructed by the maximum likelihood method using PhyML software, and then the evolutionary relationship between the strains was analyzed. Thanks to Reviewer for his reminder, we have improved the expression.

Comment 6: Please give full name of "RPA system" on first use.

Response: Thanks for your advice. The full name of "RPA" is "Recombinase Polymerase Amplification", it has already been introduced in the "Introduction" section.

Comment 7: The authors need to specify the conditions of the amplification to ensure the results are reproducible.

Response: The amplification conditions are as follows: "RPA amplification system (50µL) included Primer Free Rehydration buffer (29.5 µL), DNase-Free water (11.2 µL), Forward primer (2.4 µL, 10 µM), Reverse primer (2.4 µL, 10 µM), template DNA (*Bp* DNA, 2 µL), Magnesium Acetate (MgOAC, 2 µL, 280 mM). After adding MgOAC, the components were thoroughly mixed and subjected to brief centrifugation. The reaction mixture was promptly incubated at 39°C for 20 minutes."

Comment 8: The authors may consider to create a table for easier identification of all

the oligos used for each application. Please cite in the text for the table/figure.

Response: Thanks for your advice. We have summarized all oligos in Table 1 and cited in the text.

Comment 9: Please clarify the interpretation of positive result. Is the result valid" if only the T line showed red bands and the Cline was colorless"?

Response: Result interpretations were as follows: (1) A positive result was indicated if both the test line (T line) and C line showed red bands, or if only the T line showed red bands and the C line was colorless; (2) A negative result was indicated if only the C line showed a red band and no red band was visible on the T line.

Comment 10: What does the term "clinically positive" referring to?

Response: "clinically positive" refers to a previously collected clinical isolate of *Bp*, which may not be clear enough and has been deleted.

Comment 11: Please define the cases. For examples, true positive, false negative, etc.

Response: Thanks for your advice. The previous sensitivity and specificity formulas are not suitable and have been changed to the following formulas:

Sensitivity = True Positives / (True Positives + False Negatives) × 100%

Specificity = True Negatives / (True Negatives + False Positives) × 100%

Comment 12: Is the term "double dilution" correct?

Response: Thanks for your advice. It has been modified to "equal proportion dilution".

Results:

Comment 1: In section 3.2, The authors analyse only *BP-11*, how about the other 15 isolates? Please explain. These large data may be included as supplementary material.

Response: Sequencing results showed that the genomes of 16 strains were all

composed of 2 chromosomes, and there were similarities in structure, but only BP-11 was found to have genetic structure-plasmid except 2 chromosomes in the whole genome, so BP-11 was used as a representative to display the egg NOG function annotation results of *Bp*. We've added content and optimized image presentation.

Comment 2: Please clearly label in Figure 2 the location of I, II and III.

Response: Thanks for your advice. Figure 2 has been labeled.

Comment 3: Did the authors check on the plasmid sequence? It will be interesting to know if there's similarity with other BP or organism, possible acquired?

Response: Only BP-11 has plasmid sequence, so we carried out preliminary analysis of plasmid and found 1,331 genes without eggNOG annotation. We suspect that these genes may be mainly related to DNA replication, recombination and repair, but further research is needed to support this claim.

Comment 4: It will be easier to compare if the information in Table 3 and 4 to be incorporated in Table 2.

Response: Thanks for your advice. Table 2 presents background information of 16 isolates, and Tables 3 and 4 show information such as functional classification of virulence factors in groups. We tried to combine all information together, but compared with the previous data, it seems that the data are more chaotic, so we still hope to divide them into 3 tables for description of results. I hope to get the approval of the review experts, thank you.

Comment 5: The authors should include the footnote and define the abbreviations used in the tables/figures.

Response: Thanks for your advice. The relevant content has been supplemented.

Comment 6: The authors may remove the word “clinical” and change to antibiotic susceptibility testing. AST is an in vitro method therefore it will be confusing to use

"clinical" term.

Response: Thanks for your advice. We've removed the word "clinical" and change to "Antimicrobial Susceptibility Testing".

Comment 7: The authors are suggested to check the presence of these genes from the whole genome analysis. Majority of the genes listed are chromosomally encoded in BP genomes. It will be good to discuss on the absence of the genes in typical isolates.

Response: Thanks for your advice. This study summarizes the basic characteristics of the genome of the Hainan epidemic strain of melioidosis and annotates the function of the genome. However, our ultimate goal is to find conserved sequences in the genomic sequence. According to sequencing results and literature reports, TS1, as a part of the Bp secretion system that plays a key pathogenic role, exists in all Bp and is highly conserved, so it was selected for follow-up research.

Comment 8: In Table 6, the term "carrier rate" is confusing. The authors may use "number of isolates" instead.

Response: Thanks for your advice. We have modified the terms in the table.

Comment 9: What are the specific gene families shown from the analysis?

Response: Specific gene family is one that is unique to the species and does not exist in any of the other organisms analyzed. In Figure 3, overlapping regions represent shared gene families between species, while non-overlapping regions indicate gene families specific to each species.

Comment 10: From the phylogenetic tree, it looks like the BP1,3 and 5 (death outcome) are diverged from the others. It will be good to include this in discussion.

Response: Thanks for your advice. We tried to analyze in the discussion section. For example, BP1,3,4 and 5 were closely related, but BP 1, 3 and 4 appeared in the death group, while BP 5 only appeared in the improvement group. This may indicate that although the genetic relationship is close, the level of virulence may be different,

which enhances the complexity of clinical manifestations after BP infection.

Comment 11: Please recheck on the sentence “Conversely, ORF2, located ...”

Response: Thanks for your advice. We've removed the word “Conversely” and change to “In addition, ORF2, located within the coding region of the TTS1 gene cluster, was specific to *Bp* and absent in closely related *Burkholderia species* such as *Burkholderia thailandensis*, and *Burkholderia cepacia*. Consequently, the TTS1-ORF2 genome sequence was chosen as the target fragment for RPA (GenBank: AF074878)”.

Comment 12: Did the authors perform the LOD limit on strip tests?

Response: Thanks for your advice. Limit of detection of the SHERLOCK strip test method has been supplemented.

Discussion:

Comment 1: The authors may improve on the discussion.

Response: Thanks for your advice. The discussions have been revised and supplemented accordingly.

Reviewer #2

Comment 1: Comparative Genomics and Epidemiology

While the study asserts that *B. pseudomallei* strains in Hainan have unique epidemiological characteristics, it would be more compelling to provide concrete evidence through comparative genome structural analysis. I recommend including a comparative analysis of genome structures between Hainan isolates and publicly available global isolates to substantiate the claim of distinct epidemiological traits.

Response: According to the whole genome structure of 16 strains of *BP* isolated from this third-generation sequencing and the genome structure of international standard strain K96243, it can be seen that the binary genome structure is the structural characteristics of the genetic material of *BP*. It is worth noting that a new genome structure except two chromosomes was found in the whole genome of BP-11. Sequencing analysis showed that the genome structure was a closed circular double-stranded DNA-plasmid with a length of 217,237bp. Before this, no plasmid was found in the genome structure of *BP*. Because the genome sequence of a large number of *BP* samples was completed by second generation sequencing technology, one of the disadvantages of second-generation sequencing technology is that it is difficult to distinguish plasmid and nuclear genome. Therefore, it is necessary to obtain sufficient genome sequencing results by three generation sequencing technology to verify the credibility of plasmid structure obtained by *BP* sequencing.

Comment 2: Petal Diagram Interpretation

In Figure 3, the petal diagram depicts core and unique gene families among the 16 isolates and the reference strain K96243. Please provide detailed information about the eight unique gene families of K96243 and the two unique gene families of BP-11. This information would enhance the understanding of gene content diversity and specificity.

Response: Thanks for your advice. We will display specific information such as the

speculated functions as follows, but the results need further verification.

Strain	Number of specific gene families	Function
BP-8	1	Enoyl- (Acyl carrier protein) reductase
BP-11	2	Integrase core domain ; Transposase
K96243	8	Immunoglobulin I-set domain ; Resolvase, N terminal domain ; H-NS histone family ; Helix-turn-helix domain ; Phage P2 GpE ; LysR substrate binding domain ; Phage integrase family ; Adenylylsulphate kinase

Comment 3: Diagnostic Specificity and Related Species

The SHERLOCK detection system targeting the TTS1-ORF2 gene is promising. However, assay specificity could be further validated by including closely related *Burkholderia* species such as *B. thailandensis*, *B. mallei*, and others. This would strengthen confidence in its clinical applicability and help rule out cross-reactivity.

Response: According to sequencing results and literature reports, TS1, as a part of the *Bp* secretion system that plays a key pathogenic role, exists in all *Bp* and is highly conserved. The ORF2 in the coding region of the TS1 gene cluster is limited to *Bp* and does not exist in other closely related *Burkholderia* species such as *Burkholderia thailai*, *Burkholderia cepacia*, *Burkholderia mallei*, etc., thus ruling out the possibility of cross-reactions in subsequent experiments.

Comment 4: Calculation of Sensitivity in Methods (Section 2.7.1)

The sensitivity formula used in Section 2.7.1 is incorrect. The correct formula should be:

Sensitivity = True Positives / (True Positives + False Negatives) × 100% Please

revise this accordingly.

Response: Thanks for your advice. The previous sensitivity and specificity formulas are not suitable and have been changed to the following formulas:

Sensitivity = True Positives / (True Positives + False Negatives) × 100%

Specificity = True Negatives / (True Negatives + False Positives) × 100%

Comment 5: Clarity of Terms in Discussion

In the Discussion section (third paragraph), the phrases "dead and improved strains" and "dead group" are confusing and potentially misleading. Please clarify that these and similar words throughout the manuscript refer to patients' clinical outcomes, not the bacterial strains themselves.

Response: Thanks for your advice. This refers to grouping patients according to their clinical outcomes, divided into death group and improve group, rather than the strain itself, which we have revised in manuscript.

Comment 6: Antibiotic Resistance Gene Analysis

The analysis of resistance genes is appreciated; however, it overlooks several key resistance determinants previously reported in *B. pseudomallei*. For example:

The *PenA* gene, known to confer resistance to ceftazidime and amoxicillin-clavulanate, should be discussed. See Hii SYF et al., *Antimicrob Agents Chemother*, 2021.

Recent evidence shows that *amrR* deletions contribute to carbapenem resistance. Refer to Nimnuan-ngam et al., 2025. Please investigate and discuss the presence or absence of these genes in your isolates.

Response: Thanks for your advice. We read the literature mentioned by the reviewers and learned that mutations or over-expression of the *PenA* gene can lead to CAZ resistance. In addition, the RND operon, encoding an inner membrane protein (*amrA*) and RND transporter (*amrB*), works with an OMP (*oprA*) to form a tripartite complex

under the regulation of *amrR*. These proteins work together to actively pump a variety of antimicrobial compounds out of the bacterial cell into the external environment. The specific mutations within the *amrR* gene contribute to the upregulation of *amrAB-oprA* efflux pump transcriptional levels, which are linked to MEM resistance. However, our research did not find changes in these genes, such as mutations or deletions. This may be related to the small number of strains tested in our experiments. We very much agree with the reviewers' opinions and mentioned relevant content during the discussion.

Comment 7: Unreferenced Claims in Introduction and Discussion

Several claims throughout the Introduction and Discussion lack appropriate citations. For instance, the claim that the *R39* gene "might produce beta-lactamases" should be supported by references or clarified as speculative. Please ensure that all assertions are backed by appropriate literature.

Response: Thanks for your advice. We reviewed the introduction and discussion sections and added some citations as needed. In addition, there is currently a lack of relevant research reports on the *R39* gene. We speculate based on the functional annotation results that it may produce beta lactamases, which is related to resistance to penicillin antibiotics. Of course, this conclusion needs further research and verification.

Comment 8: Table 2 contains patients' information. These require ethical approval.

Minor Suggestions:

- Please check for consistent gene naming (e.g., use italics for gene names such as *amrR*, *penA*, *katA* etc.).
- A visual summary or schematic of the SHERLOCK workflow could enhance readability for a broader audience.

Response: For some sample information shown in Table 2, we have conducted ethical

approval, and do not include patient details, which does not have any impact on patient interests. In addition, we have checked the naming and format of genes, and we have also added to SHERLOCK workflow, hoping to be accepted by reviewers.

Reviewer #3

General Comments

Comment 1: The title may be misleading, as it implies the SHERLOCK assay is specific to Hainan isolates. Consider clarifying whether the assay targets region-specific signatures or general Bp markers.

Response: Thanks for your advice. Our study was originally aimed at epidemic strains in Hainan, and SHERLOCK method was designed to detect them according to genome sequencing results.

Comment 2: The manuscript should include line numbers to facilitate reviewer feedback.

Response: Thanks for your advice. We will ask the editorial board for advice on the basic requirements for manuscript format, whether to add line numbers to the manuscript.

Comment 3: There are numerous typographical and formatting inconsistencies (e.g., “Eggnog,” “EggNOG,” and “eggnog” appear interchangeably). Standardize terminology throughout.

Response: Thanks for your advice. We checked the manuscript and corrected these incorrect formatting.

Comment 4: Figure legends lack sufficient detail. Each should be self-contained—describing the figure's content, purpose, and analytical methods.

Response: Thanks for your advice. We re-edited the annotation of each figure and described it in detail according to the serial number.

Comment 5: Several figures contain small or non-English labels and low-resolution

images. Highquality, English-labeled vector graphics should be submitted.

Response: Thanks for your advice. We have submitted higher resolution images and edited them as requested by reviewers.

Comment 6: Many factual statements are unsupported by citations. All claims, especially comparative or quantitative ones, must be appropriately referenced.

Response: Following the reviewer's suggestion, we checked the manuscript and added citations have been highlighted in red.

Abstract

Comment 1: The statement that detection is achieved in “1–2 hours” should be substantiated in the Results section, with clear experimental data or protocol timing.

Response : Thanks for your advice. Regarding the testing time, we have added a description to the results, and the details are as follows: “In addition, the time for nucleic acid extraction (according to the manufacturer's instructions, it takes about 20 minutes) and RPA amplification (it takes about 20 minutes) is about 40 minutes. The final SHERLOCK fluorescence test (fluorescence values were read after 20 minutes) took a total of 60 minutes. The SHERLOCK test strip method requires incubation at 37 degrees Celsius for 30 minutes before testing, it takes about 70 minutes in total. Therefore, both methods can achieve rapid detection of BP within 1-2 hours.”

Introduction

Comment 1: The review of genomic studies is incomplete and omits important contributions from others such as Australian and Thai research groups, for example.

Response: Thanks for your advice. We have supplemented relevant content. “The world’s first strain of *Bp* (No. K96243) to undergo full genome sequencing was a patient from Thailand. In 2004, researchers at Sanger Laboratory in the United Kingdom completed its sequencing, assembly and functional annotation. K96243 was

used as an international standard strain and its genomic information was uploaded to NCBI in the United States to provide a reference for subsequent strain genomic research. In addition, in a genomic sequencing study of *Bp* from multiple countries, including 30 countries in Oceania, Asia, Africa, Central and South America, the results showed that there were significant genetic differences between isolates from Australia and Asia, and that isolates from Australia had longer phylogenetic branches. Combined with pan-genome analysis data, the final results show that Australia is an early reservoir for the current global population of *Bp* and is spreading to other regions.”

Comment 2: Similarly, the CRISPR diagnostics overview does not cite prior *Bp*-specific CRISPR assays developed by other groups, including those from China and Thailand. Acknowledging existing work will help situate the novelty of this study.

Response: Thanks for your advice. We supplemented relevant research from China and Thailand and presented the content as follows: “Recently, research teams from China and Thailand have also reported using CRISPR-Cas12a technology to achieve rapid detection of *BP*. However, the CRISPR RNA (crRNA) sequences used in the research are different, resulting in certain differences in the range of BP populations detected. Therefore, based on genomic sequence analysis of endemic melioidosis strains in Hainan, this study was aimed to design crRNA targeting specific and conserved gene sequences in the melioidosis genome, construct a SHERLOCK reaction system, and establish two technical platforms—fluorescence method and lateral chromatography test strip—to verify the method's accuracy and reliability for the rapid diagnosis of endemic melioidosis strains in Hainan.”

Methods

Comment 1: The Methods section lacks sufficient detail for replication. Information such as reagent sources, catalog numbers, and software versions should be included.

If space is a concern, detailed protocols can be placed in Supplementary Materials.

Response: Thanks for your advice. Information such as reagent source has been supplemented.

Comment 2: Culture conditions for *B. pseudomallei* and other bacteria should be clearly described, including media and incubation parameters.

Response: Thanks for your advice. In the method section, it is introduced that the collected strain samples are directly stored at -80°C and then used for subsequent experiments, without involving the cultivation of strains. Therefore, the conditions for culture are not described.

Comment 3: It is unclear whether clinical isolates or direct clinical specimens were used in the SHERLOCK assay. This distinction must be clarified in both the Methods and Results.

Response: Clinical isolates were used, as described in the methods.

Comment 4: The limit of detection (LOD) experiment should include the number of replicates performed. A minimum of three independent replicates per data point is generally expected.

Response: As the reviewer said, we also wrote in the method, “Each concentration was tested in triplicate to observe and record the fluorescence curve changes and detection lines.”

Results

Comment 1: Several analyses (e.g., CRISPR loci, functional annotations) are presented abruptly. Reorganize related results under thematic subheadings and provide better contextual transitions.

Response: Thanks for your advice. The results have been reorganized and provided

for better contextual transitions, with modifications highlighted with red marks.

Comment 2: Justify the choice of isolate BP-11 for detailed analysis.

Response: Sequencing results showed that the genomes of 16 strains were all composed of 2 chromosomes, and there were similarities in structure, but only BP-11 was found to have genetic structure-plasmid except 2 chromosomes in the whole genome, so *BP-11* was used as a representative to display the egg NOG function annotation results of *Bp*. We've added content and optimized image presentation.

Comment 3: Figures labeled “Figure 2-I,” “-II,” and “-III” are not clearly marked. Label all panels explicitly.

Response: Thanks for your advice. Figure 2 has been labeled.

Comment 4: Figure 3 appears simplistic and lacks scientific value. Reconsider whether it adds to the narrative or replace it with more informative content.

Response: Thanks for your advice. The petal diagram is mainly used to show the results of gene family clustering analysis between 16 isolates and the reference strain K96243.

Comment 5: A whole-genome phylogenetic analysis including all available *Bp* strains would strengthen the manuscript by placing Hainan isolates in global context.

Response: We randomly selected 16 strains from a total of 42 *Bp* strains for WGS. The results revealed that the genomes of these 16 strains each consisted of two chromosomes, with a total genome size ranging from 7.1 Mb to 7.3 Mb. Chromosome 1 was larger, while Chromosome 2 was smaller. The sequencing results showed that the genomic structure of the epidemic melioidosis strains in Hainan was similar. In subsequent experiments, conserved sequences were obtained from them to prepare for the establishment of the SHERLOCK system. Therefore, no in-depth analysis was

conducted in aspects such as genome-wide phylogenetic analysis, because this may require more samples and more financial investment.

Comment 6: Genome collinearity should be quantitatively described and discussed.

Response: Thanks for your advice. Collinearity analysis was used to describe the position and orientation of homologous genes. The alignment sequence and position of homologous genes were similar, so collinearity regions could be formed. Due to the limitation of methods, quantitative analysis could not be carried out.

Comment 7: The transition from genome analysis to SHERLOCK diagnostics lacks coherence. Currently, the CRISPR assay section seems disconnected. Consider integrating the genomic data to support assay design.

Response: Thanks for your advice. We made adjustments to the transition from genomic analysis to SHERLOCK diagnosis, and provided for better contextual transitions, with modifications highlighted with red marks.

Comment 8: The panel of non-Bp organisms (n=6) used to assess assay specificity is limited. Consider expanding the panel to include more closely related species or increase numbers of isolates

Response: *Burkholderia cepacia*, *Klebsiella pneumoniae*, *Pseudomonas aeruginosa*, *Escherichia coli*, *Staphylococcus aureus*, *Streptococcus pneumoniae* and other clinically common bacteria were set as control strains. The main purpose is to detect the specific differences between *Burkholderia pseudomallei* and other common clinical strains.

Comment 9: Statistical analysis is generally lacking across experiments. Include appropriate statistical tests to support claims.

Response: Thanks for your advice. Appropriate statistical analysis will be supplemented where necessary.

Comment 10: While VFDB and CARD databases are useful, discuss potential limitations and whether cross-validation with other tools was considered.

Response: Thanks for your advice. We described the pros and cons of using VFDB and CARD databases.

Comment 11: All sequencing data must be deposited in a public database prior to publication.

Response: Thanks for your advice. Sequencing data have been stored in the NCBI SRA database under the following number: PRJNA1280284.

Discussion

Comment 1: The Discussion is relatively stronger than other sections, providing interpretation and context. However, some of this information (e.g., rationale and implications) should be incorporated earlier into the Results section to help guide readers.

Response: Thanks for your advice. The content has been re-adjusted to make it more convenient for readers to read, with modifications highlighted with red marks.

Comment 2: A comparative discussion of the current SHERLOCK assay versus existing CRISPR diagnostics for Bp would be valuable for highlighting novelty and advantages.

Response: Thanks for your advice. Based on the timing of our research results, we found previous studies on Thailand and other places in China and compared them. We included the relevant content in the discussion section.

Conclusion

Comment 1: The phrase “we pioneered the use of the CRISPR/Cas” is not accurate

given the existence of prior studies using CRISPR diagnostics for Bp. Consider rephrasing to reflect the study's specific contributions (e.g., adaptation to local strains or integration with third-generation sequencing data).

Response: Thanks for your advice. Rephrasing this sentence for: “Additionally, we pioneered the use of the third-generation of sequencing results combined with the CRISPR/Cas system for detecting Bp...”

Dear Editor and Reviewers,

Thank you for offering us an opportunity to improve the quality of our submitted manuscript "Whole Genome Study and Construction of SHERLOCK Detection Method for Endemic Strains of Melioidosis in Hainan based on Third-generation Sequencing"(Spectrum00592-25). We appreciated very much the reviewers' constructive and insightful comments. In this revision, we have addressed all of these comments/suggestions. We hope the revised manuscript has now met the publication standard of your journal.

We have highlighted all the revisions in red

On the next pages, point-to-point responses to the queries raised by the reviewers are listed.

Reviewer #1

Comment 1: Public repository details (Required):

The authors need to specify the accession numbers for all the isolates used in this study.

Response: The accession numbers of the 16 strains were as follows:

SAMN49520277, SAMN49520278, SAMN49520279, SAMN49520280,
SAMN49520281, SAMN49520282, SAMN49520283, SAMN49520284,
SAMN49520285, SAMN49520286, SAMN49520287, SAMN49520288,
SAMN49520289, SAMN49520290, SAMN49520291, SAMN49520292.

Specific information is included in supplementary materials.

Introduction

Comment 1: The authors need to use the correct term. It is very confusing. Melioidosis refers to the disease, *B. pseudomallei* is the bacteria causing melioidosis. Please do the necessary changes.

Response: Thanks for your advice. Relevant terms have been reviewed and revised.

Methods

Comment 1: Please use just one term: isolate or strain. It is misleading.

Response: Thanks for your advice. After modification, retain one term: strain.

Comment 2: Line 144 The use of "human disease" is not appropriate. Instead, the authors may include the isolation site/sample type (Table 2).

Response: Thanks for your advice. Removed "human disease", and adjusted the sentence.

Comment 3: Line 218 Antibiotic susceptibility testing should be sectioned separately. Please refer to CLSI M45, 2016 for testing and interpretation.

Response: Thanks for your advice. We have highlighted all the revisions in red.

Comment 4: Line 298: A qPCR machine is required for fluorescence detection. Is it only one cycle at 37°C used? The authors need to specify the time required for the incubation and is ct value a factor to consider for this test?

Response: Yes, the qPCR reaction temperature is set to 37°C and maintained. Then, the fluorescence intensity is measured every minute to monitor the change in the fluorescence signal. We use the change in fluorescence value as a judgment factor for the qualitative result, and then consider the ct value and other factors after accumulating data. In addition, at 20 minutes of reaction, the difference in fluorescence signal values between the experimental group strain and the control group strain was very obvious, and the explanation of the results has been introduced in the article.

Comment 5: Line 315: Is it correct that the result is still valid when the C line does not show colour. What does the C line incorporated with? Please ensure correct interpretation criteria is met.

Response: Thanks for your advice. Line C is the control line, and we will revise the interpretation of the results: (1) A positive result was indicated if both the test line (T line) and C line showed red bands; (2) A negative result was indicated if only the C line showed a red band and no red band was visible on the T line; (3) An invalid result was indicated if no red bands was visible on both the T line and the C line, or if only the T line showed red band and the C line was colorless.

Comment 6: The tested DNA were extracted from bacterial isolates or clinical samples? The sentence Line 326-327 is misleading.

Response: Thanks for your advice. The tested DNA were extracted from bacterial isolates. We have highlighted all the revisions in red.

Results

Comment 1: It will be easier to understand if the authors stated in Line 342 that 1 strain contains additional plasmid.

Response: Thanks for your advice. We added relevant content.

Comment 2: Did the authors observe the missing of group Y/Z in other BP genomes. It looks like BP11 has slightly smaller full genome size even with an additional plasmid. Does the presence of plasmid results in genes deletion in BP11?

Response: The missing of group Y/Z were also observed in other BP strains, and there was no obvious evidence that the presence of plasmids would lead to the deletion of the BP11 gene.

Comment 3: How many bootstrap replicates used? There is no distance value available in Figure 4. Is this phylogenetic tree compared shared 5192 genes clusters generated from 3.3.1?

Response: We use “single-copy homologous gene” to build Phylogenetic tree. The single-copy homologous gene sequence of the sequenced strain was compared with the single-copy homologous gene sequence of the model strain K96243, and the evolutionary tree was constructed by the maximum likelihood method using PhyML software, and then the evolutionary relationship between the strains was analyzed. In Figure 4, branches with bootstrap values greater than 75 were considered reliable.

Comment 4: The authors may need to elaborate on the phylogenetic analysis. It is better to view in a global perspective. Recommended to add more isolates including earlier Hainan's (as mentioned in introduction) and from other countries.

Response: Thanks for your advice. Several studies have been reported on the relationship between *BP* Hainan isolates and global isolates. For example, Zheng et al. pointed out that the phylogenetic analysis of global *BP* strains showed that the *BP* strain in Hainan may have originated from Southeast Asian countries and is closely related, especially the *BP* isolate in Thailand (Zheng H, Qin J, Chen H, Hu H, Zhang

X, Yang C, Wu Y, Li Y, Li S, Kuang H, Zhou H, Shen D, Song K, Song Y, Zhao T, Yang R, Tan Y, Cui Y. Genetic diversity and transmission patterns of *Burkholderia pseudomallei* on Hainan island, China, revealed by a population genomics analysis. *Microb Genom.* 2021 Nov;7(11):000659. doi: 10.1099/mgen.0.000659). Taking into account that there are already relevant reports, the main research route of this study is to first find conserved sequences through genomic research, and then build a SHERLOCK detection system to detect Hainan BP epidemic strains. Therefore, the phylogenetic analysis will not be discussed, hoping to get the support of reviewers.

Comment 5: Line 469: How many replicates were used for serial dilution testing?

Response: The number of repeated tests is introduced in methods section. “*Bp* DNA concentration was serially diluted from 1 ng/ μ L to 10^{-1} , 10^{-2} , 10^{-3} , 10^{-4} , 10^{-5} , 10^{-6} , and 10^{-7} ng/ μ L using equally proportional dilution method, and RPA was performed for each dilution. The negative control template was substituted with DEPC-treated water. The diluted DNA templates were then used for both SHERLOCK fluorescence detection and lateral flow immunochromatographic strip detection. Each concentration was tested in triplicate to observe and record the fluorescence curve changes and detection lines.”

Discussion

Comment 1: The content is lengthy and there is little discussion related to the aim of the study.

Response: Thanks for your advice. We have made adjustments to the introduction and discussion sections. Because there is no research on structural genomics, functional genomics and comparative genomics analysis of Hainan BP strain using third-generation sequencing technology, this study first used the PacBio sequencing platform to obtain the high integrity and accuracy of Hainan endemic BP strain. The full genome sequence was completed to complete structural genomics, functional genomics and comparative genomics research. On this basis, crRNA targeting specific conserved gene sequences in the genome of endemic Bp strains in Hainan Province

was further designed and a SHERR-PCR reaction system was constructed.

Re: Spectrum00592-25R2 (**Whole Genome Study and Construction of SHERLOCK Detection Method for Endemic Strains of *Burkholderia pseudomallei* in Hainan based on Third-generation Sequencing**)

Dear Dr. Xinping Chen:

Your manuscript has been accepted, and I am forwarding it to the ASM production staff for publication. Your paper will first be checked to make sure all elements meet the technical requirements. ASM staff will contact you if anything needs to be revised before copyediting and production can begin. Otherwise, you will be notified when your proofs are ready to be viewed.

Sincerely,

Vittal Prakash Ponraj Ph.D
Editor
Microbiology Spectrum